# Deciphering cell states and genealogies of human haematopoiesis

Chen Weng[1,2,3,4,5], Fulong Yu[1,3,4,17], Dian Yang[2,5,18], Michael Poeschla[1,3,4], L. Alexander Liggett[1,3,4], Matthew G. Jones[2,5,6,7], Xiaojie Qiu[2,5,19], Lara Wahlster[1,3,4], Alexis Caulier[1,3,4], Jeffrey A. Hussmann[2,5], Alexandra Schnell[2,5], Kathryn E. Yost[2,5], Luke W. Koblan[2,5], Jorge D. Martin-Rufino[1,3,4], Joseph Min[2,5], Alessandro Hammond[1,3,4], Daniel Ssozi[4,8], Raphael Bueno[9], Hari Mallidi[9], Antonia Kreso[10], Javier Escabi[11,12], William M. Rideout III[13], Tyler Jacks[13], Sahand Hormoz[4,11,12], Peter van Galen[4,8,14,15], Jonathan S. Weissman[2,5,13 ✉] & Vijay G. Sankaran[1,3,4,16 ✉]

The human blood system is maintained through the differentiation and massive amplification of a limited number of long-lived haematopoietic stem cells (HSCs)[1]. Perturbations to this process underlie diverse diseases, but the clonal contributions to human haematopoiesis and how this changes with age remain incompletely understood. Although recent insights have emerged from barcoding studies in model systems[2–5], simultaneous detection of cell states and phylogenies from natural barcodes in humans remains challenging. Here we introduce an improved, single-cell lineage-tracing system based on deep detection of naturally occurring mitochondrial DNA mutations with simultaneous readout of transcriptional states and chromatin accessibility. We use this system to define the clonal architecture of HSCs and map the physiological state and output of clones. We uncover functional heterogeneity in HSC clones, which is stable over months and manifests as both differences in total HSC output and biases towards the production of different mature cell types. We also find that the diversity of HSC clones decreases markedly with age, leading to an oligoclonal structure with multiple distinct clonal expansions. Our study thus provides a clonally resolved and cell-state-aware atlas of human haematopoiesis at single-cell resolution, showing an unappreciated functional diversity of human HSC clones and, more broadly, paving the way for refined studies of clonal dynamics across a range of tissues in human health and disease.

Haematopoietic stem cells (HSCs), which sustain the lifelong production of blood and immune cells, have broad therapeutic applications and serve as a paradigm for understanding stem cell biology[1]. Recent studies suggest that HSCs are functionally heterogeneous with diverse clonal behaviours[2,3,6,7]. For a deeper understanding of the functional diversity of HSCs it is critical to track clonal and subclonal relationships in haematopoiesis to uncover HSC contributions and behaviours in health, as well as in blood diseases, cancers and the setting of ageing in which HSC functions are frequently perturbed[8,9].

Transplantation assays have demonstrated clonal heterogeneity in HSCs but the relevance to homeostatic haematopoiesis remains unclear[10,11]. In model organisms, genetic labelling of HSCs can be used to investigate steady-state HSC behaviours[12–17] but variability in labelling efficiencies and experimental methods has given rise to contrasting views of how HSC clones contribute to haematopoiesis[3–5,17–19]. Although genetic labelling of human HSCs is possible in rare settings of transplantation during gene therapy trials, such exogenous labels cannot be routinely used in humans[20].

Somatically acquired mutations serve as naturally accumulating barcodes that can be used for retrospective lineage tracing in human samples[21–24]. Recent studies using whole-genome sequencing of colonies comprising differentiated cells derived from single haematopoietic progenitors have advanced our understanding of the clonal dynamics underlying human haematopoiesis[21,25,26]. However, the original cell state

[1]Division of Hematology/Oncology, Boston Children's Hospital, Harvard Medical School, Boston, MA, USA. [2]Whitehead Institute for Biomedical Research, Cambridge, MA, USA. [3]Department of Pediatric Oncology, Dana-Farber Cancer Institute, Harvard Medical School, Boston, MA, USA. [4]Broad Institute of MIT and Harvard, Cambridge, MA, USA. [5]Department of Biology and Howard Hughes Medical Institute, Massachusetts Institute of Technology, Cambridge, MA, USA. [6]Department of Dermatology, Stanford University, Stanford, CA, USA. [7]Center for Personal Dynamic Regulomes, Stanford University, Stanford, CA, USA. [8]Division of Hematology, Brigham and Women's Hospital, Department of Medicine, Harvard Medical School, Boston, MA, USA. [9]Division of Thoracic and Cardiac Surgery, Brigham and Women's Hospital, Boston, MA, USA. [10]Division of Cardiac Surgery, Massachusetts General Hospital, Boston, MA, USA. [11]Department of Systems Biology, Harvard Medical School, Boston, MA, USA. [12]Department of Data Science, Dana-Farber Cancer Institute, Boston, MA, USA. [13]Koch Institute For Integrative Cancer Research at MIT, MIT, Cambridge, MA, USA. [14]Department of Medicine, Harvard Medical School, Boston, MA, USA. [15]Ludwig Center at Harvard, Harvard Medical School, Boston, MA, USA. [16]Harvard Stem Cell Institute, Cambridge, MA, USA. [17]Present address: State Key Laboratory of Respiratory Disease, Guangzhou Medical University, Guangzhou, P.R. China. [18]Present address: Department of Molecular Pharmacology and Therapeutics, Department of Systems Biology, Vagelos College of Physicians and Surgeons, Columbia University, New York, NY, USA. [19]Present address: Department of Genetics and Computer Science, BASE Research Initiative, Betty Irene Moore Children's Heart Center, Stanford University, Stanford, CA, USA. ✉e-mail: weissman@wi.mit.edu; sankaran@broadinstitute.org

is not preserved with these approaches and such measurements are critical to showing how cell states impact the behaviour and contributions of HSCs and other cell types to haematopoiesis. Technologies that can simultaneously provide rich cell-state readouts in single cells and yield detailed genealogical information from natural cellular barcodes would, in principle, overcome this limitation. We and others previously demonstrated the potential for mitochondrial DNA mutations to serve as natural cellular barcodes in humans[27–31]. However, existing methods can detect only a limited subset of mtDNA mutations, hampering the ability to resolve fine-scale subclonal relationships and hierarchies.

Here we introduce a new approach, single-cell Regulatory Multiomics (transcriptomics and chromatin accessibility) with Deep Mitochondrial Mutation Profiling (ReDeeM), with approximately tenfold increase in mutation detection rate. We applied ReDeeM to generate a clonally resolved, single-cell transcriptomic and accessible chromatin atlas for around 150,000 human haematopoietic cells from 12 donors, these having being enriched to ensure appropriate coverage of rare haematopoietic stem and progenitor cell (HSPC) populations. Through this approach we define the clonal architecture of human haematopoiesis and also show the contributions of individual HSC clones to overall and lineage-specific output. Finally we assess how these patterns vary with human ageing.

## Single-cell deep mtDNA mutation recovery

A number of features make mtDNA uniquely well suited as a natural evolving barcode, including the compact nature of its genome (roughly 16.7 kb), high copy number (hundreds to thousands per cell) and high rate of spontaneous mutations (estimated to be ten- to 100-fold greater than nuclear DNA)[27,32,33]. Accordingly there have been a number of efforts to utilize mtDNA mutations as endogenous, evolving cellular barcodes for lineage tracing and clonal inference that have provided insights into processes such as studies of blood cancers[28,30,34]. However, the resolution of the resulting phylogenetic analyses has had limitations. The ability to detect rare mtDNA mutations found in specific subclones is hampered by challenges in discrimination of sequencing artefacts from true variants. To improve our ability to call a fuller set of mtDNA mutations we sought to use single-molecule consensus correction, which can minimize the impact of sequencing and PCR errors (Methods). We developed ReDeeM by modification of the droplet-based, single-cell multiome of the 10X Genomics platform using whole cells and further optimized protocols that maximize mtDNA coverage while also preserving single-cell RNA sequencing (scRNA-seq) and single-cell assay for transposase-accessible chromatin using sequencing (scATAC-seq) library quality (Fig. 1a, Supplementary Fig. 1 and Methods). We designed tiling mtDNA-specific probes for hybridization-based capture (Supplementary Data 1 and Methods). Three separate libraries (mtDNA, ATAC and RNA) were generated for sequencing with matched cell barcodes for downstream integration (Fig. 1a). The cell barcode, plus starting and ending positions of the mtDNA fragments, serve as endogenous unique molecular identifiers (eUMIs) without the need for artificial barcodes (eUMI collision rate of approximately 3%; Methods and Extended Data Fig. 1a). eUMI enables single-molecule consensus error correction, resulting in markedly improved sensitivity and accuracy in variant calling, in turn facilitating the detection of rare mtDNA mutations with low heteroplasmy (Extended Data Fig. 2 and Supplementary Methods). We have developed an open-source computational pipeline (redeemV and redeemR packages) based on eUMIs for consensus mtDNA mutation calling with single-cell multiomic profiling.

As an initial benchmark of ReDeeM we profiled 7,104 human CD34+ HSPCs from a healthy young donor (age 31 years). Deep sequencing of the targeted mtDNA library yielded substantially increased mtDNA fragment coverage (on average, 51.7 mitochondrial genome copies per cell versus 14.3 without enrichment) and an ideal eUMI group size for

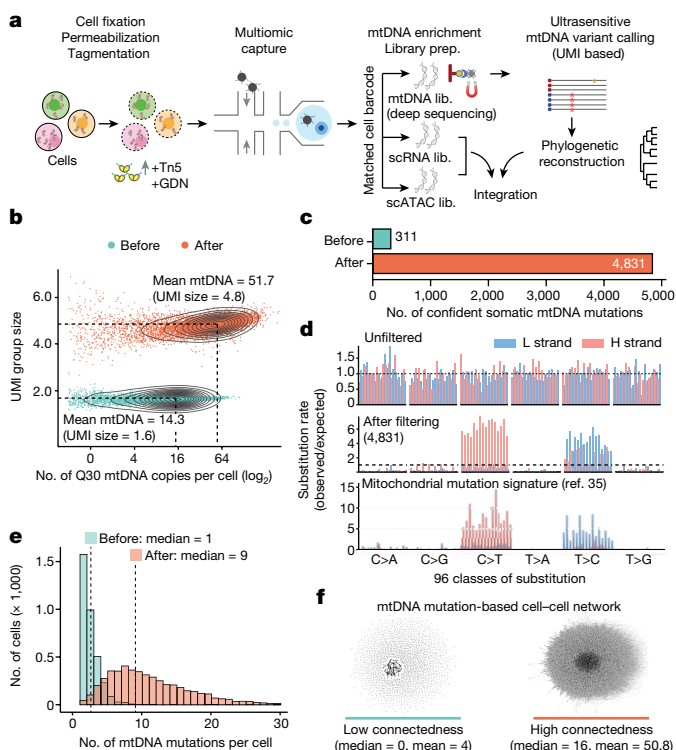

**Fig. 1 | Single-cell deep mtDNA mutation detection with joint multiomics.**
**a**, Schematic of ReDeeM workflow. GDN, 1% glyco-diosgenin (Methods).
**b**, Comparison of mtDNA copy number and UMI group size per cell before and after mtDNA enrichment. UMI group size is the number of raw reads in each UMI group. Q30, sequencing quality score of 30 or above (accuracy ≥99.9%).
**c**, Comparison of the total number of confident mtDNA mutations in 7,104 cells before mtDNA enrichment (via the mgatk package[28]) and after (via UMI consensus calling). **d**, Mutational signatures in each class of mononucleotide and trinucleotide change by heavy (H) and light (L) strands under the optimized protocol. Mutational signatures are compared across unfiltered (top), 4,831 mtDNA mutations via UMI consensus calling (middle) and a previously reported mtDNA mutational signature in bulk (bottom, adapted from ref. 35). **e**, Distribution of the number of confident mtDNA mutations per cell before (via mgatk) and after mtDNA enrichment (via UMI consensus calling). **f**, Network connectedness analysis before (via mgatk, left) and after mtDNA enrichment (via UMI consensus calling, right). Each dot represents one cell and each line connects cells with shared mutations. Connectedness is defined as the number of 'neighbour' cells sharing at least one mtDNA mutation with any given cell. Lib., library.

consensus correction (on average, 4.8 raw reads for each eUMI copy versus 1.6 without enrichment; Fig. 1b). Following stringent multistep filtering we identified 4,831 high-confidence mtDNA mutations across 7,104 cells, which is more than tenfold higher than achieved by previously reported methods[28,29,31] (Fig. 1c, Extended Data Fig. 3, Methods and Supplementary Notes). We further examined these 4,831 mtDNA mutations and validated that they were generally well supported by multiple reads in each eUMI group, having both high consensus scores and consistent overlap between paired-end strands (Extended Data Fig. 1c,e–g). Notably, the mutational signatures of these 4,831 mtDNA mutations closely matched previously reported mtDNA mutational spectra[35] (Fig. 1d). Consequently, each cell presented a far higher number of mtDNA mutations (a median of nine versus one without enrichment) shared by other cells, which increased cell–cell connectedness by one order of magnitude (Fig. 1e,f). This enhanced cell–cell connectedness provides an unprecedented opportunity for fine-scale subclonal and phylogenetic analyses. We benchmarked the data quality of the other two modalities, scRNA-seq and scATAC-seq, from the same cells. Both modalities showed excellent capture efficiency, with a

median of 5,084 transcripts and 15,590 ATAC fragments per cell. ATAC insertions showed the expected size distributions and were highly enriched at transcription start sites (Supplementary Fig. 1c,e,f). Moreover, no significant signatures of selection were identified for most mtDNA mutations, suggesting overall neutrality and enabling these mutations to serve as an innocuous tracer (Extended Data Fig. 4 and Supplementary Notes).

To test the accuracy of phylogenetic reconstructions generated by ReDeeM, we used a Kras;Trp53-drive lung adenocarcinoma lineage-tracer mouse model[36] for detection of both engineered CRISPR-based evolving barcodes in the nuclear genome and naturally occurring mitochondrial somatic mutations by ReDeeM in the same single cells. Across two experimental batches a total of ten tumours were sampled (six in batch 1 and four in batch 2). The measure of cell–cell relatedness and clonal groupings as determined by ReDeeM was significantly supported by CRISPR-based methods at both the single-cell level (median positive agreement of closeness, or agreement of closeness ratio, is 0.78) and clonal cluster level (adjusted Rand index 0.2–0.7 across different clustering resolutions and samples; Extended Data Fig. 5, Supplementary Figs. 2 and 3 and Methods). Furthermore, reanalysis of mitochondrial mutations from single-colony, whole-genome sequencing-based lineage-tracing data[37] showed both clonal and subclonal agreement, albeit with limited sensitivity, compared with that achievable with enhanced mutation detectability by ReDeeM (Extended Data Fig. 6 and Supplementary Notes). These findings are in agreement with a recent report showing agreement in regard to high-frequency mtDNA mutations with whole-genome sequencing of colonies, but with more noise in lower-frequency mtDNA mutations[38] (Supplementary Notes). Taken together, these independent validations support the ability of ReDeeM to robustly detect mtDNA mutations and enable phylogenetic inferences.

## Haematopoietic phylogenies and cell states

We next used ReDeeM to investigate human haematopoiesis. We collected bone marrow aspirates from two healthy young donors aged 31 and 26 years (young-1 and young-2, respectively) and isolated mononuclear cells (predominantly differentiated blood cells and precursors) and CD34+ HSPCs to ensure robust representation of both undifferentiated and more differentiated cells. We profiled 11,009 haematopoietic cells (5,415 bone marrow mononuclear cells (BMMCs) and 5,594 HSPCs) and 15,101 haematopoietic cells (7,147 BMMCs and 7,954 HSPCs) in young-1 and young-2, respectively, for all three modalities (Fig. 2a). We confidently identified 3,896 and 4,803 mtDNA mutations in young-1 BMMCs and HSPCs, and 4,087 and 5,137 mtDNA mutations in young-2 BMMCs and HSPCs, respectively. Based on shared deep mtDNA mutation profiles we reconstructed the phylogenetic trees of each donor's haematopoietic compartment using the neighbour-joining algorithm (Fig. 2b, Supplementary Fig. 5a and Methods). The resulting trees, which were well supported by multiple mtDNA mutations (Supplementary Fig. 4a), were highly polyclonal, consistent with recent phylogenetic analysis based on nuclear genome sequencing of haematopoietic colonies from healthy donors[21,25].

Next we assessed cell state using the transcriptomic and epigenomic information available for each leaf (single cell) in our phylogenetic trees. We used weighted nearest-neighbour (WNN) metrics to integrate both modalities and identified 17 major haematopoietic cell types/clusters (Fig. 2c and Supplementary Data 2). Pairing of scRNA-seq and scATAC-seq profiles from individual cells also enabled us to explore the regulatory circuits in haematopoietic cell fate decisions. For instance, on bifurcation paths between other myeloid lineages and the megakaryocyte/erythroid lineage we observed how master transcriptional regulators SPI1 and GATA1 were turned on with specific regulatory elements and subsequent promotion of differentiation trajectories, characterized by increased accessibility of the transcription factor

motif of one or the other (Fig. 2d). We found that the GATA1 motif begins to be activated earlier during HSC differentiation, even at low GATA1 expression, compared with SPI1, which is consistent with previous studies[39,40]. Interestingly, HSCs show significantly lower mtDNA mutation burden than more committed progenitors and differentiated cells, suggesting that there is acquisition of additional subclonal mtDNA mutations occurring as cells rapidly divide during differentiation from relatively quiescent HSCs[41] which, as discussed below, provides an opportunity to explore phylogenetic relationships between different cell types (Fig. 2e and Supplementary Fig. 5c). Taken together, our data provide a clonally resolved, cell-state-aware atlas of human haematopoiesis at single-cell resolution, allowing previously unachievable inferences on the regulatory mechanisms underlying this complex differentiation process.

## Haematopoietic cell-type origins

The cell-state-aware phylogenetic trees of human haematopoiesis allow us to explore the developmental origins and relationships among different blood and immune cell types, some of which are still incompletely understood. Mapping of multiomic, data-derived cell-type annotations onto the phylogenetic tree showed that different haematopoietic cell populations were widespread across the tree due to the polyclonal origins. Interestingly, however, we also identified many fine-scale subclonal structures, or clades (that is, the full set of cells that descend from a common ancestor and thus encompass a branch of a phylogenetic tree), in which 1,650 and 2,079 clades are significantly enriched for specific cell types (false discovery rate (FDR) < 0.2, fold change > 2) in the two donors, respectively (Fig. 2f, Supplementary Figs. 4b and 5d and Supplementary Data 3). Next we quantitatively assessed cell-type origins using mtDNA mutation-based nearest-neighbour analysis. As expected, the nearest clonal neighbours of most cell types (11 of 13) are identical cell types. Notably, this analysis largely reconstructed the hierarchical organization of blood cell-type origins previously described and characterized extensively in conventional studies of haematopoiesis[1] (Fig. 2g and Supplementary Fig. 5e). However, some unexpected insights emerged from our analysis. For example, it has been challenging to define clearly the progenitor populations that give rise to conventional and plasmacytoid dendritic cells (cDCs and pDCs, respectively)[42,43]. In our data, cDCs and pDCs show less restricted clonal origins and both appear to have a more myeloid-derived origin, which echoes recent lineage-tracing studies in mice[44]. Together, our method resolves clonal and subclonal relationships for native steady-state human haematopoiesis, also linking these relationships with rich readouts of cell state.

## HSC cell-state heterogeneity

Coupling between more closely related clones in the phylogenetic tree and haematopoietic cell states can arise from one of two factors: (1) mtDNA mutations emerging in HSC clones that show lineage bias and (2) mtDNA mutations acquired later during differentiation. The former possibility—or the extent to which HSCs have clonal and functional heterogeneity—is of major clinical importance but remains unclear in regard to native human haematopoiesis. The technical advances we made provide a unique opportunity to address these distinct possibilities, specifically to dissect HSC heterogeneity. To enhance HSC recovery we first enriched for HSCs by deep profiling of the phenotypic CD34+CD45RA−CD90+ population. We then filtered for cells that specifically express HSC marker genes *HLF* and *CRHBP* (Methods, Fig. 3a and Extended Data Fig. 7a–e). We identified 5,393 and 3,292 HSCs in young-1 and young-2, respectively, which were independently validated by examination of the expression of other markers known to be specifically enriched in HSCs, including *MECOM*, *MLLT3* and *RBPMS* (Fig. 3b and Methods). Importantly, to examine the stability of HSC

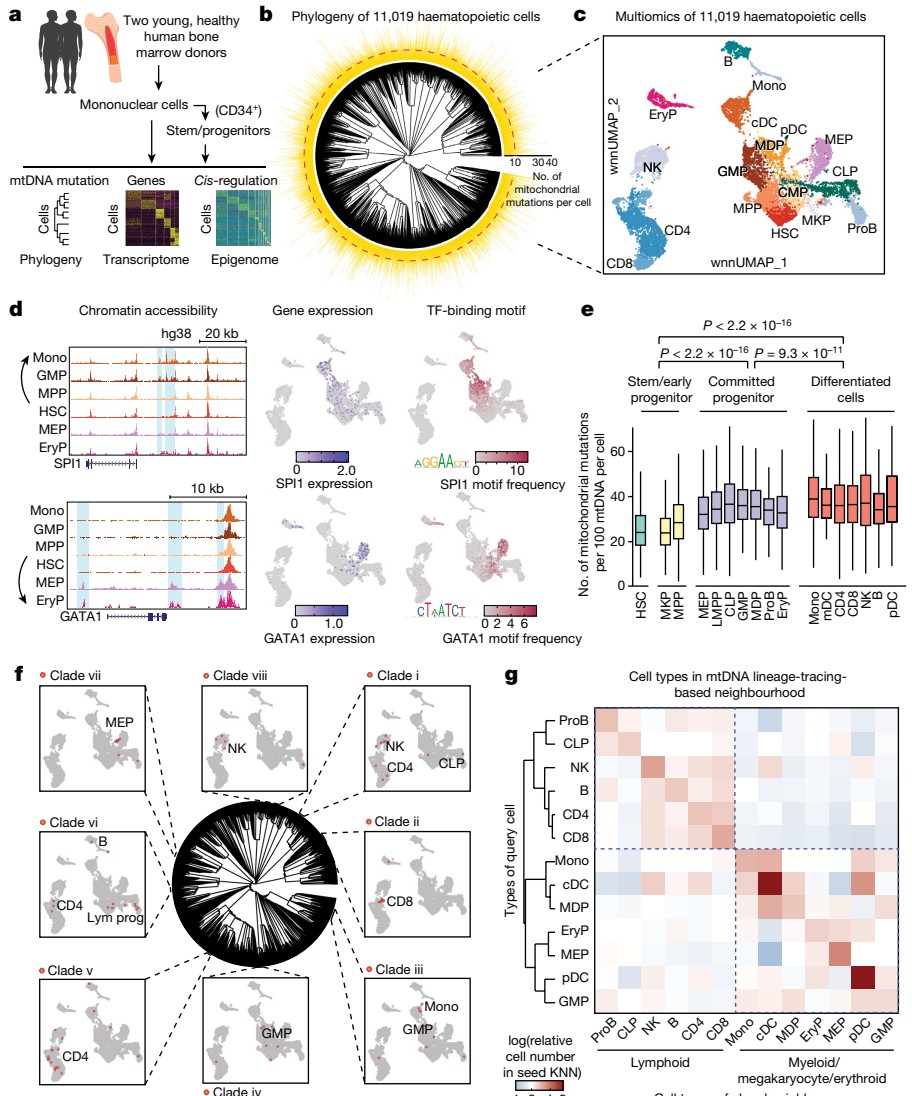

**Fig. 2 | Fine-scale lineage tracing with simultaneous state profiling for human haematopoiesis at steady state. a**, Schematic of the ReDeeM experiment for human haematopoietic cells. **b**, Phylogenetic tree for haematopoietic cells of donor young-1, based on shared mtDNA mutations using the neighbour-joining algorithm. The numbers of shareable mtDNA mutations for each cell are indicated, with a median of ten (cladograms are used for tree visualization in this manuscript). **c**, Joint multiomics clustering of young-1 (for the same cells from **b**). Weighted nearest-neighbour uniform manifold approximation and projection (wnnUMAP) showing combined ATAC and RNA profile for 11,019 single cells. HSC, haematopoietic stem cell; MPP, multipotent progenitor; MKP, megakaryocyte progenitor; CMP, common myeloid progenitor; GMP, granulocyte–monocyte progenitor; MDP, monocyte–dendritic cell progenitor; MEP, megakaryocytic–erythroid progenitor; CLP, common lymphoid progenitor; LMPP, lympho–myeloid primed progenitor; ProB, B cell progenitor; EryP, erythroid precursor; Mono, monocyte; cDC, conventional dendritic cell; pDC, plasmacytoid dendritic cell; NK, natural killer cell. **d**, Analysis of chromatin accessibility (pseudo-bulk

ATAC, left), mRNA expression (middle) and DNA-binding activity (right) of SPI1 and GATA1 transcription factors (TFs) in HSCs differentiating towards myeloid versus mega-erythroid trajectories. Deviation of transcription factor DNA-binding motif frequency was computed using ChromVar based on the JASPAR2020 human transcription factor database. **e**, Measurement of mtDNA mutation burdens across different cell types; $n = 11,019$ cells. Boxplot shows data from the 25th–75th percentile and whiskers extending to the minimum and maximum within 1.5× interquartile range (IQR). $P$ values were derived from two-sided Wilcoxon rank-sum test. **f**, Integrative analysis between phylogenetic tree- and multiomics-based cell types. Examples of cell-type-restricted local clades are highlighted (clades i–viii). Enrichment $P$ values were computed by one-sided binomial test followed by $q$-value correction. **g**, Analysis of cell-type origins based on lineage-informative mtDNA mutations (11,009 cells versus 631 variants). Colour intensity indicates the proportion of each target cell type ($x$ axis) within the mtDNA mutation-based $k$-nearest neighbourhood (KNN) of the queried cell type ($y$ axis).

molecular and behavioural heterogeneity—and thus establish a definitive link between our phylogenetic trees and HSC clonal behaviours—we sampled HSCs twice over the course of 4 months from the same donor (young-1) (Fig. 3a). We further performed unsupervised clustering of HSCs based on WNN space using combined transcriptomic and accessible chromatin states and identified 14 subpopulations in this donor (Fig. 3c). Notably, all subpopulations were consistently identified in both ATAC and RNA space and were reproducibly detected at both

time points (Fig. 3c and Extended Data Fig. 7f). Across HSC subpopulations we identified differentially expressed genes and differential transcription factor accessibility (Extended Data Fig. 7g and Supplementary Fig. 6a). For instance, although overall highly expressed in all HSCs, some key HSC genes, including *MECOM*, *FLT3*, *CDK6*, *JUN* and *FOS*, are differentially expressed across subpopulations (Fig. 3d and Supplementary Fig. 6b). These genes are known to be important in regard to HSC functions, including HSC maintenance, self-renewal,

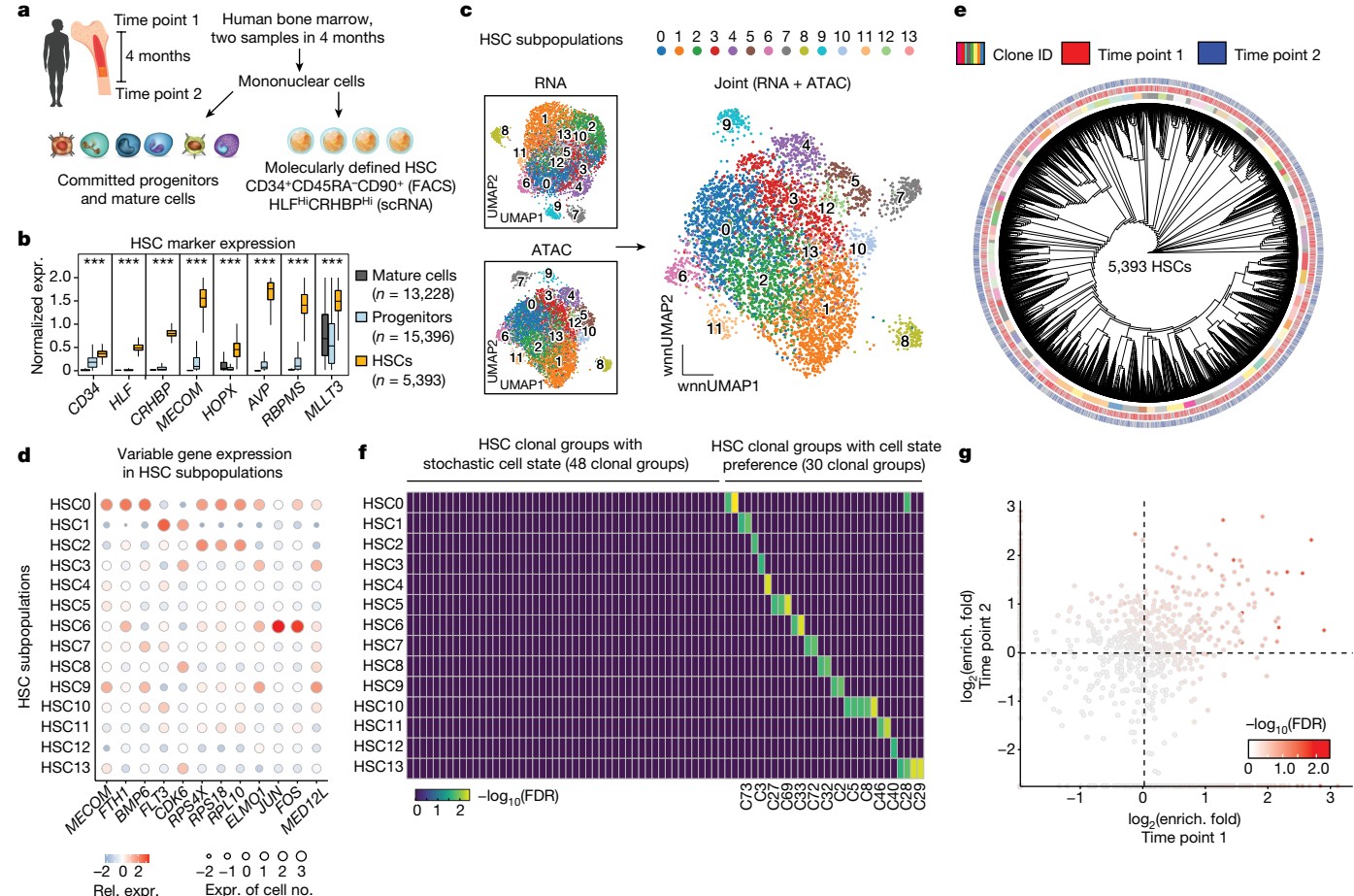

**Fig. 3 | HSC clonal architecture and clonal-dependent cell-state biases.**
**a**, Schematic of the experimental design. Bone marrow samples were obtained from the same individual at two different time points, 4 months apart, and were processed by ReDeeM. HSCs were enriched by fluorescent activated cell sorting (FACS) and further defined by single-cell gene expression (expr.) markers. **b**, Validation of HSC classification. Gene expression of multiple independent HSC markers is shown; $n = 34,017$ cells. Boxplot shows data from the 25th–75th percentile and whiskers extending to the minimum and maximum within $1.5 \times IQR$. ***$P < 2.2 \times 10^{-16}$, derived from one-sided Wilcoxon rank-sum test. **c**, Subpopulations of HSCs based on single-cell RNA and ATAC profiling alone, and on combined WNN space. **d**, Examples of differentially expressed genes across HSC subpopulations. **e**, Phylogenetic tree of HSCs sampled from two time points using shared mtDNA mutations (donor young-1). **f**, Overlap analysis between HSC clonal groups and HSC state subpopulations using hypergeometric tests. Colour intensity indicates combined enrichment FDR (Supplementary Data 4). **g**, Comparison of HSC clone-in-state enrichment (enrich.) (as in **f**) across the two time points; enrichment fold change is compared. Colour intensity indicates combined enrichment FDR.

differentiation and inflammatory responses, and dysregulation of these factors can contribute to leukaemogenesis[45–49]. We also found several differential pathways across subpopulations, with evidence at both the level of gene expression and transcription factor activity changes, such as BMP–SMAD signalling alterations and changes in AP1 signalling (Extended Data Fig. 7g and Supplementary Fig. 6a), reminiscent of previous studies in mice suggesting key roles for these pathways in HSC heterogeneity[50,51]. Notably, we found that the major HSC subpopulations are reproducible in young-2 but we also identified rarer subpopulations found specifically in each individual (Extended Data Fig. 7h–n and Supplementary Fig. 6c). Overall our data provide a multiomic resource that allows us to decipher human HSC heterogeneity.

## HSC clonal structure

Next, based on shared mtDNA mutations across 5,393 molecularly defined HSCs, we reconstructed a phylogenetic tree showing clonal relationships across HSCs. To study HSC clonal features, we defined HSC clonal groups by dividing the tree structure into small clades, which are groups of the most closely related HSC clones (Fig. 3e and

Methods). For clarity the terms 'HSC clones' and 'clonal groups' used hereafter refer to a group of HSCs that share origins during development, rather than referring to individual HSCs. The resulting tree shows a balanced polyclonal architecture of HSCs. In total we defined 78 HSC clonal groups out of 5,393 profiled single HSCs. Notably, the majority of HSC clonal groups can be reobserved in the sequential sampling of the same donor, suggesting that they are representative of HSCs that contribute to haematopoiesis over at least several months in vivo, a timescale over which the majority of non-HSC cell types are believed to have turned over at least once.

It is unclear whether different HSCs have heritable cell states or whether the variation in HSC states represents stochastic, short-lived fluctuations. Our data link clonal identity and cell states from the same cells, and thus we could directly measure the distributions of the 78 HSC clonal groups across the 14 multiomic, cell-state-based HSC subpopulations. We found that 48 (around two-thirds) of HSC clonal groups are stochastically distributed across different HSC states whereas 30 (around one-third) show significant enrichment in one or a small number of specific state subpopulations (Fig. 3f). Interestingly, we found that HSC clone-to-subpopulation enrichment was significantly

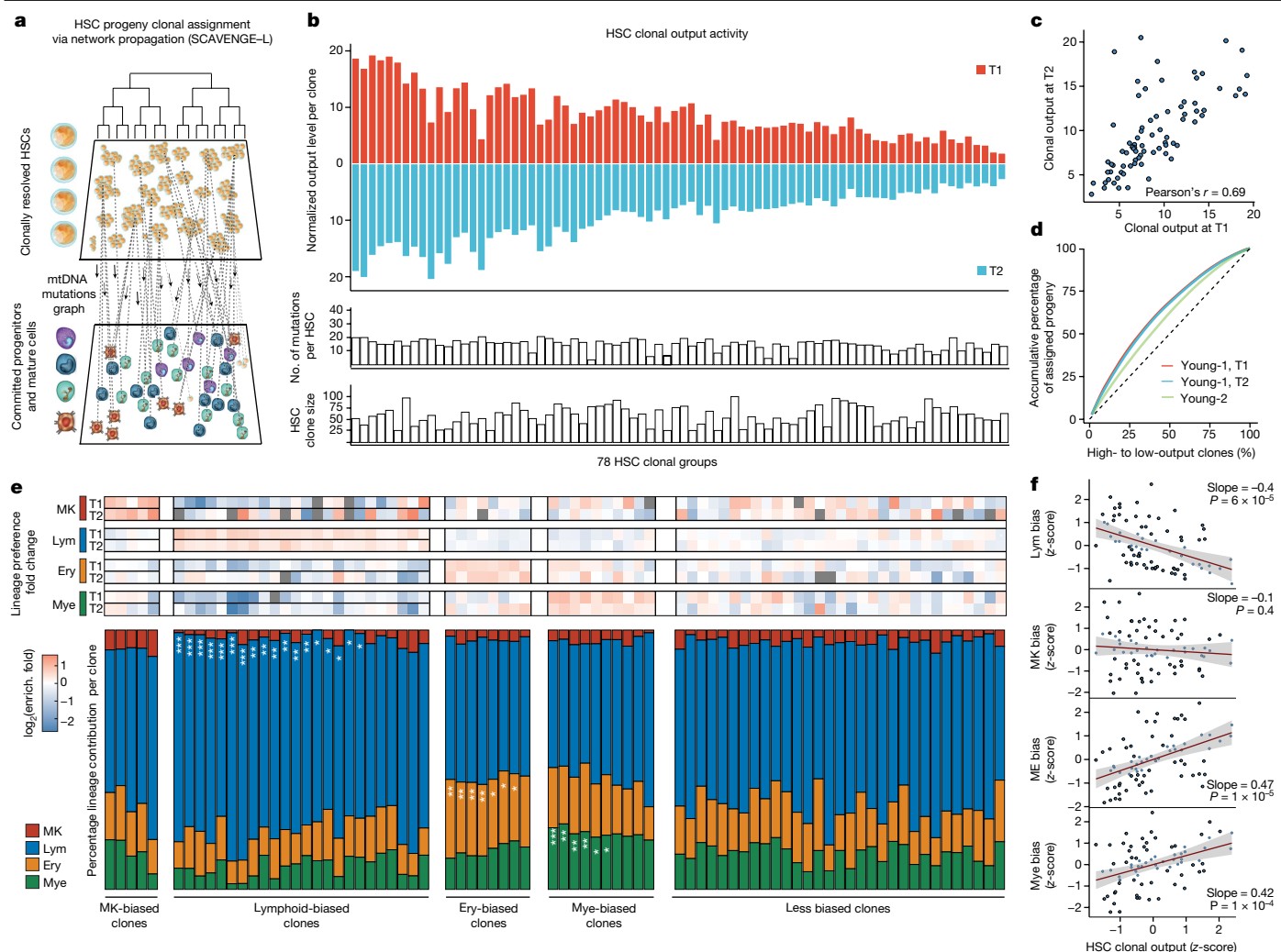

**Fig. 4 | HSC clonal output activity and lineage biases. a**, Schematic of the strategy that assigns progeny cells to HSC clonal groups using network propagation via mtDNA mutation-based cellular networks. **b**, Summary of HSC clonal output activity (number of progeny cells from each HSC clone) across two sampling time points in young-1. Progeny numbers are normalized to HSC clone size. **c**, Correlation analysis of clonal output activity between the two time points (time point 1, T1; time point 2, T2). **d**, Output contribution from each HSC clone, at both time points, in young-1 and young-2, ranked from highest to lowest contribution to total progeny population. Dashed line indicates the expectation of equal contribution from all clones. **e**, For each HSC clonal group the percentages of progeny that differentiate into one of the four main lineages are shown: megakaryocyte (MK), lymphoid (Lym), erythroid (Ery) and myeloid (Mye) cells. Clones with consistent enrichment at both time points are grouped as biased clones. The significance of clonal lineage biases is indicated (FDR *0.05–0.20, **0.01–0.05, ***<0.01; Supplementary Data 4 and Methods). Top, fold changes of clonal lineage biases for each clone are indicated for both time points. **f**, Correlation between HSC clonal output activity and clonal lineage biases. Error bands are 95% confidence level interval for predictions from the linear model. *P* values derived from Wald test.

correlated in the same donor across the two time points, which span 4 months (Fig. 3g), suggesting that HSC biases can be sustained over a period of at least months in humans. We also examined the clonal structure for HSCs in young-2. Consistent with the analysis of young-1, we also observed a polyclonal structure with both stochastic HSC variation and other clones, demonstrating cell-state preference at proportions similar to those observed in young-1 (Extended Data Fig. 7o,p). In sum, we surveyed HSC clones with their molecular states, which suggests a partially heritable and relatively stable state preference for approximately one-third of HSC clones.

## HSC clonal output and cell-type biases

Traditionally, the functional output of HSCs could be measured only in transplantation settings or by barcoding in model systems. Given our advances in detection of deep mtDNA mutations as natural cellular barcodes, we reasoned that tracking of human HSC output in native

haematopoiesis was now possible. To avoid confounding from dropout or detection failure of specific mtDNA mutations, we developed Single-Cell Analysis of Variant Enrichment through Network Propagation of Genomics for Lineage-tracing data (SCAVENGE–L), a computational method based on a network propagation strategy that maximally utilizes informative mtDNA mutations to identify the progeny of distinct HSC clones (Fig. 4a and Methods). With SCAVENGE–L analysis we found that most cells can be mapped to one distinct HSC clonal group with an exclusively high probability of assignment (Supplementary Fig. 7a). To further benchmark the accuracy of this approach we compared network propagation-based assignment with the originally identified HSC clonal groups (ground truth). As expected, the accuracy of the assignment increases for those cells with higher maximum assignment probability. We then filtered cells with a maximum probability of 0.7, by which more than 80% of HSCs could be assigned to the correct HSC clonal group (Supplementary Fig. 7b and Methods). Collectively, the benchmarking analyses at both time points across the two

donors demonstrate the robustness and consistency of SCAVENGE–L (Supplementary Fig. 7b–e).

The extent to which HSC output variation and lineage biases exist is controversial, and most previous studies have relied on labelling in mouse models and/or transplantation assays[3–5,17–19]. In donor young-1, 22,349 (or 59%) committed and differentiated cells were confidently assigned to HSC clonal groups with the highest assignment probability greater than 0.7 (via SCAVENGE–L). The output of a specific clone to differentiated blood and immune cells can then be directly measured and compared across different HSC clonal groups following normalization for clone size (Fig. 4b and Methods). We found that all HSC clonal groups are actively producing progeny but there is some variation in the extent of output between clones, with a 4.9-fold difference in output between the top and bottom clonal deciles (Fig. 4b). Interestingly, this variable output activity shows high consistency at the clonal level between the two time points spanning 4 months (Pearson's $r = 0.69$) (Fig. 4b,c). As expected, in young-2 the HSC clonal output activity also shows variability with a similar pattern (4.5-fold change between top and bottom clonal deciles; Supplementary Fig. 7f). We further quantified overall HSC clonal contributions in haematopoiesis and found that the top 50% of HSC clones based on output gave rise to approximately 60% of mature haematopoietic cells at both time points and across both donors (Fig. 4d). These results suggest that most HSC clones actively contribute to human steady-state haematopoiesis but that some sustained variability over many months is observed between HSCs.

The degree to which HSCs show lineage bias in native human haematopoiesis is unclear. Our data allow us to investigate the cell states of progeny assigned to different HSC clonal groups. For clarity, the terms 'lineage' or 'lineage biases' used in this context refer to the differentiation trajectory based on cell states. We defined four main lineages by grouping cell states based on the multiomic data: myeloid (monocytes, GMP, MDP, cDC), lymphoid (CD4, CD8, natural killer (NK), B, ProB, CLP), erythroid (MEP, EryP) and megakaryocyte (MK) (Fig. 2c). We then computed the lineage contribution for each HSC clonal group. Compared with the expected lineage distribution using all cells, we identified 47 (60%) HSC clonal groups with consistent lineage preference across the two time points, with 31 (40%) HSC clones showing no detectable lineage bias (Fig. 4e, Supplementary Data 4 and Methods). Notably, the lineage preference of biased clones shows only a moderate effect size (with a median 1.55-fold change) but is highly reproducible across the two time points spanning several months (Pearson's $r = 0.59$). Consistently we also observed 69% of lineage-biased HSC clones in young-2 (Supplementary Fig. 7g). When we explored the relationships between clonal output and lineage preference we found the lymphoid lineage bias negatively correlated with HSC clonal output; erythroid and myeloid lineages were positively correlated with HSC clonal output, and MK lineages showed no significant difference (Fig. 4f). This is consistent across both donors and with findings using orthogonal approaches from previous reports (Supplementary Fig. 7h)[17]. Finally we developed a method for 'clonal behavioural trajectory analysis' to survey the potential molecular drivers of distinct clonal functions in terms of output activity and differentiation biases (Extended Data Fig. 8a). We identified multiple accessible regions, but not gene expression changes, that are significantly associated with one or more behavioural trajectories (2,931 differential peaks, FDR < 0.01; Extended Data Fig. 8b and Supplementary Data 5). We investigated nearby genes for peak groups associated with different biases by gene set enrichment and motif analyses (Extended Data Fig. 8c–e). Interestingly, the functions of these nearby genes are reminiscent of the respective output and lineage biases examined, which suggests that chromatin accessibility variation might foreshadow fate decisions in HSCs, echoing previous reports[12,52]. Taken together, these results suggest that HSCs have moderate, but relatively stable, lineage biases across time in native human haematopoiesis.

## Oligoclonal expansions in ageing

Recent studies have suggested that there is both attrition of HSCs with age and expansion of specific clones that harbour disease driver mutations, which can increase the risk for acquiring leukaemia and other morbidities, a phenomenon termed clonal haematopoiesis[8]. However, the detection of such clonal expansions has mostly relied on monitoring of specific driver mutations with bulk-sequencing methods and so the extent of clonal complexity that would be observed at single-cell resolution remains unstudied. To explore this question we used ReDeeM to profile 9,519 and 14,715 haematopoietic cells from two older donors, aged 76 and 78 years, which we termed aged-1 and aged-2, respectively. We detected a significantly increased mtDNA mutation burden in these aged donors across all identified cell types, consistent with reports on somatic mutations in nuclear genomes[25,53] (Fig. 5a and Methods). Based on shared mtDNA mutations, we reconstructed the phylogenetic tree for each aged donor. Remarkably, the resulting trees exhibit markedly more oligoclonal structure compared with that of the young donors (Fig. 5b,c). We identified 48 and 84 clonal groups for aged-1 and aged-2, respectively, by reducing the phylogenetic tree structure using the same methods (Methods). The aged donors had several large clones that dominated the haematopoietic architecture, with lower clonal diversity (Shannon diversity index) compared with the young donors, which was further confirmed by analysis of five additional young and three additional aged donors in a hashed and pooled manner (Fig. 5d,e, Extended Data Figs. 9c and 10d–g and Methods). For examination of subclonal dynamics we adapted a statistical test to quantify clade size relative to that expected if HSCs were evolving under a neutral evolution model (Methods). We identified multiple expanded clades in the aged donors (those of a size greater than 500 cells under positive selection with $P < 0.01$), which were almost completely absent in the young donors (Extended Data Fig. 9a,b). The proportions of cells in the expanded clades were 34.4% in aged-1 and 46.3% in aged-2 but only 3.4% in young-1 and 8.7% in young-2. Next we inferred the 'fitness score', which is defined as the growth advantage compared with the remainder of the population, for every single cell in the aged donors (Methods). These analyses show variation in single-cell fitness within the same donor. As expected, cells in expanded clades showed high fitness scores (Fig. 5f and Extended Data Fig. 9d).

Haematopoietic mosaic loss of the Y chromosome (mLOY) is frequently observed with ageing in men and is associated with a number of morbidities. However, the causes and consequences of mLOY are unclear[54]. Here, based on single-cell ATAC fragments on chromosome Y, we developed quantitative metrics for estimation of LOY in single cells (Methods). We identified 119 and 11 cells with LOY in aged-1 and aged-2, respectively, but none in young male donors (Extended Data Fig. 9i,j). For aged-1 we mapped the identity of cells with or without LOY on the phylogenetic tree, finding that LOY cells appear in multiple branches but are most significantly enriched in expanded clade A, which shows the highest fitness scores. Interestingly we also identified other expansions, such as expanded clade B without LOY enrichment but which is probably caused by a different driver (Fig. 5f). These results suggest that, in aged-1, LOY events occur with low frequency but may occur independently multiple times and are enriched in cells with higher fitness scores, consistent with previous reports[25]. An important caveat to this analysis is that the detection of LOY using single-cell ATAC fragments is limited by the scarcity of accessible reads on chromosome Y.

Finally we investigated cell-type composition within each expanded clade, which is enabled by the joint multiomic readouts available through our method. We found that different expanded clades showed skewed cell-type distribution in both aged donors. This finding is further supported by analysis of the additional, aged, donors (Fig. 5g and Extended Data Figs. 9e and 10h–j). Interestingly, the expanded clade A in aged-1 that showed enrichment for LOY is biased towards the lymphoid lineage, which echoes our recent analysis using bulk population

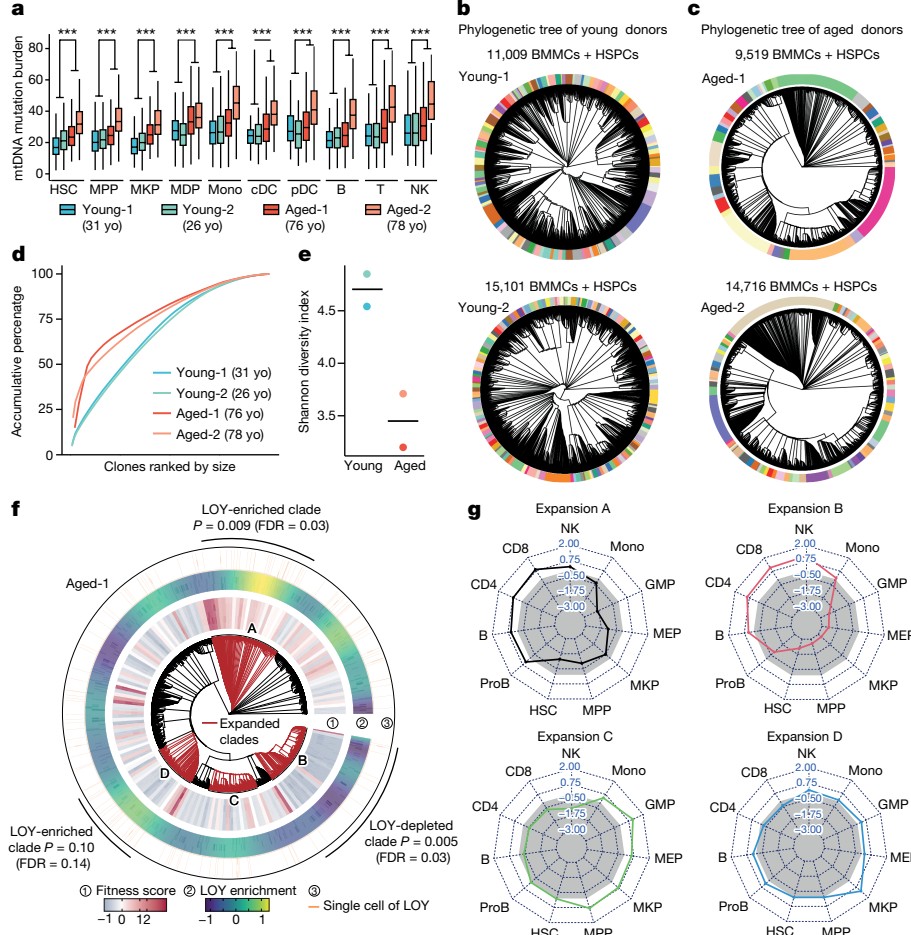

**Fig. 5 | Clonal structure alterations in human haematopoiesis with ageing. a**, Comparison of mtDNA mutation burden between young and aged donors across different cell types. $n = 11,009, 15,101, 9,519$ and $14,715$ cells in young-1, young-2, aged-1 and aged-2, respectively (yo, years old). Boxplot shows data from the 25th–75th percentile and whiskers extending to the minimum and maximum within $1.5\times$ IQR. ***$P < 2.2 \times 10^{-16}$, derived from one-sided Wilcoxon rank-sum test. **b**,**c**, Phylogenetic trees from young (**b**) and aged donors (**c**). Clonal groups are indicated by different colours on the outer rings.

**d**, Contribution of each clone to the total population, in two young and two aged donors. **e**, Shannon diversity index of clonal composition between young and aged donors. **f**, Mapping single-cell fitness score and cells with LOY on the phylogenetic tree for aged-1. Raw and smoothed LOY cell distribution is shown in outer rings. $P$ values (and FDR using $q$-value) of LOY enrichment (one-sided binomial test) are shown. **g**, Cell-type contribution in each expanded clade. Grey area indicates the expected balanced cell-type distribution.

data showing a strong correlation between LOY and individual lymphocyte counts[45]. In addition, in one of the additional aged donors (aged-5) with a known clonal haematopoiesis mutation detected in bulk (ASXL1-Q373X), the identified expansions were depleted for erythroid cells, which is reminiscent of the phenotype observed in Asxl1 mutant mouse models (Extended Data Fig. 10j)[55,56]. Further incorporation of single-cell genotyping with ReDeeM in the future will be valuable in regard to definitive determination of clones with driver mutations and definition of the underlying molecular mechanisms for the observed expansions[57,58]. Collectively these results reshape our view of aged haematopoiesis and, rather than detecting a single clonal expansion as is typically thought to occur with age-related clonal haematopoiesis, we detected a more complex and pervasive oligoclonal architecture.

## Discussion

The study of human haematopoiesis has served as a paradigm for our understanding of stem cell biology. Despite decades of effort, central questions on human haematopoiesis remain unresolved. For example, the extent to which the models 'clonal succession' (only a few stem cells contribute) versus 'clonal stability' (many stem cells

contribute simultaneously)[59,60] best describe native haematopoiesis is unclear, as is the extent to which unperturbed populations of HSCs have restricted differentiation potency or lineage biases[6,61]. Various transplantation-based assays, as well as cell labelling-based methods, have provided important insights but with respective limitations, especially for the exploration of these problems in a native human context[3,14,62,63].

Here we present a high-resolution, engineering-free, massively parallel, single-cell lineage-tracing approach with direct application to human samples. Using this approach we provide a clonally resolved and cell-state-aware single-cell atlas for native human haematopoiesis and use this atlas to explore the clonal architecture and heterogeneous behaviour of human HSCs at steady state in vivo. We show, that in young individuals, the majority of HSC clones are actively contributing to haematopoiesis at steady state but with some differences (around fivefold) in clone-specific output activity, and that these differences are stably maintained over a timescale of at least several months. We also demonstrate that there are inherent clone-specific lineage biases that, like the clonal differences in output, are confined in magnitude but sustained across time. Finally we identify HSC subpopulations using joint transcriptomic and epigenomic states and find that a notable subset of HSC clonal groups are enriched in certain HSC subpopulations

as defined by gene expression and epigenomic states. Interestingly, we found that the HSC clone-specific cell-state preference in the human native context is also an inherent feature that is relatively stable, which echoes some findings using labelling-based methods in mice[3,64]. Of note, we describe behavioural and cell-state biases for HSC clonal groups, which share common ancestors, rather than for individual HSCs. Due to the limited sampling of cells in bone marrow aspirates, the HSCs in a clonal group may not be the most immediate siblings. Therefore, further improved sampling by increasing cell numbers, locations and time points will provide an improved view of the phylogenetic relationships and is crucial to identifying mechanisms underlying the observed cell-state and behavioural biases for more recently derived clonal groups, and even single human HSCs.

Thus, together with previous studies, a picture of normal haematopoiesis emerges from our work in which, in young individuals, there is a rich and balanced polyclonal architecture for HSC contributions to haematopoiesis, with each subclone having distinct but confined preferences in cell-state, output and lineage biases. By contrast, in aged individuals there is a marked breakdown in this clonal diversity. Clonal expansion, or the alteration of clonal diversity, is involved in various cancers and premalignant conditions. However, the causes and consequences of diminished clonal diversity are largely unknown and difficult to study in humans. Our results suggest that clonal expansions may arise with multiple origins and different lineage biases. Our ability to capture and characterize clonal expansions at single-cell resolution in ageing should enable the in-depth exploration of the molecular nature of these expanding clones.

More broadly, somatic mutations have increasingly been found to contribute to a variety of disease processes beyond haematopoiesis and cancer[65]. Compared with single-colony or single-cell whole-genome sequencing, ReDeeM markedly enhances mtDNA mutation detectability through consensus error correction and also provides comprehensive cell-state information. It offers high scalability and significantly reduces the cost per cell, facilitating extensive exploration of subclonal changes in human health and disease. Future advances aiming to improve phylogenetic inference with ReDeeM that consider the unique dynamics of mitochondrial genomes and other biological features will enable improved lineage tree reconstruction, paving the way for a deeper understanding of how clonal mosaicism can contribute to a diverse range of human diseases.

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

## Methods

### Bone marrow donors

Fresh bone marrow samples from healthy young donors were aspirated, with informed consent, under a sample-banking protocol approved by the Institutional Review Board of Boston Children's Hospital. Sternal bone marrow from aged donors was collected following sternotomy for cardiac surgery, with informed consent, under a sample-banking protocol approved by the Institutional Review Board of Mass General Brigham. Individual donor information is shown in Supplementary Table 1.

### Primary BMMC extraction

Bone marrow was collected from healthy young and aged donors. Bone marrow aspirates were diluted with an equal volume of wash buffer (PBS, 2% fetal bovine serum (FBS), 1 mM EDTA). Ficoll medium was added to SepMate tubes (STEMCELL Technologies, catalogue no. 85460) and the diluted bone marrow sample was then layered on top followed by centrifugation at 1,200g for 20 min at room temperature. The top layer, containing the mononuclear cells, was transferred to a new tube which was then filled up by wash buffer. Mononuclear cells were centrifuged at 300g for 8 min. The supernatant was discarded and cells washed twice and resuspended in either wash buffer for further enrichment or freezing buffer (10% DMSO in FBS).

### Enrichment for HSPCs

Starting with BMMCs isolated from the previous step, we enriched CD34[+] cells with the EasySep Human Cord Blood CD34 Positive Selection Kit II (STEMCELL Technologies, catalogue no. 17896). Briefly, EasySep Human CD34 Positive Selection Cocktail (STEMCELL Technologies, catalogue no. 18096 C) was added to the BMMC suspension up to 100 µl ml$^{-1}$ with incubation at room temperature for 10 min. EasySep Dextran RapidSpheres (STEMCELL Technologies, catalogue no. 50100) was vortexed and added to each sample up to 50 µl ml$^{-1}$ and the mix incubated for 3 min at room temperature. Next, wash buffer (7 ml) was added to the tube and cells were washed four times in The Big Easy EasySep Magnet (STEMCELL Technologies, catalogue no. 18001). Finally, cells were resuspended in wash buffer and centrifuged at 300g for 10 min. The CD34[+] cell pellet was then resuspended in freezing buffer (10% DMSO in FBS).

For further enrichment of HSCs, an aliquot of enriched CD34[+] cells was stained by one of the following antibody panels: (1) CD34 PerCP-Cy5.5 (BD Biosciences catalogue no. 347222), CD45RA Alexa Fluor 488 (BioLegend catalogue no. 304114) and CD90 PE-Cy7 (BD Biosciences catalogue no. 561558) with DAPI (Thermo Fisher Scientific catalogue no. D1306) as viability dye; or (2) CD34 BV421 (BD Biosciences catalogue no. 562577), CD45RA-APC-H7 (BD Biosciences catalogue no. 560674) and CD90 PE-Cy7 (BD Biosciences catalogue no. 561558) with 7-AAD as viability dye (BD Biosciences catalogue no. 559925). This was followed by 3 µl of each antibody being used for staining of the cell resuspension in 100 µl. Cells were further sorted with BD FACSAria for CD34[+]CD45RA[−]CD90[+] to enrich HSCs. The gating strategy is shown in Supplementary Information.

BMMCs, as well as enriched CD34[+] and CD34[+]CD45RA[−]CD90[+] cells, were cryopreserved in freezing buffer (10% DMSO in FBS). Following thawing, cells were immediately processed for experimental use as soon as possible with no culturing.

### Principle of ReDeeM

Here we developed ReDeeM, which is a modified, massive parallel single-cell protocol that simultaneously profiles multiomics with deep mtDNA sequencing based on the 10X Genomics platform. The key features of this system are the following: (1) an optimized protocol for maximization of mtDNA yield; (2) specific enrichment of the mtDNA library that can be subject to very high sequencing coverage; (3) unique molecular identifiers that label individual mtDNA molecules, allowing for the use of error correction to enable high-precision calling of mtDNA mutations[66–70]; (4) a robust inference algorithm that uses deeper and improved mtDNA mutation detection for phylogenetic reconstructions; and (5) concomitant scRNA-seq and scATAC-seq that link phylogenetic relationships with cell-state readout. With ReDeeM three separate libraries are generated, including an enriched mtDNA library for deep sequencing and mutation profiling, a RNA library for gene expression and an ATAC library for chromatin accessibility profiling, all of which are linked via matchable single-cell barcodes,

Following the principle of our previous work[28,29], we first modified the droplet-based 10X Genomics multiomics protocol (catalogue no. 100283) by processing the whole cell, rather than nuclei, with fixation and mild permeabilization for maximal retention of mtDNA. Next we designed mtDNA-specific probe sets to enrich mitochondrial fragments using DNA hybridization. RNA and ATAC library preparation followed the standard 10X Genomics protocol, with some modifications.

Further method details are described in Supplementary Methods and the ReDeeM protocol. ReDeeM is further computationally supported by the consensus variant-calling pipeline redeemV, as well as the R package redeemR for downstream mutation quality control and single-cell phylogenetic and integrative analysis.

### ReDeeM protocol

The detailed protocol is available as a Supplementary Protocol.

### CRISPR lineage-tracing experiment with ReDeeM

Mouse experiments were approved by the Massachusetts Institute of Technology Institutional Animal Care and Use Committee (Institutional Animal Welfare Assurance, no. A-3125-01). A male mouse ESC line harbouring the conditional alleles KrasLSL-G12D/+ and Trp53fl/fl was engineered with lineage-tracer cassettes. The detailed engineering process, including vector information, tumour harvest and single-cell suspension, was prepared as described in ref. 36. Two independent mouse ESC lines were used for batch 1 and batch 2 experiments.

The single cells of batch 1 (six tumours) and batch 2 (four tumours) were labelled with Cell Hash and profiled using ReDeeM except for the following modification: additional target site libraries were needed. Amplified cDNA libraries were further amplified with target site-specific primers containing Illumina-compatible adaptors and sample indices (oDYT023-oDYT038, forward: 5′CAA-GCAGAAGACGGCATACGAGATNNNNNNNNGTCTCGTGGGCTCGGAG ATGTGTATAAGAGACAGAATCCAGCTAGCTGTGCAGC; reverse: 5′-AAT GATACGGCGACCACCGAGATCTACACNNNNNNNNTCTTTCCCTACAC GACGCTCTTCCGATCT; N denotes sample indices) using Kapa HiFi ReadyMix (Roche), as previously described[36].

For sequencing of scRNA, scATAC and mtDNA libraries the strategy described for ReDeeM was used except that four sets of mouse-specific probes were designed to enrich mitochondrial fragments (Supplementary Methods and Supplementary Data 1). For sequencing of target site libraries, 15,000 total reads per cell were expected and the following read lengths were used: Read1, 26 cycles; i7, eight cycles; Read2, 290 cycles).

The integration analysis of CRISPR- and ReDeeM-based lineage tracing is detailed in Supplementary Methods.

### mtDNA mutation burden

We estimated mtDNA mutation burden using a quantitative method. The number of detected mutations per cell is a function of both biological mutation burden and technical detectability, which is influenced by mtDNA capture rate. We computed mtDNA mutation burden by both normalization against mtDNA coverage (number of mtDNA copies per position per cell) and eUMI filtering rate, which was used to correct technical batch effects across different experiments arising from

variation in sequencing depth, sequencing quality and so on. Given a single cell $i$ in sample $j$, the mutation burden is computed as

Mutation burden

$$= \frac{\text{no. of mtDNA mutations} | \text{cell } i}{(\text{eUMI filtering rate} | \text{sample } j) \times (\text{mtDNA coverage} | \text{cell } i)}.$$

## Inferring lineage distance and phylogenetic tree using mtDNA mutations

Following all the filtering steps with R package ReDeeM-R (https://github.com/sankaranlab/redeemR), including variant and cell filtering (Extended Data Fig. 1i; all parameters included can be adjusted to control stringency), we generated sparse matrix $C$ to contain all the variant allele count (cell versus mtDNA mutation). The allele count matrix was further divided by the matrix of mtDNA copy number per position per cell, which generated heteroplasmy matrix $H$ for visualization. Because mutation count data were sparse, the quantitative heteroplasmy level was susceptible to variation in mtDNA coverage. To minimize biases of coverage and heteroplasmy dynamics in the downstream analysis we performed binarization of matrix $C$ into matrix $C_{bin}$. We found that binarization is more reliable and provides sufficient resolution, given the number of variants identified per single cell. Nonetheless, both quantitative matrix $C$ and binarized matrix $C_{bin}$ are provided for downstream analysis in ReDeeM-R.

Based on matrix $C_{bin}$, we computed the cell-to-cell weighted Jaccard distance. The prior of the mtDNA mutation frequency across multiple donors was used to weight Jaccard distance to account for potential homoplasy. Intuitively, weighted Jaccard distance measures the level to which any two cells share mutations—that is, following proper normalization, the more mtDNA mutations are shared the closer the relationship of the two cells. We first defined a prior probability for each mutation, which prioritizes mutations with lower mutation rate across donors (that is, less likely to be the same mutation occurring independently). For cells $x$ and $y$ the weighted Jaccard distance ($D_{w\_Jaccard}$) is defined as

$$\text{Prior}_i = (1 - \text{average mutation rate across donors})$$

$$D_{w\_Jaccard} = 1 - \frac{\sum_{i \in [x_i = 1 \& y_i = 1]} \text{prior}_i}{\sum_i \text{prior}_i}.$$

Next, the weighted Jaccard distance was fed into the neighbour-joining algorithm for phylogenetic tree reconstruction and visualization using the packages ape and ggtree (cladograms are used for visualization throughout this manuscript, to focus on the topology of the tree structure).

## Lineage origins of haematopoietic cell types

Initially we selected 'lineage-informative' mtDNA mutations by modelling mutation distributions across all cell types. We removed mutations randomly distributed, which probably arose in certain unbiased stem cell clones and thus were less informative in regard to studying cell-type subclonal origins. Specifically we first grouped all cell types into four major differentiation trajectories: myeloid (GMP, MDP, monocyte), lymphoid (CLP, ProB, CD4, CD8, B, NK), MKs (MK progenitor) and erythroid (MEP, erythroid progenitor). The frequency of each mtDNA mutation was tested between any two differentiation trajectories using a binomial test. When $P$ values of all comparisons were greater than 0.05, mtDNA mutation was defined as randomly distributed. We filtered out all randomly distributed mutations and generated a list of lineage-informative mtDNA mutations (631 lineage-informative mutations are used in Fig. 2g). Using these mutations we generated matrix $C_{bin}$ and computed weighted Jaccard distance. We then generated KNN graph $G$ that describes cell-to-cell lineage relationships based on

shared mutations. We then integrated cell-type annotations from the multiomics analysis with graph $G$. For any given cell (query cell), the proportion of each cell type (target cell types) within KNN on graph $G$ was computed. Target cell-type proportions for each query cell type were then aggregated and scaled, as shown in Fig. 2g and Supplementary Fig. 5. Finally, query cell types were grouped by hierarchical clustering based on target cell-type proportions within neighbourhoods.

## HSC subpopulations and clone-to-state preferences

For specific study of HSCs we experimentally enriched the $CD34^+CD45RA^-CD90^+$ population as described previously. We further refined HSC populations using a semi-unsupervised method. First we performed community detection-based clustering for all cells on WNN using Seurat[71]. Second, we averaged $HLF$ gene expression level for each cluster and defined $HLF^{hi}$ and $HLF^{low}$ clusters. Third, we simultaneously examined $HLF$ and $CRHBP$ gene expression levels for every single cell[45,72]. We required that any HSC cell highly expresses both $HLF$ and $CRHBP$ and is also grouped within $HLF^{hi}$ clusters. The defined HSCs were further examined using additional HSC signatures, including $MECOM$, $HOPX$, $AVP$, $MLLT3$, $RBPMS$ and other[45,73,74]. To improve the robustness of weakly expressed genes, expression data were enhanced using the Rmagic package for visualization[75].

For the refined HSCs above we performed secondary clustering on WNN to define subpopulations. These were identified using Seurat at a resolution of 0.6. Subpopulations were visualized on RNA-, ATAC- and WNN-based UMAP. Differentially expressed genes and accessible chromatins were identified using FindMarker function by Seurat. The DNA-binding motifs of differential peaks were analysed by 'find individual motif occurrences' scanning with the HOCOMOCOv11_full_HUMAN_mono human transfection factor motif database, followed by a binomial test across HSC subpopulation-specific open chromatin peaks (related to Supplementary Fig. 6). Visualization of differential motifs at the single-cell level was performed using chromVar[76,77].

To best capture the principal HSC clonal structures we performed normalization and dimension reduction using term frequency–inverse document frequency and singular value decomposition on a binarized mtDNA variant-by-cell matrix. Top 30 latent semantic indexing was used to measure Euclidean distance, which was further passed on to the neighbour-joining algorithm to build the phylogenetic tree. Next, mtDNA mutations were assigned to tree branches using a maximum-likelihood method as described previously, which has been incorporated in redeemR (Add_AssignVariant function)[26]. We defined HSC clonal group as the minimum clade unit containing at least 50 single cells, with based edges having at least one confidently assigned mutation ('edge' refers to a line connecting two nodes in the phylogenetic tree; the Add_tree_cut function from ReDeeM-R was used).

Next we examined the distribution of each HSC clonal group across all HSC subpopulations as defined by RNA- and ATAC-based cell state. Compared with the background, the fold enrichment of a given clonal group in each cell-state subpopulation was computed and the $P$ value estimated by hypergeometric test. Fold enrichment and $P$ values were compared across HSCs from two sampling time points. $P$ values from two time points were combined using Fisher's method, and FDR was computed using the qvalue R package. The cutoff used to define HSC clone-to-cell-state preference is as follows: combined $P < 0.01$ and FDR $< 0.05$ and $\log_2$fold change(time point 1) $> 0.25$ and $\log_2$fold change(time point 2) $> 0.25$. Full statistics are shown in Supplementary Data 4.

## HSC progeny clonal assignment using network propagation

Combining the sampling of HSCs with committed and differentiated progenies in the same donor, we aimed to use the similarity of mtDNA mutation profiles to assign progenies into one of the HSC clonal groups. Briefly we first built an inclusive clonal network using shared mtDNA mutations for all cells from the same donor. Next, HSC cells from each

HSC clonal group served as seeds to propagate clonal information through the clonal network until a stationary state was reached. Each clonal group was used for network propagation iteratively. Following network propagation, the information carried by each cell represents the probability of the assignment for the given HSC clonal group, and normalized probabilities are compared across all clonal groups to determine the final assignment.

Because the mtDNA variant-by-cell matrix is highly sparse, the task of confident single-cell assignment is challenging. Our previous study showed that the phenotypic relevance of individual cells can be faithfully modelled in a cell-to-cell similarity graph and effectively identified by a network propagation algorithm[78], despite the inherent high dimensionality and extensive sparsity nature of single-cell genomics data. Here, using a similar principle, we developed SCAVENGE–L, which uses the network propagation strategy that utilizes clonal neighbourhood information and efficiently assigns cells with probabilistic metrics. We reasoned that the clonal structure of individual cells can be faithfully distilled into a network in which each node represents a cell and each edge represents mtDNA mutation profile similarity among cells. By defining cells of interest (that is, HSC clonal group) we could exploit this network to search highly relevant cells (that is, progeny) using both network topology structure and cell-to-cell distance.

We first generated a fully binarized mtDNA variants-by-cell matrix that included all stem, progenitor and differentiated cells from a given donor. We performed term frequency–inverse document frequency followed by singular value decomposition for normalization and dimension reduction. The top 30 latent semantic indexings were used for construction of a mutual KNN graph (mKNN). Next, we highlighted each HSC clonal group on the mKNN graph then used a random walk-with-restart method to discover the progeny for cells of each HSC clonal group, which we termed seed cells. The information on this mKNN graph can spread across, and the information retained in the network at stationary state can be used to measure, the probability of any given cell belonging to a HSC clonal group (seed cells). We performed network propagation analysis with a damping factor of 0.05 from each HSC clonal group (seed) iteratively. Finally this generated a cell-by-clonal group probability matrix that measured the confidence of assignment. We took the maximum probability of above 0.7 as cutoff to filter out ambiguous progenies (Supplementary Fig. 7a–e).

Because HSCs were also included in the mKNN network and processed with network propagation, they could be assigned to a clonal group using the algorithm via network propagation; meanwhile, the actual HSC clonal group was used as ground truth. By comparison of predicted HSC clonal group with ground truth we managed to benchmark the robustness of SCAVANGE–L before applying it to assigning progenies to HSC clonal groups (Supplementary Fig. 7).

## HSC clonal output and lineage biases

For the study of HSC clonal output activity we collected both HSCs and all differentiated progenies from the same donor across two sampling time points. Based on mtDNA mutations we applied SCAVENGE–L to assign differentiated progenies to each HSC clone. Next we measured clonal output level by counting the number of progenies for each HSC clonal group, followed by normalization with HSC clone size (the number of HSCs per clonal group). We compared clonal output level across the two sampling time points and computed Pearson's correlation. For evaluation of the contribution of haematopoiesis across different HSC clones we ranked them from highest to lowest and computed the cumulative proportion of the differentiated progenies contributed by these clones.

Next, for each HSC clone we computed the proportion of the four main lineages as defined by cell state: myeloid (monocytes, GMP, MDP, cDC), erythroid (MEP, EryP), Meg (MKP) and lymphoid (CD4, CD8, NK, B, ProB, CLP). Lineage biases were modelled by binomial distribution against the background by all cells at two sampling time points. HSC

clones with consistent enrichment fold change at both time points were categorized as biased clones. Enrichment $P$ values at both time points were combined using Fisher's method, and combined $P$ values were adjusted using the R package qvalue as FDR. Enrichment fold change was calculated for each sampling time point independently. Finally, HSC clonal output levels and lineage biases were scaled, and Pearson's correlations were computed to assess the relationship between output activity and lineage biases.

## Clonal expansion analysis in ageing

First we collected both BMMCs and CD34+ HSPCs from two young donors (31-year-old female and 26-year-old male, young-1 and young-2, respectively) and two aged donors (76-year-old male and 78-year-old male, aged-1 and aged-2, respectively). Using the same consensus variant-calling pipeline and neighbour-joining algorithm described previously, we reconstructed the phylogenetic tree for all four donors. Clonal expansions were estimated by two methods: clone- and clade-based. For the former we first identified clonal groups as described above. Briefly, variants were assigned to branches probabilistically and then we cut the tree down on branches having at least $n$ confident variants and with clone group size at least $m$. The parameters involved were $m$ (minimum number of cells in a clone, with default 50), $n$ (minimum number of cumulative variants on the branch to be cut, with default 1), $P$ (probability of the variant being assigned, with default 0.6) and $D$ (dump small clones with fewer than $D$ cells). We compared the distribution of clone sizes between young and aged donors by cumulative proportions. To rule out the potential bias of parameters that define clonal groups, we adjusted the parameter combinations ($m$, $n$, $P$, $D$) and compared clonal size distribution between young and aged donors (Extended Data Fig. 9c). Next, Shannon diversity index, $S$, was also computed for each donor to measure clonal diversity between young and aged donors. Given clone group $i$, $size_i$ is the cell number of that clone. Shannon diversity index is calculated as

$$n = \sum_1^i size_i$$

$$S = -\sum_1^i \frac{size_i}{n} \times \log\left(\frac{size_i}{n}\right).$$

For the clade-based method we identified expansion clades as previously described and implemented them using the function cassiopeia.tl. compute_expansion_pvalues from the Cassiopeia package (available at https://github.com/YosefLab/Cassiopeia)[79]. Briefly, we compared the number of cells included in the subclone with its direct 'sisters' and computed the probability of this observation under neutral selection with a coalescent model. Clades with $P < 0.01$ and at least 5% cells were annotated as expanded clades (Extended Data Fig. 9a). Finally, the proportions of cells contributed by the expanded clades were summarized for each donor (Extended Data Fig. 9b).

## Inferring single-cell fitness

The phylogenetic structure can be used to infer cell fitness[36,80,81]. We applied the function infer_fitness function from the jungle package (available at https://github.com/felixhorns/jungle), which implements a previously described probabilistic method for inferring relative fitness coefficients between samples in a clonal population.

## Analysis of loss of chromosome Y

The loss of chromosome Y was inferred at the single-cell level using the scATAC reads. From uniquely mapped reads using CellRanger-arc (bam file) we first removed PCR duplicates and counted the number of unique fragments per chromosome per cell. The number of chromosome Y fragments per cell was modelled using binomial distribution

out of total fragment numbers. We defined cells of LOY as those with chromosome Y fragment count tenfold lower than expected ($P < 0.001$). Local LOY density was computed as number of cells of LOY/clade size. Enrichment score was computed as the $z$-score of LOY density normalized by total density. Enrichment was further analysed using a one-tailed binomial test.

## Reporting summary

Further information on research design is available in the Nature Portfolio Reporting Summary linked to this article.

## Data availability

All data generated in the manuscript have been deposited in GEO (GSE219015). Processed Seurat objects are available on figshare: https://doi.org/10.6084/m9.figshare.23290004. Processed mutation-calling files are available on figshare: https://doi.org/10.6084/m9.figshare.24418966.v1. Single-colony, whole-genome sequencing data are derived from dbGaP (phs002308.v1.p1). Transcription factor motif database JASPAR2020 (https://jaspar2020.genereg.net/) was used with ChromVar analysis. The HOCOMOCOv.11 (https://hocomoco11.autosome.org/downloads_v11) human transcription factor database was used for 'find individual motif occurrences' analysis.

## Code availability

ReDeeM datasets can be processed by the consensus variant-calling command tool REDEEM-V (https://github.com/sankaranlab/redeemV) and by the inhouse R package REDEEM-R (https://github.com/sankaranlab/redeemR) for downstream phylogenetic and integrative analysis. The reproducibility codes of the analyses included in this work are also provided (https://github.com/sankaranlab/redeem_reproducibility).

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

**Acknowledgements** This work was supported by the New York Stem Cell Foundation (V.G.S.), the Edward P. Evans Foundation (V.G.S.), the Starr Cancer Consortium (V.G.S.), National Institutes of Health (NIH) grant nos. R01CA265726, R01DK103794, R01HL146500 and R33CA278393 (V.G.S.), the Mathers Foundation (V.G.S. and J.S.W.), NCI Cancer Target Discovery And Development (CTD²) and the NIH Centers of Excellence in Genomic Science (NIH no. 1RM1 HG009490-06 to J.S.W.), the Howard Hughes Medical Institute (J.S.W.) and the Ludwig Center at MIT (J.S.W.). J.S.W. was supported by the Howard Hughes Medical Institute and the Ludwig Center at MIT. V.G.S. is a New York Stem Cell Foundation-Robertson Investigator. We thank R. G. Rowe, K. Brundige, M. Fitzgerald, K. Rosen, A. Bass, A. George and S. Edwards Overy for assistance in obtaining bone marrow samples. We thank S. Zhang, C. Nemec, K. Laricchia and J. Yu, as well as members of the Sankaran and Weissman laborarories, for valuable discussions.

**Author contributions** C.W. helped conceive the project, developed ReDeeM, performed experiments and data analysis and wrote the manuscript. F.Y. developed SCAVENGE–L and helped with data analysis. D.Y. provided CRISPR lineage-tracing mice and helped with experiments. M.P. and A.S. helped with ReDeeM experiments. M.P., L.A.L., M.G.J., X.Q., J.A.H., K.E.Y., L.W.K., J.D.M.-R., J.M., P.v.G., J.E., S.H. and A.H. helped with data analysis and interpretation. L.W. and A.C. helped with cell collection and enrichment. D.S., R.B., H.M., A.K. and P.v.G. helped with sample collection. W.M.R. and T.J. helped develop the mouse lineage-tracer model. J.S.W. and V.G.S. conceived the project, supervised and directed the studies and wrote the manuscript.

**Competing interests** C.W., J.S.W. and V.G.S. are listed as inventors on a patent application covering the ReDeeM method. V.G.S. is an inventor on PCT/US2019/036583 covering the application of lineage tracing with mitochondrial genome mutations. V.G.S. serves as an advisor to and/or has equity in Branch Biosciences, Ensoma, Novartis and Cellarity. J.S.W. serves as an advisor to and/or has equity in 5 AM Ventures, Amgen, Chroma Medicine, KSQ Therapeutics, Maze Therapeutics, Tenaya Therapeutics and Tessera Therapeutics. The other authors declare no competing interests.

**Additional information**
**Correspondence and requests for materials** should be addressed to Jonathan S. Weissman or Vijay G. Sankaran.

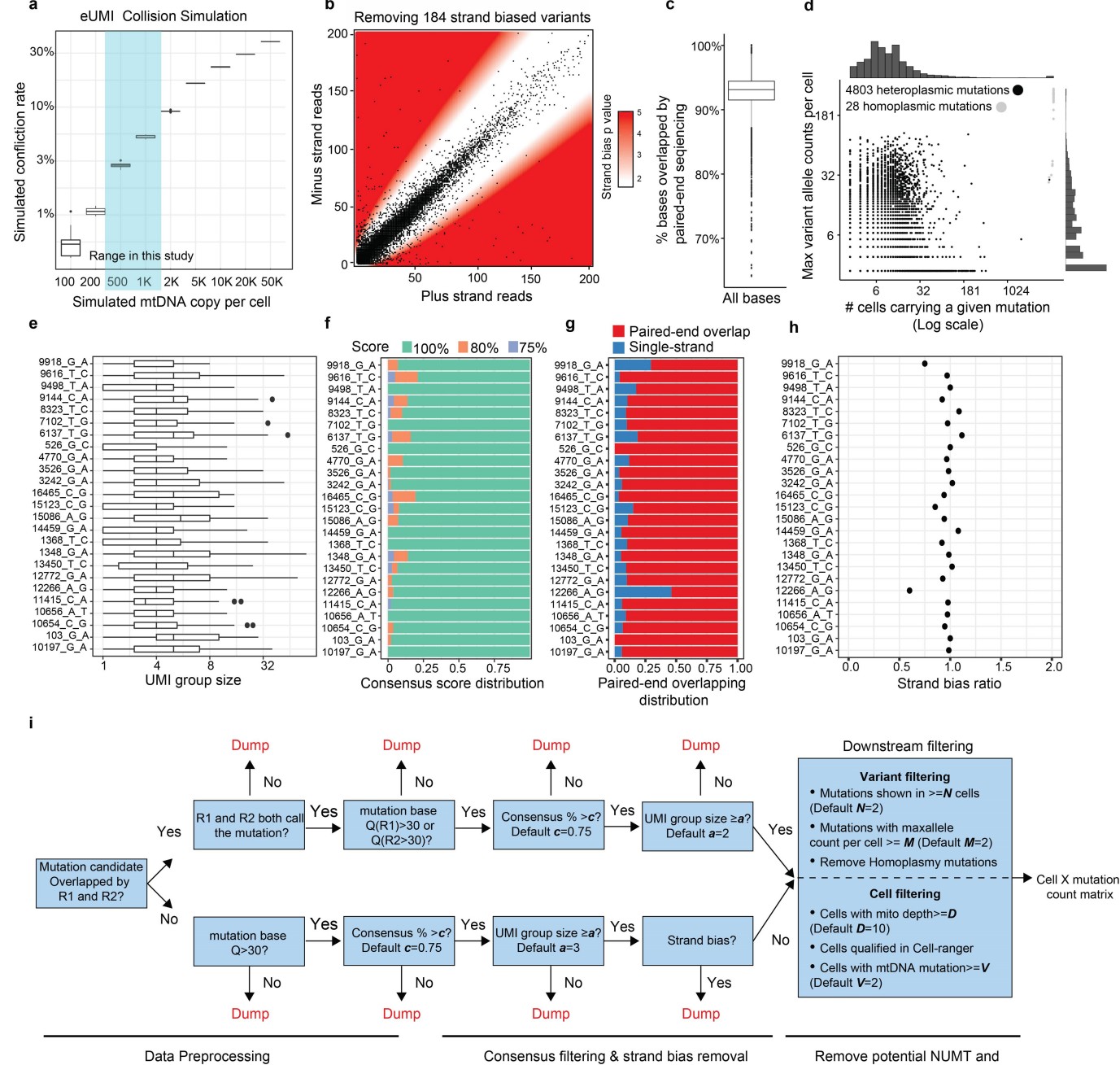

**Extended Data Fig. 1 | Improved mtDNA coverage and mutation detection with ReDeeM pipeline. a**, Justification of the use of endogenous UMI (eUMI) based on cell barcode plus starting and ending sites through simulation. The starting and ending sites were simulated based on empirical distribution of Tn5 cutting sites. The number of mtDNA copies per cell was simulated (x-axis) and the collision rate was calculated accordingly. n = 10 simulations in each box. Boxplot displays data from the 25th to 75th percentile, and whiskers extending to the minimum and maximum within 1.5 IQR. **b-h**, Quality controls for ReDeeM variant calling, related to main Fig. 1. **b**, Removing strand-biased mtDNA variants using a binomial distribution model. White zone includes mutations for which the null hypothesis (no strand biases) holds true, whereas red zone indicates rejection of this null hypothesis (both p < 0.01 and fold change > +/− 2) in favour of strand biases. P-values are derived from two-sided binomial test. **c**, Percentage of bases that were overlapped by both reads in a paired-end sequencing. n = 7,404 cells. Boxplot displays data from the 25th to 75th percentile, and whiskers extending to the minimum and maximum within 1.5 IQR. **d**, Summary of the mutations regarding the number of cells carrying the mutation (x-axis) and the maximum number of mutant alleles per cell (y-axis). Heteroplasmic mutations in black dots and homoplasmic mutations in grey dots. **e-h**, Detailed mutation quality control metrics (random examples are shown). **e**, The number of supporting reads for each mtDNA molecule (UMI group) containing the mutation. Boxplot displays data from the 25th to 75th percentile, and whiskers extending to the minimum and maximum within 1.5 IQR. **f**, Consensus score: the percentage of mutant reads in a UMI group. Mutations with consensus scores of less than 75% were discarded. **g**, For a given mutation, the proportion of molecules that were covered by both reads in a paired-end sequencing. **h**, Strand bias ratio (a ratio of 1 indicating no strand bias). **i**, Workflow of the ReDeeM variant calling pipeline. NUMT: Nuclear mitochondrial DNA segments.

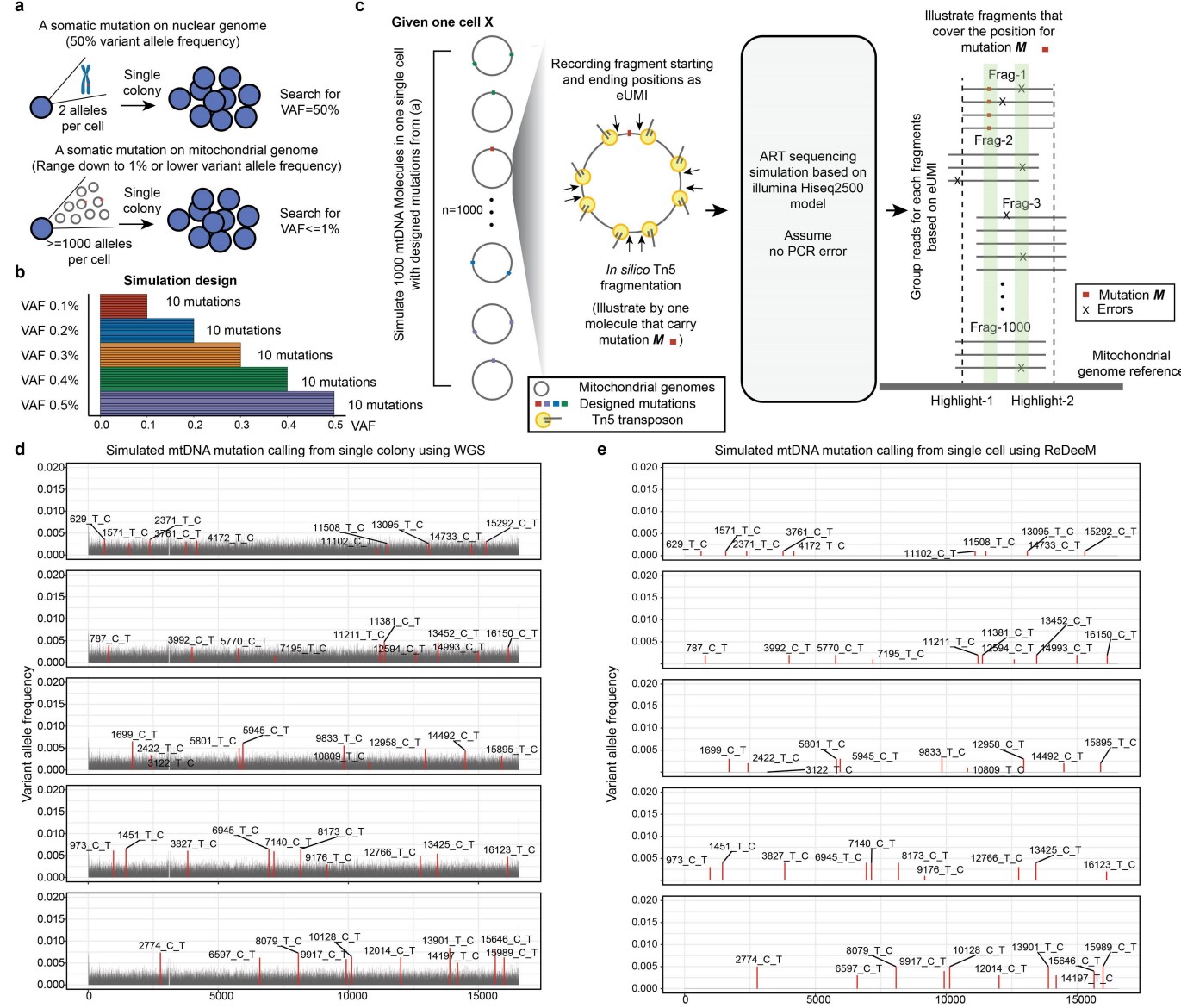

**Extended Data Fig. 2 | eUMI-based error correction via ReDeeM versus conventional mutation detection. a**, The challenge of mtDNA mutation calling using conventional WGS in single colony. **b-e**, Simulation analysis of mtDNA mutation calling using WGS vs ReDeeM. **b**, The design of the mtDNA mutations with low heteroplasmy (0.1% ~ 0.5%) for simulation analysis. 10 mutations are randomly picked for each variant allele frequency (VAF). **c**, Illustration of the simulation analysis process. One single cell with 1000 mtDNA copies is simulated, with the designed mutations from panel **a**. Next, in silico Tn5 fragmentation followed by artificial sequencing is simulated. The resulting simulated data is analyzed by ReDeeM or conventional mutation calling pipeline. The highlight-1 (Real mutation M) and highligh-2 (Error) have the same total frequency which can only be distinguished by ReDeeM. **d-e**, Mutation calling results using conventional WGS in **d** and the eUMI-based ReDeeM pipeline in **e**. Also see Supplementary Notes.

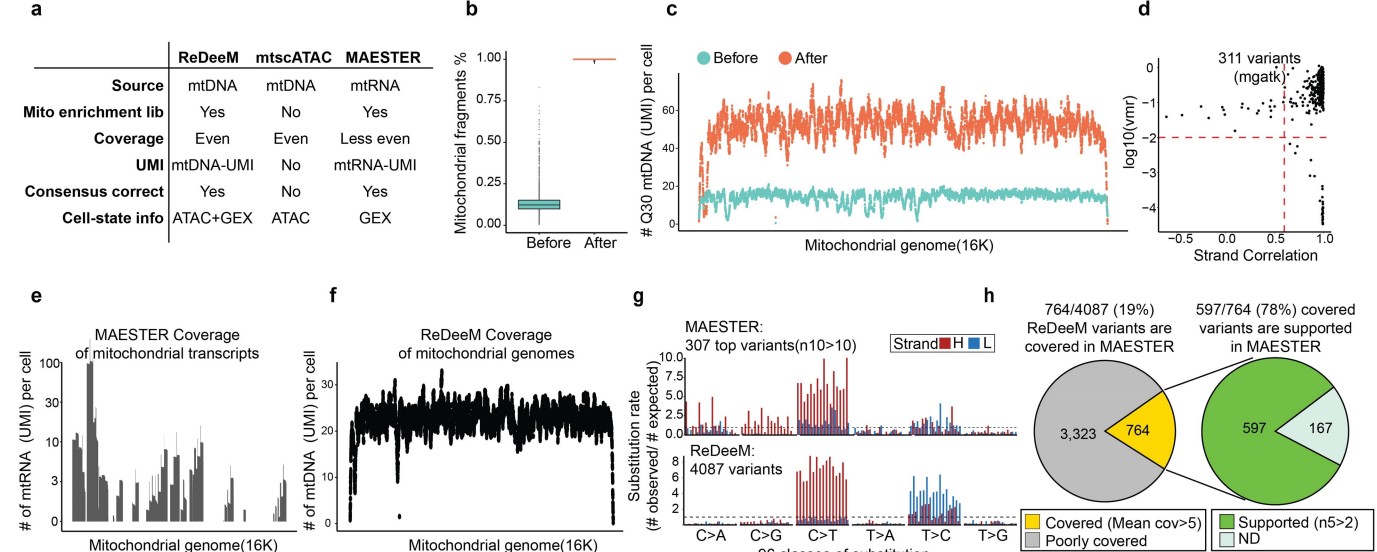

**Extended Data Fig. 3 | Comparative analysis of ReDeeM, mtscATAC, and MAESTER. a**, Comparison of the design and features of ReDeeM, mtscATAC and MAESTER to highlight their similarities and differences. **b-d**, Comparative analysis of before (as mtscATAC or DOGMA-seq) and after mtDNA enrichment (as ReDeeM), which is exemplified by young-1 HSPC dataset. **b**, Percentage of total reads mapped to mitochondrial genome per cell, before and after mtDNA enrichment. n = 14,808 cells. Boxplot displays data from the 25th to 75th percentile, and whiskers extending to the minimum and maximum within 1.5 IQR. **c**, Averaged mitochondrial genome coverage per cell at each position before and after mtDNA enrichment. **d**, Variant calling before enrichment using mgatk. 311 confident variants are identified. VMR: per mutation variance mean ratio. **e-h**, Comparative analysis of ReDeeM and MAESTER, which is exemplified by young-2 BMMC dataset. **e**, Mitochondrial genome coverage by MAESTER. **f**, mitochondrial genome coverage by ReDeeM. **g**, Mutational signatures of 307 top mutations by MAESTER, and 4087 variants by ReDeeM. n10 > 10: variants present in at least 10 cells with a VAF of >10%. **h**, Consistency between ReDeeM and MAESTER. n5 > 2: variants present in at least 5 cells with a VAF of >2%. See Supplementary Notes.

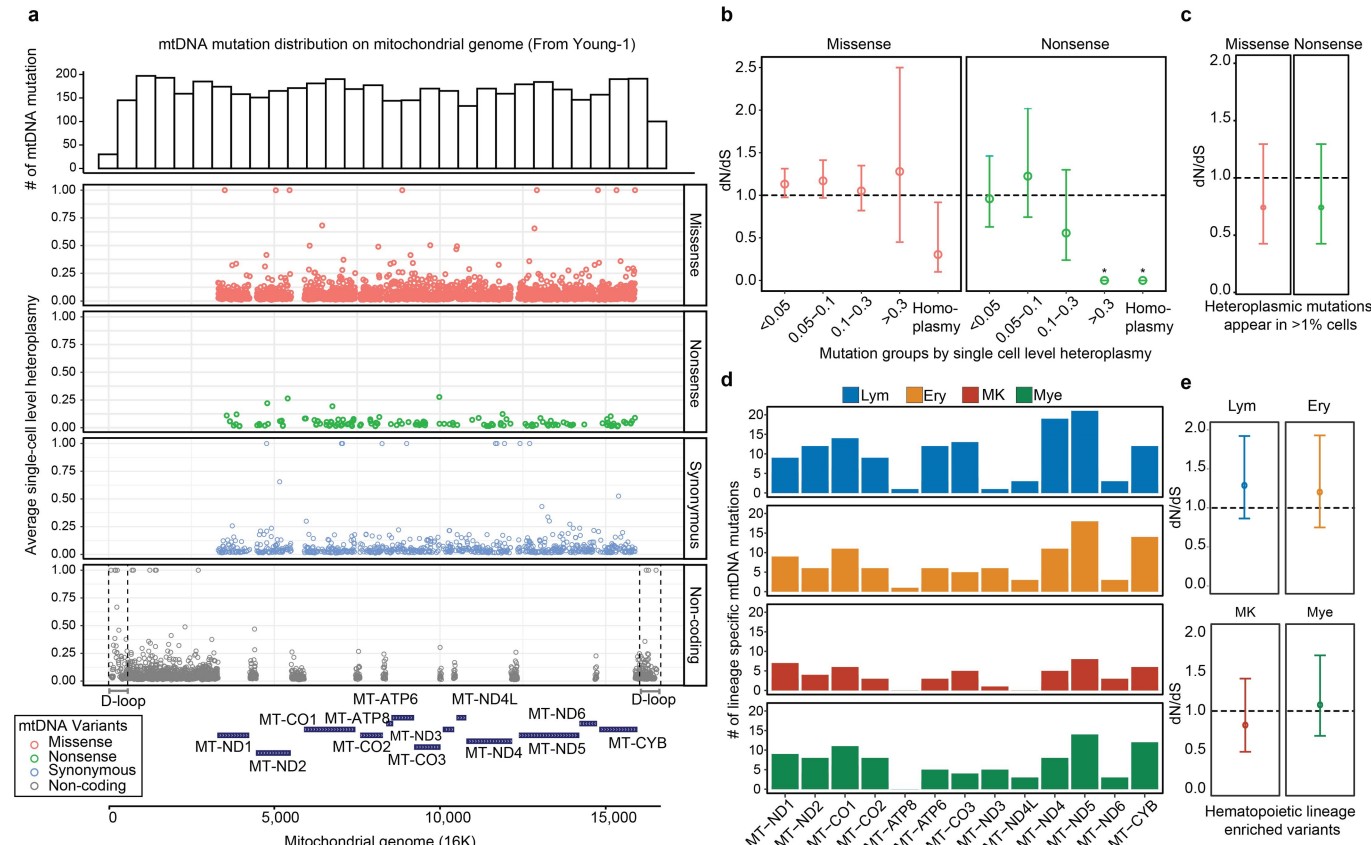

**Extended Data Fig. 4 | Potential functional impacts of mitochondrial mutations. a**, mtDNA mutation distribution on mitochondrial genome. Top panel: histogram that summarizes the mtDNA mutation numbers across mitochondrial genome. Bottom panel: individual mtDNA mutation coordinates and single-cell heteroplasmy level are shown simultaneously, with 4 categories of mutations: missense, nonsense, synonymous and non-coding. **b-c**, Mitochondrial genome-wide dN/dS ratio for missense and nonsense mutations in different mutation groups, based on single-cell heteroplasmy levels (as fraction) in **b** and based on the percentage of cells that carry the mutations in **c**. The bars indicate the 95% confidence interval. n = 4,837 mtDNA mutations. Asterisks indicates dN/dS ratios where the confidence intervals from dndscv were infinitive. **d**, Summarise of the number cell-type restricted mtDNA mutations on each of mitochondrial coding genes. **e**, dN/dS analysis for cell-type restricted mutations. Also see Supplementary Notes. The bars indicate the 95% confidence interval. n = 933 mtDNA mutations.

**Extended Data Fig. 5 | Validation of ReDeeM lineage tracing via dual-tracer experiment. a**, The design of the dual lineage-tracer experiment with a Kras;Trp53(KP)-drive lung adenocarcinoma lineage-tracing mouse model. CRISPR-based and ReDeeM-based lineage information were analyzed for the same cells. 6 Independent tumours were profiled in batch1 (T1-T6), and 4 tumours were profiled in batch2 (T7-T10). **b**, CRISPR indel-based lineage relationships were computed with weighted hamming distance and visualized by multidimensional scaling (MDS). Tumour 1 (T1) is shown as an example with 214 single cells. **c**, Example mtDNA somatic mutations that agree with the CRISPR indel-based MDS map are highlighted. **d**, Schematic of the Agreement of Closeness (AOC). **e**, The phylogenetic trees based on mtDNA mutations are illustrated. The "clonal groups" are indicated as the colored bars. The positive AOC ratio for each clonal group is shown within each colored bar. The individual

AOC scores (middle) and mutation numbers (bottom) are shown for every single cell (each leaf). The p values are computed by 1000 times permutations (one-sided, Supplementary Methods). The whole tree-level metrics of positive AOC ratios are shown for each tumor below the colored bar. **f**, Adjusted Rand Index (ARI) for clonal cluster consistency between CRISPR and ReDeeM across 10 tumors (T1-T10). Various clonal cluster resolutions are tested (presented as each dot). n = 16 resolution pairs for each tumor. Boxplot displays data from the 25th to 75th percentile, and whiskers extending to the minimum and maximum within 1.5 IQR. **g-h**, Illustration of the clonal cluster consistency for T1 (214 single cells) and T9 (410 single cells) on CRISPR-based embedding map using one example cluster resolution. Colors indicate either ReDeeM-based or CRISPR-based cluster identification. Also see Supplementary Methods, Supplementary Figs. 2–3.

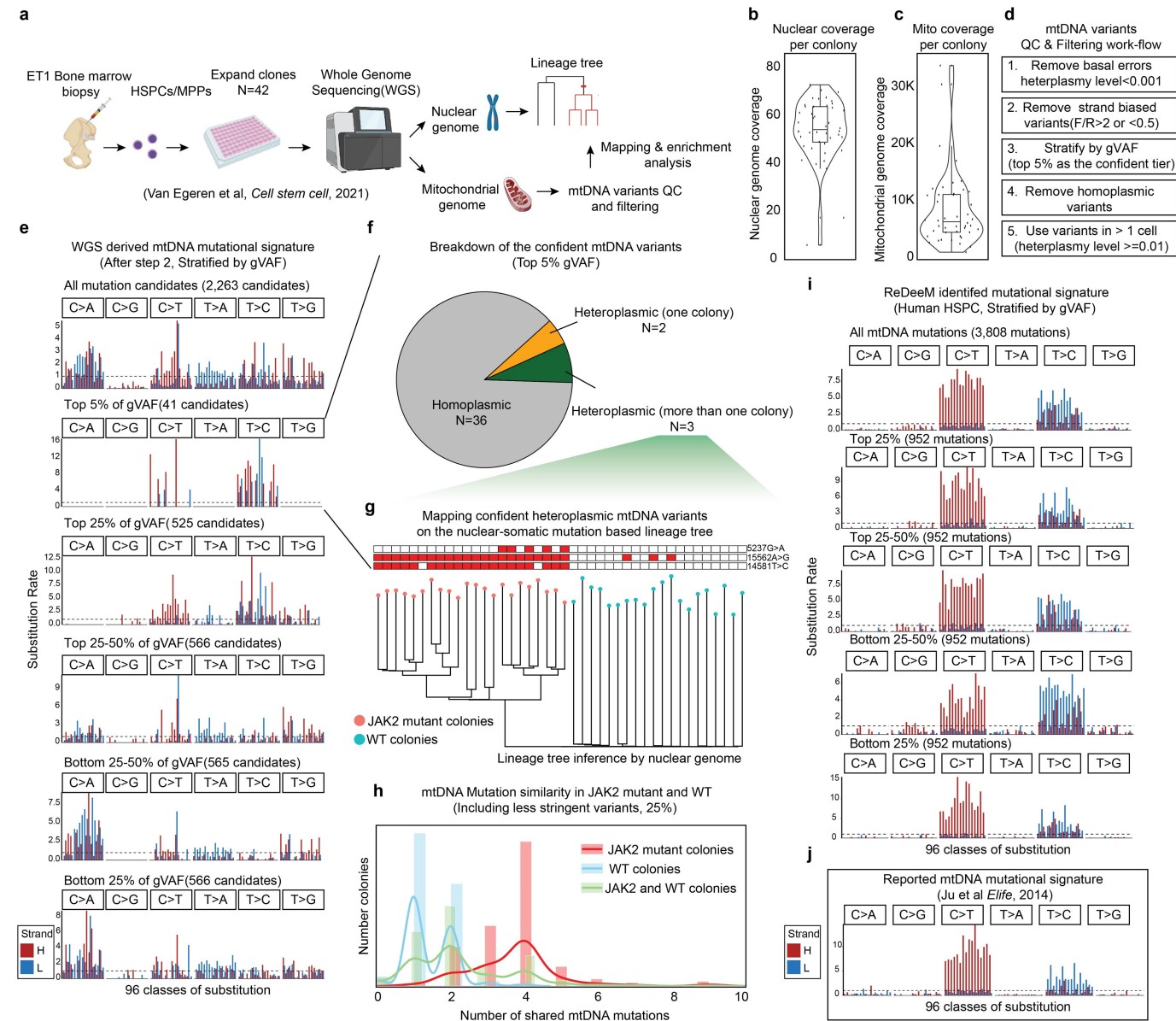

**Extended Data Fig. 6 | Mitochondrial mutation analysis in single colony WGS data. a**, Schematic of the experimental design and reanalysis strategy. **b-c**, Single colony WGS data quality control. Sequencing coverage on nuclear genome in b, and on mitochondrial genomes in c are shown. Each dot represents one colony. n = 42 colonies. Boxplots display data from the 25th to 75th percentile, and whiskers extending to the minimum and maximum within 1.5 IQR. **d**, Mitochondrial mutation calling and quality control pipeline for WGS data. **e**, Mutational signatures in each class of mononucleotide and trinucleotide change by the heavy (H) and light (L) strands. The mtDNA mutations are after step 2 from **d** and stratified by global VAF (gVAF). n = 2,263 all mutation

candidates. **f**, Categories of the top 5% confident mtDNA mutations. The pie chart shows the proportions of homoplasmic mutations (or appear in all colonies), singular heteroplasmic mutations (or appear only in one colony) as well as heteroplasmic mutations (or appear in a subset of colonies). **g**, Direct mapping for confident heteroplasmic mtDNA mutations onto the nuclear genome inferred tree. **h**, mtDNA Mutation similarity (# of shared mutations) within JAK2 mutant clones; within WT clones; or between JAK2 mutant clones and WT clones. **I**, Mutational signatures for mtDNA mutations identified by ReDeeM in Young-1 HSPC dataset, stratified by gVAF. **j**, Expected true mtDNA mutational signature. Also see Supplementary Notes.

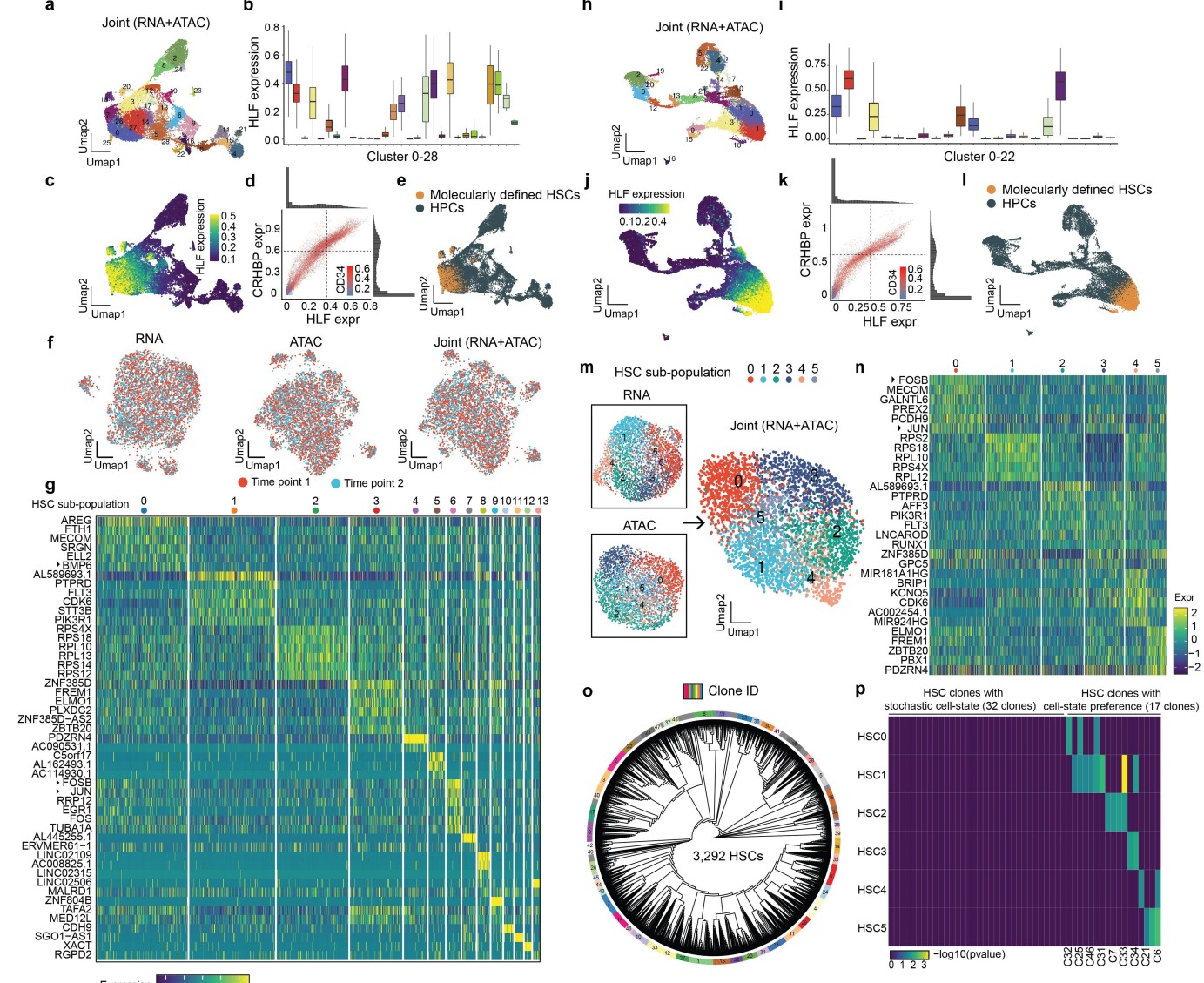

**Extended Data Fig. 7 | Single-cell multiomics analyses of HSC subpopulations.** Panel **a-g** displays data for donor young-1, while **h-p** for donor young-2. **a-e**, Identify hematopoietic stem cells (HSC) based on molecular markers for young-1 **a**, Unsupervised clustering of hematopoietic stem and progenitor cells (CD34+ cells, and CD34⁺CD45RA⁻CD90⁺ cells) based on joint ATAC and RNA modalities. **b**, Expression of HLF mRNA level, a molecular marker of HSCs, in each cluster. n = 14,661 cells. Boxplot displays data from the 25th to 75th percentile, and whiskers extending to the minimum and maximum within 1.5 IQR. **c**, Distribution of HLF expressing levels on wnnUMAP **d**, Define HSCs for CD34⁺CD45RA⁻CD90⁺ cells with HLF^hi, CRHBP^hi, and CD34^hi expression levels, and in HLF high clusters from b. **e**, Highlighting the defined HSCs on wnnUMAP. **f**, HSCs distributions on UMAP from two different time points. **g**, Top examples of HSC subpopulation-specific gene expression profiles, based on RNA modality. **h-l**, same analyses as a-e for donor young-2. n = 23,114 cells. Boxplot displays data from the 25th to 75th percentile, and whiskers extending to the minimum and maximum within 1.5 IQR. **m**, same analysis as in main Fig. 3c for donor young-2. **n**, Same analysis as g, for donor young-2. **o-p**, same analysis as in main Fig. 3e, f for donor young-2. P-values are derived from hypergeometric test.

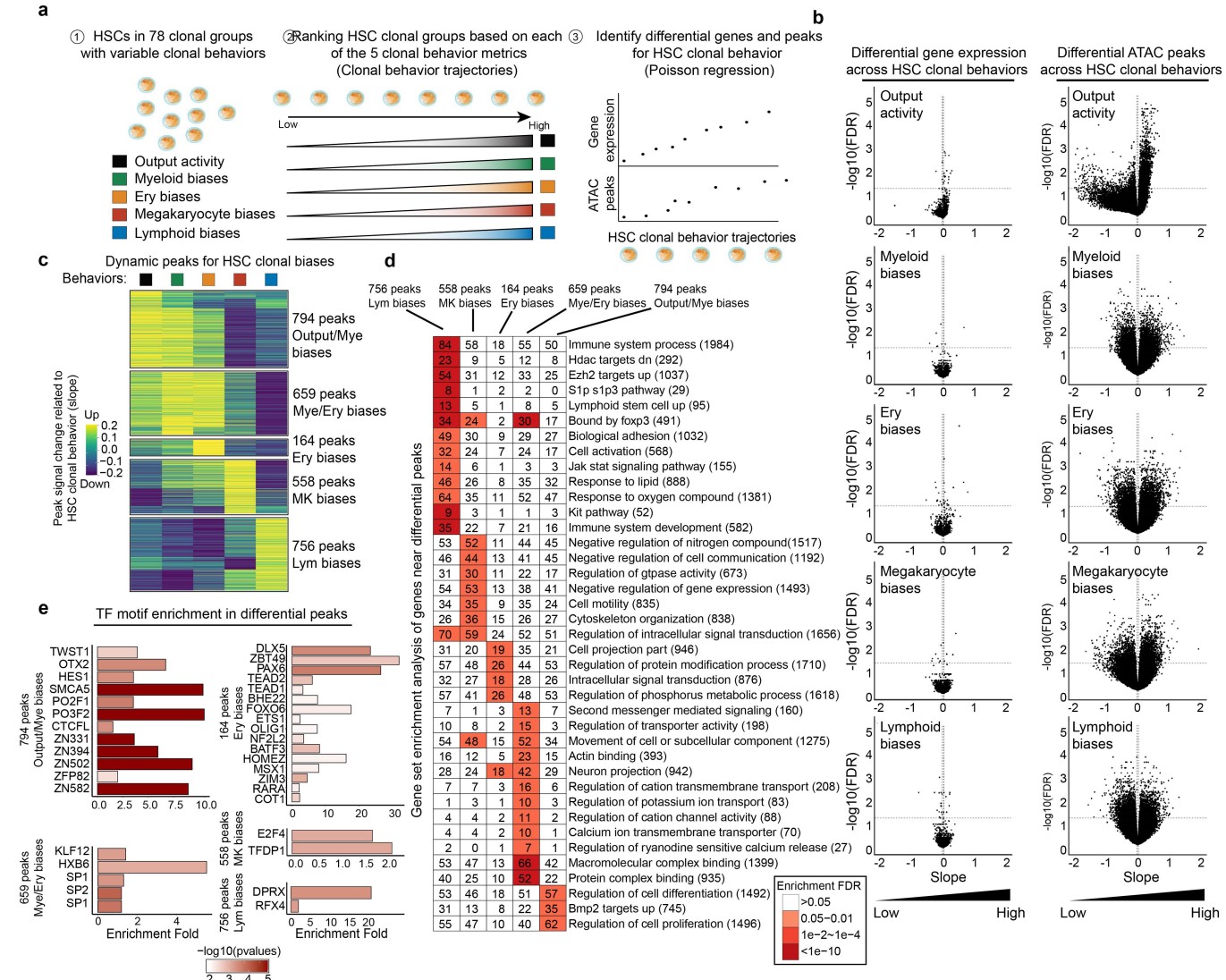

**Extended Data Fig. 8 | Analysis of HSC clonal behavioral trajectories.**
**a**, Schematic of the clonal behavioral trajectory analysis. All 78 HSC clonal groups, comprising 5,393 HSCs, are ranked based on one of the 5 behavioral measurements (see main Fig. 4) to construct the behavioral trajectories. Differential genes/peaks are identified along each of the trajectories based on Poisson regression modeling. **b**, Volcano plots representing the statistical analysis for identifying differential genes/peaks. **c**, 2,931 differential peaks are clustered based on the peak signal changes (slopes) across the 5 behavioral metrics, resulting in 5 different peak categories. **d**, Gene Set Enrichment Analysis for nearby genes of each peak category. **e**, Transcription factor DNA binding motif enrichment analysis for each peak category.

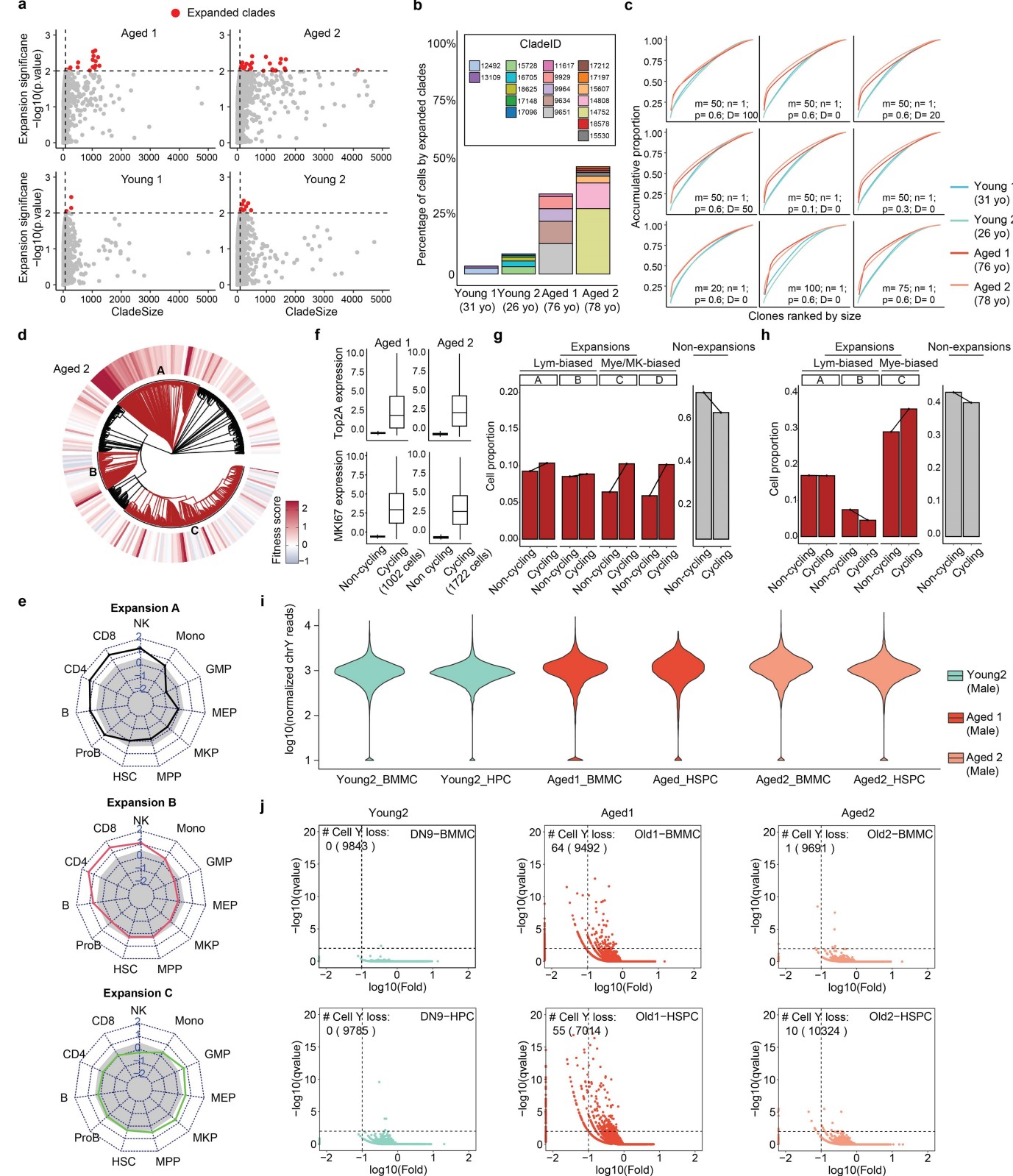

**Extended Data Fig. 9** | See next page for caption.

**Extended Data Fig. 9 | Quantification and validation of clonal structure alteration in aging hematopoiesis. a**, Identification of "expanded clades" in young and aged donors, which are defined as the clades with more than 0.5% of total cell numbers and with expansion significance lower than 0.01. **b**, The percentage of cells that are contributed from expanded clades are summarised for each donor. **c**, Related to Fig. 5d, Measuring the clonal contribution by changing the parameters that affect the definition of "clones". The parameters involved are m (minimum number of cells as a clone, default is 50), n(minimum number of cumulative variants on the branch to cut, default is 1), p (The probability of the variant to be assigned, default is 0.6), and D (Dump small clones with less than D cells). **d**, Related to Fig. 5f. Single-cell fitness analysis in donor aged-2. **e**, Related to Fig. 5g. Cell type contributions for each expanded clade for donor aged-2. **f-h**, Cell-cycle gene expression analysis for expanded versus non-expanded cells. n = 9,519 and 14,715 cells for Aged-1 and Aged-2. Boxplot displays data from the 25th to 75th percentile, and whiskers extending to the minimum and maximum within 1.5 IQR. **i-j**, Identification of cells with loss of chromosome Y. **i**, The normalized number of reads on chromosome Y per cell across different donors. **j**, Binomial test to identify cells that significantly lose chromosome Y with fold-change <0.1 and q-value < 0.001.

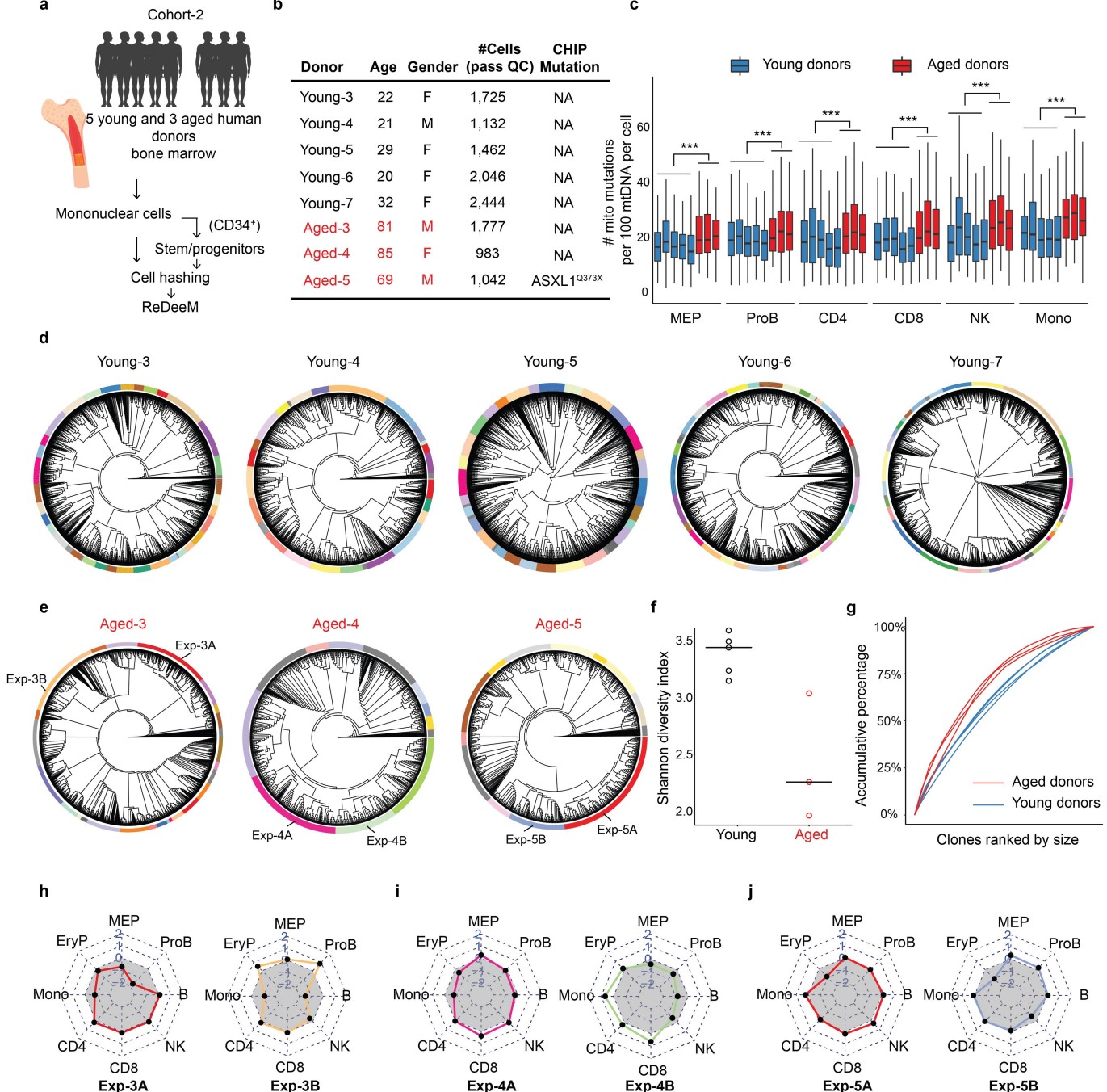

**Extended Data Fig. 10 | ReDeeM phylogenetic analysis for extended young and aged donors. a**, Schematic of the experimental design for the extended donors (cohort-2). Bone marrow samples from 5 additional young and 3 additional aged donors were collected. Isolated bone marrow mononuclear cells and the enriched CD34+ cells were hashed and pooled for ReDeeM profiling and lineage tracing. **b**, The extended donor information. **c**, mtDNA mutation burden between extended young and aged donors across different cell types. n = 9,709 cells from extended young donors, n = 3,802 cells from extended aged donors. Boxplot displays data from the 25th to 75th percentile, and whiskers extending to the minimum and maximum within 1.5 IQR. ***indicates p-value < 2.2*10-16, derived from one-sided Wilcoxon rank sum test (**d-e**) Phylogenetic trees of the extended donors from **d**, young, and **e**, aged donors. Clonal groups are indicated by different colors on the outer rings. **f**, Shannon diversity index of the clonal composition, between extended young and aged donors. n = 5 and 3 donors **g**, Contribution of each clone to the total population in extended young and aged donors. (**h-j**) Cell type contributions in each expanded clone, corresponding to the highlighted clones in panel **e**. The grey area indicates the expected balanced cell type distribution.

# Reporting Summary

## Statistics

For all statistical analyses, confirm that the following items are present in the figure legend, table legend, main text, or Methods section.

| n/a | Confirmed | |
|---|---|---|
| ☐ | ☒ | The exact sample size (*n*) for each experimental group/condition, given as a discrete number and unit of measurement |
| ☐ | ☒ | A statement on whether measurements were taken from distinct samples or whether the same sample was measured repeatedly |
| ☐ | ☒ | The statistical test(s) used AND whether they are one- or two-sided *Only common tests should be described solely by name; describe more complex techniques in the Methods section.* |
| ☐ | ☒ | A description of all covariates tested |
| ☐ | ☒ | A description of any assumptions or corrections, such as tests of normality and adjustment for multiple comparisons |
| ☐ | ☒ | A full description of the statistical parameters including central tendency (e.g. means) or other basic estimates (e.g. regression coefficient) AND variation (e.g. standard deviation) or associated estimates of uncertainty (e.g. confidence intervals) |
| ☐ | ☒ | For null hypothesis testing, the test statistic (e.g. *F*, *t*, *r*) with confidence intervals, effect sizes, degrees of freedom and *P* value noted *Give P values as exact values whenever suitable.* |
| ☒ | ☐ | For Bayesian analysis, information on the choice of priors and Markov chain Monte Carlo settings |
| ☒ | ☐ | For hierarchical and complex designs, identification of the appropriate level for tests and full reporting of outcomes |
| ☐ | ☒ | Estimates of effect sizes (e.g. Cohen's *d*, Pearson's *r*), indicating how they were calculated |

*Our web collection on statistics for biologists contains articles on many of the points above.*

## Software and code

Policy information about availability of computer code

| | |
|---|---|
| Data collection | For cell sorting and gating, BD FACSDiva software was used |
| Data analysis | cellranger-arc-2.0.0 was used for single cell RNA+ATAC preprocessing.<br>Amulet v1.1 (https://github.com/UcarLab/AMULET) was used for doublet removal<br>ART-MountRainier-2016-06-05 was used for next-gen sequencing reads simulation<br>Cassiopeia 2.0.0(https://github.com/YosefLab/Cassiopeia) was used for CRISPR lineage tracing data<br>redeemV v1.0.0 (https://github.com/chenweng1991/redeemV) was used for ReDeeM data preprocessing<br>redeemR v1.0.0 (https://github.com/chenweng1991/redeemR) was used for downstream phylogenetic and integrative analysis<br>ScEasyMode 1.0.1 was used for CellHashing demultiplexing<br>Seurat v4.3.0 and Signac v1.5.0 was used for single cell multimodal analysis<br>SCAVENGE(https://github.com/sankaranlab/SCAVENGE) was used for HSC progeny analysis<br>jungle(https://github.com/felixhorns/jungle) was used for phylogenetic fitness analysis<br>FIMO from meme-5.4.1 was used for TF motif scanning |

For manuscripts utilizing custom algorithms or software that are central to the research but not yet described in published literature, software must be made available to editors and reviewers. We strongly encourage code deposition in a community repository (e.g. GitHub). See the Nature Portfolio guidelines for submitting code & software for further information.

## Data

Policy information about availability of data

All manuscripts must include a data availability statement. This statement should provide the following information, where applicable:

- Accession codes, unique identifiers, or web links for publicly available datasets
- A description of any restrictions on data availability
- For clinical datasets or third party data, please ensure that the statement adheres to our policy

All data generated in the manuscript have been deposited in GEO (GSE219015). The processed Seurat objects are available on figshare: https://doi.org/10.6084/m9.figshare.23290004. The processed mutation calling files on figshare: https://doi.org/10.6084/m9.figshare.24418966.v1. Single colony WGS data are from dbGAP:phs002308.v1.p1. TF motif database JASPAR2020 (https://jaspar2020.genereg.net/) was used with ChromVar. HOCOMOCOv11 (https://hocomoco11.autosome.org/downloads_v11) Human TF database was used for FIMO analysis.

## Human research participants

Policy information about studies involving human research participants and Sex and Gender in Research.

| | |
|---|---|
| Reporting on sex and gender | We have used the term sex or gender carefully throughout the manuscript. Findings are applied generally. Genders were considered in study design. Gender was determined based on self-report. Informed consent was obtained under a sample banking protocol that was approved by the IRB of Mass General Brigham and Boston Children's Hospital |
| Population characteristics | Age is an important characteristic. We have collected 7 young (range from 20 to 32 yo) and 4 aged donors (range from 69 to 85 yo) and performed systematic comparison across age groups |
| Recruitment | Participants were recruited with self report. Donors with reported conditions or diseases were excluded, but specific screens were performed. These donors were considered as in general healthy donors |
| Ethics oversight | The Fresh bone marrow samples from healthy young donors were aspirated with informed consent under a sample banking protocol that was approved by Institutional Review Board (IRB) of Boston Children's Hospital. The sternum bone marrow from aged donors was collected following sternotomy for cardiac surgery with informed consent under a sample banking protocol that was approved by the IRB of Mass General Brigham. |

Note that full information on the approval of the study protocol must also be provided in the manuscript.

# Field-specific reporting

Please select the one below that is the best fit for your research. If you are not sure, read the appropriate sections before making your selection.

☒ Life sciences          ☐ Behavioural & social sciences          ☐ Ecological, evolutionary & environmental sciences

For a reference copy of the document with all sections, see nature.com/documents/nr-reporting-summary-flat.pdf

# Life sciences study design

All studies must disclose on these points even when the disclosure is negative.

| | |
|---|---|
| Sample size | No sample size calculations were performed a priori. Analyses involved thousands of cells per comparison, providing a robust sample size in-line with similar high-throughput scRNA-seq comparisons and technologies. To well cover different hematopoietic cell populations and clones. We sampled large sample sizes for main donors: 54,221 cells for young-1, 34,721 cells for young-2; 20,496 cells for aged-1; 25,717 dells for aged-2 |
| Data exclusions | The data preprocessing for the joint single-cell RNA and ATAC data was performed using 10X Genomics data preprocessing software Cell-ranger-arc. The basic quality control analysis was as follows: RNA UMI: 1,000~25,000 transcripts per cell, unique ATAC fragment: 1,000 ~ 70,000, Fragment on peak minimum percentage: 10%, minimum mtDNA copies per position per cell: 10X. Finally, the possible doublets are removed by Amulet (using default parameter). |
| Replication | All findings discussed in the manuscript are reproducible in two independent sampling from the same individuals as well as two independent individuals. |
| Randomization | There were no variables or interventions to randomize in this study. |
| Blinding | Blinding is not relevant to our study. Analyses were performed in an exploratory manner where blinding is not possible. |

# Reporting for specific materials, systems and methods

We require information from authors about some types of materials, experimental systems and methods used in many studies. Here, indicate whether each material, system or method listed is relevant to your study. If you are not sure if a list item applies to your research, read the appropriate section before selecting a response.

## Materials & experimental systems

| n/a | Involved in the study |
|-----|----------------------|
| ☐ | ☒ Antibodies |
| ☐ | ☒ Eukaryotic cell lines |
| ☒ | ☐ Palaeontology and archaeology |
| ☒ | ☐ Animals and other organisms |
| ☒ | ☐ Clinical data |
| ☒ | ☐ Dual use research of concern |

## Methods

| n/a | Involved in the study |
|-----|----------------------|
| ☒ | ☐ ChIP-seq |
| ☒ | ☐ Flow cytometry |
| ☒ | ☐ MRI-based neuroimaging |

## Antibodies

| | |
|---|---|
| Antibodies used | To further enrich the hematopoietic stem cells (HSCs), an aliquot of the enriched CD34+ cells were stained by one of the following two antibody panels. 1) CD34 PerCP-Cy5.5 (Catalog #347222); CD45RA Alexa Fluor 488 (Catalog #304114); CD90 PE-Cy7 (Catalog #561558); and  DAPI (Catalog #D1306) as viability dye. Or 2) CD34 BV421 (Catalog #562577); CD45RA-APC-H7 (Catalog #560674); CD90 PE-Cy7 (Catalog #561558), and 7-AAD as viability dye(Catalog #559925). The cells were further sorted using BD FACSAriaTM for CD34+CD45RA-CD90+ to enrich HSCs.  All antibodies are from BD biosciences. |
| Validation | Each lot of these antibodies is quality control tested by immunofluorescent staining with flow cytometric analysis. The validations are routinely done in the Sankaran lab by staining human hematopoietic stem and progenitor cells. |

## Eukaryotic cell lines

Policy information about cell lines and Sex and Gender in Research

| | |
|---|---|
| Cell line source(s) | K-562 cell line from ATCC |
| Authentication | None of the cell line was authenticated |
| Mycoplasma contamination | Cell lines are routinely tested for mycoplasma contamination. Results were consistently negative. |
| Commonly misidentified lines (See ICLAC register) | No commonly misidentified lines were used as part of this study. |

