## [Peer Review File · Nature]

Manuscript Title: Deciphering cell states and genealogies of human hematopoiesis

Reviewer Comments & Author Rebuttals

Reviewer Reports on the Initial Version:

Referees' comments:

Referee #1 (Remarks to the Author):

In this manuscript, Weng et al. have studied the clonal architecture of human hematopoiesis using single-cell multiomics and mtDNA-lineage tracing. For the first time, the authors have applied these methods to study the bone marrow from aged individuals, resulting in a very exciting and rich new dataset. The senior authors are experts in the development of these single-cell tracing methods, having pioneered much of the early technology that is applied throughout this new manuscript (e.g. PMID: 32788668). As expected from such a team, this is a superbly written manuscript, with clear figures that are easy to understand and a very thorough methods section that should enable data reproducibility. I expect this manuscript to be a tour de force, provided some concerns are addressed.

The paper begins by describing a significant methodological improvement of single-cell mtDNA sequencing and variant calling, ReDeeM, which uses probes to capture the mt transcriptome, and then UMIs to correct for seq errors. Whereas previous methods had to rely on a few shared mtDNA variants to connect cells in clones, ReDeeM now identifies many more shared variants per cell, which increases the confidence of cell phylogenies. The authors even validate ReDeeM with a different method of phylogenetic mapping, which uses CRISPR-based evolvable barcodes in mice. With this tool, the authors then describe a series of ReDeeM profiling experiments on 2 young and 2 aged bone marrow aspirate donors. The authors profile about 60,000 cells in total, across all samples, including, longitudinal samples to track how stem cell clones behave dynamically in situ. Previous attempts at this had been limited by the low number of connecting mtDNA variants, and thus a much poorer capacity to connect stem cells and their progeny. ReDeeM appears to excel at this, although some amount of data-collapsing (i.e. pooling clones into "clades" and using network propagation) appears to be required to be able to reach meaningful results. Perhaps the most exciting result from this part is that the authors obtain a hematopoietic fate decision map of human hematopoiesis in situ, built just from the somatic mtDNA variant data (Fig 2f), something that had been impossible using previous methods for somatic variant calling.

Next, the paper turns to analyze the epigenetic states and fate biases of different HSCs. Intriguingly, the authors identify a striking variety of cell-state clusters within human HSCs (which, surprisingly, is also one of the best available multiome datasets on human hematopoiesis), and then group the 5400 HSCs into 78 "clonal groups" based on shared mtDNA variants to study three major properties: i) their cell-state heritability, ii) their differentiation output, iii) their fate bias. For the first time in native human hematopoiesis, the authors show they are able to identify "clonal groups" with state heritability/stability, differences in output, and differences in fate biases, although the effect sizes of these differences appear to be small in all cases, likely due to methodological reasons.

Finally, the authors profile two aged individuals. Focusing their analyses on the clonal composition, they confirm that aged hematopoiesis is more oligoclonal, with specific clades of HSCs being expanded (increased "fitness"), some of which could be explained by the loss of the Y chromosome (LOY).

In summary, this manuscript addresses the technological challenge of studying clones in human native hematopoiesis with cell-state information, and it uses this clonal information to help define differences in HSC function and state with better confidence and thoroughness than previous mtDNA-tracing studies. I believe this is a relevant and timely topic, with technological innovations such as these being able to drive the field, but I have a series of issues that would first need

addressing. These are mostly:

- some technical questions on the method and its validation
- the limitations of studying "clonal groups" and connecting these with progeny through network-propagation strategies
- some additional analyses/comparisons of young and aged hematopoiesis

These points (and other minor ones) are detailed below.

The mtDNA matrix is binarized "to minimize coverage bias", but this could mean that cells with widely different heteroplasmy levels for a certain mutation end up considered as part of the same clone. Isn't ReDeeM supposed to be able to estimate and avoid this to a certain extent, based on eUMI counts? Is binarization really essential?

The agreement of closeness measure is very interesting way of comparing the two methods (mtDNA- and CRISPR-based lineage tracing), but I still cannot figure out how significant this result is. In other words, how many times is the ReDeeM fate-mapping method going to misclassify (false positive) two cells that are actually not related? And how relevant is this when clone-aggregation (clades) and network-propagation strategies are used for measuring HSC output and fate bias?

In the approach of selecting lineage-informative mtDNA mutations for building the human hematopoietic lineage tree, how was it determined if a barcode was randomly distributed? How many total cells and "barcodes" were used for this analysis? Also, are these "lineage-informative" mtDNA mutations ever found in HSCs? If so, how many? I suspect they will be quite rare, and thus it will be quite difficult to find them frequently in HSCs.

"Clonal groups" seem to be a rather arbitrarily designed method for grouping HSCs. For instance, why do they have to contain at least 50 single cells? Why that number? Also, why enable using just one shared (binarized) confidently assigned mutation to call a group? Shouldn't authors take advantage of ReDeeM and increase confidence with more than 1 variant? As a note, what happens to the "agreement of closeness" if one chooses a similar grouping approach when using CRISPR-based benchmarking? What is the agreement of closeness of "clonal groups" in that case?

Based on the fact that "clonal groupings" are studied, instead of the fates of individual stem cells, the manuscript is sometimes clear on this, and sometimes not, assuming that the data are revealing the fate decisions (and fate heterogeneity) of individual HSCs. Rather, the "clonal groupings" used in the manuscript are likely to have an origin at some point during early development (based on their mtDNA burden, their size, and the BM aspirate sampling rate, I think it may even be possible to estimate when). I think it may be important for readers to fully grasp this as part of the main text.

The SCAVENGE-L method to use network propagation to connect HSCs with their progeny is very interesting but I worry about its classification error. The authors "validated" the method with the HSC clonal groupings, but these were created with more stringent rules. My understanding is that many (or even most) of the connections the authors make between HSCs and committed cells are being made through very large distances. This is not just due to the sparsity and data coverage, but because of the extreme cellular undersampling that is taking place when performing a BM aspirate. I fully understand the limitations in terms of clinical sampling and the will of the authors to extract the most information from their analysis. But perhaps it would be important to add this in a section mentioning the limitations of this study. It is definitely possible that the relatively small effect sizes that are being detected (Fig 4b and 4e) are partly driven by the chosen approach.

To my understanding, this is the first time someone connects HSC (clonal group) fates and states in native human hematopoiesis. However, the authors do not try to make any connections between the data in Fig 3f and Fig 4b/e. Are there any transcriptional or chromatin-accessibility

differences between high and low-output HSC clones (clonal groups)? Are there any differences in myeloid- versus mega- versus lymphoid-biased HSC clones (clonal groups)?

While an extremely unique dataset, none of the findings in the aged hematopoiesis experiments seem to be very novel. Of course, it is good to validate previous findings with a new method, but I feel this dataset should be analyzed a bit more. Are there any differences in output in the aged HSC clonal groups (compared to the young)? What is the cell-state heterogeneity in the aged HSC clonal groups? Are there any transcriptional or chromatin-accessibility differences that characterize HSCs of different fitness groups in the aged setting?

In the methods, "sequencing" section, it is mentioned that 50-60,000 reads (per cell?) are used for mtDNA libraries. Is this coverage really necessary? Perhaps I am not understanding Ext Fig 1a.

Figure 3d is called after Figure 3e.

ExtFigure 6g is called after ExtFig 6h-i.

In the methods section, "progines" instead of progenies.

- Alejo E. Rodriguez-Fraticelli

Referee #2 (Remarks to the Author):

The manuscript by Weng and colleagues presents a method, ReDeeM, for deep mitochondrial genotyping from single-cell mutli-ome data and thereby identify clones and 'phylogenies' of cells. If the proposed method works as claimed, it would constitute a major step forward and be of maximal interest, as it permits lineage tracing in human.

The authors then go on to apply their method to study clonal output and relationships in hematopoiesis; their biological findings in this context are original and of interest, but do not fundamentally add beyond what has been published in the last years both from the mouse and human system (e.g. work by Fernando Camargo, Hans-Reimer Rodewald and Peter Campbell). Therefore, in my opinion, the manuscript should be primarily evaluated on the claim that ReDeeM can reliably identify clones (sets of genetically related cells) and phylogenies.

I do not doubt that mitochondrial genomes occasionally mutate somatically, and that such somatic mutations can then be used as lineage tracers. I am not convinced, however that this occurs at a sufficient rate to assign more than a few cells to clones and even assemble phylogenies.

- The frequency of mutations reported here conflicts with what is known about the frequency of somatic mitochondrial mutations in clonal hematological malignancies (see my first point)
- Controls included here are not convincing (see my 2nd and 3rd point)
- There are reasonable alternative biological explanations for the observations (see my 4th point)
- Hence the authors would need to provide strong further experimental validation of the clones and phylogenies they identify (see my 5th and most important point)

Besides these conceptual points, I have several technical points listed below, in particular regarding statistical analysis.

Conceptual point 1

Leukemia, in particular AML, involves a rapid and massive clonal expansion of a single HSC. Therefore, if really every HSC has (somewhat) specific mitochondrial mutations, in every leukemia patient there should be somatic mitochondrial mutations when comparing to germline. An analysis of PCAWG data (Yuan et al., Nature Genetics 2020, DOI: 10.1038/s41588-019-0557-x) suggests that this is only the case in ~30% of AML cases and <20% of Myeloproliferative Neoplasms (Figure 1f), and this includes mitochondrial mutations that occur after the leukemia initiating genetic

event. Accordingly, ReDeeM should only identify mitochondrial mutations in maximally 20% of cells, rather less. Here, several mutations are reported in most cells. This raises doubts on the validity of the results. How would the authors explain this difference?

Conceptual point 2

The most important control currently included is Figure EV3. Here, a genetically encoded CRISPR based lineage tracing system was used (in mouse) and the resulting clones are compared to the clones identified by the mitochondrial lineage tracing. The experimental design is elegant but the analysis is not convincing for several reasons:

- The examples from panel c seem cherry picked, all mitochondrial mutations should be shown (e.g. as a supplementary file if space is limiting). Even with the selected mutations, it appears that they often occur across different clones identified from CRISPR, does this not show that these mutations are not specific to clones? See also my 4th point.
- The CRISPR clonal identity (as in panel b) should be shown below the ReDeeM based trees in panel e to allow a real comparison.
- I don't understand the description of "Agreement of closeness" (panel d,e, Methods). Can the authors use established statistical tools to compare between CRISPR-based clones and ReDeeM-based clones, such as the Rand index?

Conceptual point 3

Another observation from the manuscript that raises doubts on the central claim is figure 5f. Here ReDeeM was applied to an elderly individual with loss of chromosome Y. It is very surprising to see that loss of Y is only weakly enriched in some "clades" identified by ReDeeM and by and large stochastically distributed across the phylogeny. Are these enrichment p values accounted for multiple hypothesis testing (several clades were tested for enrichment, so this is necessary)? LoY is a genetic mark that should be inherited in clades. In the Mitchell et al Nature 2022 paper a similar individual is analyzed with whole genome sequencing of colonies. Loss of Y, although it occurs several times, is there consistently observed within more expanded clades (see their figure 3c), as expected. The authors should validate their unexpected result, e.g. by WGS of HSC-derived colonies from the same individual. See also my 5th point.

Conceptual point 4

There could be an alternative explanation for the observations presented in this study: Heteroplasmy fluctuates over time. Hence, mitochondrial variants present in germ line might just sometimes be present at higher heteroplasmy (and be captured by ReDeeM, which captures approx. 50 of "100s to 1000s" mitochondrial genome copies) and sometimes at lower heteroplasmy (and mostly drop out in ReDeeM). Over ageing, some germline variants will disappear by random drift.

Can the authors prove that the mutations arise de novo and do not exist in germ line? Deep sequencing of (bulk) mitochondrial genomes from other tissues of the same donor (or maybe even T cells) could be used to investigate this.

If they exist in germ line, can the authors exclude that there is no selection for variants in certain lineages? Work by Leif Ludwig suggests that this is sometimes the case (10.1101/2022.11.20.517242) and we have no clue of the extent to which this happens (The metabolic requirements of erythroblasts are quite different from those of neutrophils...). This would fundamentally limit the ability of such variants to be used for lineage tracing.

Conceptual point 5

The authors should add more validations of their phylogenies and sanity checks.

- WGS from HSC derived colonies plus ReDeeM for the same individual would provide a gold standard validation of ReDeeM. The same phylogenies should be obtained with both methods.
- The publicly available sequencing data from Lee-Six et al., Mitchell et al. etc. (WGS of HSC-derived colonies) should contain deeply covered mitochondrial genomes. Are data from these mitochondrial genomes sufficient to reproduce the clonal hierarchies that these manuscripts report based on nuclear mutations? Resulting phylogenies would need to be qualitatively and

quantitatively compared. This has the advantage that technical dropout of single cell methods will not be a problem, so it can provide a basic proof of concept for the idea that phylogenies can be obtained from mitochondrial mutations only.

Technical point 1

Assuming de novo, spontaneous somatic mutation, Is the way that phylogenies are computed ideal? There is very large technical noise here, since only a small percentage of all "100s of 1000s" mitochondrial DNA molecules per cell are covered, and heteroplasmies of the observed mutations are low. So evidently there is plenty of dropout (false negatives). In such a noisy setting, simple distance metrics for clustering are known to produce incorrect phylogenies/clonal hierarchies; the computational methods by the Beerenwinkel group (SCITE etc) and PhISCS by Salem Malikic address this issue and infer correct phylogenies even in a noisy setting. PhISCS is faster/copies with a larger cell and mutation numbers. Such methods should preferably be used.

Technical point 2

It would be good to know more precisely how many mtDNA copies there are per cell, so as to compute the false negative rate of this method and its sensitivity.

Technical point 3

Related to figure 1e: Does the heteroplasmy of mitochondrial mutations change/increase during differentiation? This could indicate a selection of variants (see also main point 4).

Technical point 4

In figure 2f, I do not understand the color scale. They could show a percentage (i.e. out of k clonal neighbors of T cells, y% are B cells).

Technical point 5

In figure 3g, are p values corrected for multiple hypothesis testing? Take note that a total of $13 * (51+27) = 1014$ statistical tests were performed. It is not enough to correct all tests for a single HSC subpopulation or clonal group.

Related, in the context of figure 4: "we identified 47 (60%) HSC clonal groups with significant lineage enrichment (binomial test $pt1 < 0.05$ & $pt2 < 0.05$), while 31 (40%) HSC clones showed low or no lineage bias. Is this adjusted for multiple hypothesis testing?

Technical point 6

In the clonal analysis of lineage bias (Figure 4e) and HSC subsets (figure 3g) effects are rather weak, and typically biological measurements suffer from some form of overdispersion. Are the observed lineage biases stronger than what one would observe in a sample without lineage biases, measured with the same method? I would suggest the following control: They could take a BM sample, split into 4 fractions, stain each fraction with a different hashing antibody, pool at some ratio (e.g. 5:10:25:60), and perform ReDeeM. Then they could check if different clones are "enriched" for different hashing antibodies. If that's the case, the statistical test in figure 4e needs to be adjusted to account for technical overdispersion (e.g. beta-binomial test).

Technical point 7

Data availability: It would be good to provide the data and annotations (clones, cell states, dimensionality reductions) also as a ready-to-use Seurat (or scanPy) object on a platform such as figshare.

Referee #3 (Remarks to the Author):

In this manuscript, Weng et al report the development of a single-cell multi-omics methodology for simultaneous RNA-seq/ATAC-seq/clonal tracking(mtDNA-based) and use it to investigate human hematopoiesis in two young and two aged donors. The major methodological advance derives from

improved recovery of mtDNA sequences by improving yield and applying a target-enrichment strategy.

Using this methodology, the authors derive a detailed snapshot of hematopoiesis including clonal output, lineage contribution, hematopoietic stem cell (HSC) numbers and lineage bias. By analyzing the same (young) person twice, 4 months apart, they provide insights into the stability of HSC states and lineage output/bias. By comparing two young (26 and 31 yrs) with two aged (76 and 78 yrs), they capture the oligoclonality and presence of large clones in the latter.

Whilst some of these findings are not novel in isolation, the ability to capture them simultaneously represents an important advance and I expect that the methodology will be used to study hematopoiesis, in many different contexts, in a detail that has not been possible before.

Below are some suggestions aiming to further improve the manuscript:

1. The heteroplasmic nature of mitochondria and their inheritance introduces caveats in the use of mtDNA mutations as clonal markers (for example mtDNA mutations can be lost by stochastic drift in heteroplasmy). The authors should discuss this briefly, as readers will not necessarily be fully aware of the implications. Importantly, the demonstration that ReDeem agrees well with CRISPR-based lineage tracing provides important reassurance in this regard. However, the concordance between ReDeem and CRISPR was not complete. Overall, I am persuaded that the methodology is robust enough to support the authors' findings/conclusions. However, it would add to my confidence if the phylogenetic relationships are seen to be retained after pooling multiple donor samples. This, for example, would provide confidence that the technical and analytical approaches are resistant to experimental conditions etc. So, pooling donors and re-running algorithms would be a better test of the robustness of the methodology.

2. The use of BM (rather than blood) cells enriches for progenitor cell types, but may give a biased view of phylogenetic relationships given the physical proximity of sampled cells. Whilst this was not observed in colony-based studies, but these only analyzed very few samples and relatively small numbers of cells/colonies. Analysis of blood-derived HSPCs from at least one of the four donors would be very helpful and would give additional insights into the relationship between BM-resident and circulating HSCs/HSPCs.

3. The aged donors display fairly large expanded clones, even in the absence of clonal hematopoiesis (CH) driver mutations. Previous publications (e.g. Mitchell et al, Nature 2022) have demonstrated this to be common in older individuals. However, it would be good to know that this is what the current manuscript/methodology is also showing (i.e. apparently "driverless" expansions). So, it would help to determine whether CH driver mutations can be identified in bulk DNA from the aged donors using conventional targeted sequencing (the clade sizes suggest that any such mutations should be identifiable in bulk DNA).

Referee #4 (Remarks to the Author):

In this work, Weng and colleagues improved an endogenous lineage tracing system by supporting mitochondrial variant enrichment and detection in a single-cell multiomics platform and applied it to study human hematopoiesis aiming to link cell states and lineages. Most of the analyses are technically solid and the paper is well written with results clearly presented.

A few key discoveries of this work include 1) HSC mtDNA clones showing functional differences and differentiation bias, where the fate bias is relatively small but can be sustainable for months; 2) HSC clonal diversity decreases with age, where positive expansion is observed with higher fitness. These biological findings are of potential interest. As not an expert in hematopoiesis, the reviewer will focus on technical/computational aspects.

1. As far as I understand, this paper, for the first time, introduced mtDNA enrichment in the single-cell multi-omics platform. This is a technical innovation and has good application potential. However, mtDNA enrichment has been recently achieved in both ATAC level (mtscATAC, ref 29) and RNA level (MAESTER, ref 32) by some of the authors and colleagues. Therefore, it would be key to have a comparison to these technologies, in terms of mtDNA enrichment and mutation detection.

2. Related to the previous point, the multi-omics readout is a technical advance, which allows the authors to detect the variability of TF motif's frequencies (Fig. 2d & Extended Fig. 6g-i). On the other hand, such analysis can partly be achieved by computing a "gene set" score for each TF's target genes (e.g., from DoRothEA database) or by 5' based scRNA-seq probing TSS. Also, the scRNA+ATAC generally gives minimal contributions to cell type identification compared to scRNA only. Therefore, it is important to highlight if there are more benefits from multi-omics, as it costs a lot more on sequencing than scRNA-seq only (e.g., MAESTER system).

3. In the analysis, more than 4,000 variants (e.g., young-1) have been detected out of ~16.7KB mitochondrial genome. The number sounds a bit too high that one out of 4 genome positions has undergone somatic mutations. This is probably related to the setting that the variant calling pipeline aims for high selectivity by selecting all variants that are present in >1 and <all cells. However, it is not clear whether a more stringent variant calling will improve or harm the downstream analysis in both clonality reconstruction and progenitor estimation. The more stringent variant calling can either be adding more criteria in their RedeemV pipeline or using alternative methods like mgatk/meagatk or MQuad.

4. Here, the authors used the variants in a binarization way, i.e., either present or absent. Such a setting is probably reasonable given the coverage is even (compared to MEASTER and mtscATAC). However, the allele frequency in each cell can be informative for indicating the kinship between cells, hence it is worth a comparison using the allele frequency directly. As another related note, although the analysis here focuses on "new" mutations for cell lineage analysis, it remains unclear how informative the allele frequency stochasticity is for lineage relatedness, especially those homoplasmic variants (i.e., the variants present in most or all cells).

A few minor comments to consider:

5. In the paper, it says that the mtDNA enrichment has been achieved by using ~142 probes (120bp each) but it is unclear whether the enriched mtDNA is from the RNA layer, ATAC layer or the DNA layer. It will be helpful to write it clearly.

6. The eUMI is a nice strategy to remove PCR duplicates (or even fragmentation duplicates). However, this strategy may largely be similar to the commonly used strategy in MarkDuplicates (Picard) and also the MAESTER analysis pipeline (ref 32). Though the implementation for support droplet-based single-cell data is valuable, it would still be good to mention the relevant tools and possibly highlight the difference.

7. The accuracy of eUMI is assessed by a simulated, where it is a bit unclear if this is based on an assumption of uniform distribution for the cutting sites. However, the uniformness may not broadly hold, hence it will be good to check the distribution of the cutting sites from data with real UMIs.

8. For the TF frequency (Fig. 2d, right panel), it is not very clear how it has been performed. Adding some extra details to the methods section will help.

9. For Fig. 2g (and also text on page 7), why is a relatively high cutoff on $FDR < 0.2$ chosen (though fold change > 2)? Is it because many clades are relatively small? If so, how about using a larger clade size (e.g., with lower resolution in tree cut) and will it affect the analysis?

10. In Fig. 2e, it's very interesting to see lower mutation burdens in HSC. Adding some discussion on the possible relation between cell differentiation and division (or the potential reason) can be very valuable.

11. Fig. 3f, it is unclear what statistical method is used for performing the enrichment analysis. Please write it in the text or figure caption.

Author Rebuttals to Initial Comments:

Response to Reviewer Comments:

We are grateful to the Reviewers for their valuable and supportive comments on our work. These comments and suggestions have been very helpful to us in guiding our revisions which involve new experiments, analyses, and textual changes. We feel that these changes both strengthen the major conclusions of our manuscript and make the work more accessible. We address each of the comments in depth in a point-by-point manner below. Our responses are in blue font, while the original comments are in black font. In addition, all changes in the text are highlighted in blue font. We highlight the following major changes and modifications to the manuscript:

- To bolster confidence in the reliability of the mtDNA mutations detected with ReDeeM, we have performed a number of additional analyses, including stratified mutational signature analysis, mtDNA eUMI simulations, etc. These analyses highlight that ReDeeM, with the single-molecule consensus correction, has achieved significantly higher sensitivity and greater accuracy compared to other related approaches including bulk sequencing of clonal populations (new **Extended Data Fig. 3, 4**, see also response to Reviewer 2 point 1 below).
- To strengthen our validation of ReDeeM's capability to perform accurate lineage inference, we profiled 4 additional K-Ras/Tp53 mutant lung tumors using a dual-lineage tracer experiment (CRISPR-based+ReDeeM) for further benchmarking. We have expanded our analyses with more interpretable quantifications, which demonstrates concordance at both the single-cell and clonal levels between the two orthogonal lineage tracing approaches (revised and new **Extended Data Fig. 5, 6**).
- To further support the validity of the clonal structure observed and the biological insights that can be provided, we have now included data from 5 additional healthy young donors and 3 additional older donors, including one with an identified CHIP mutation in bulk (ASXL1 Q373X). As expected, we still observe the oligoclonal expansions in all of the aged but not the young donors. Moreover, in the donor with the ASXL1 mutation, the expanded clones exhibited a notable depletion of erythroid cells, which mirrors the phenotype observed in *Asxl1* mutant mouse models (Nagase *et al.*, 2018; Fujino *et al.*, 2021). These new data thus reproduce the oligoclonal expansion in aging and demonstrate the potential to investigate the mechanism and consequences underlying these expansions (new **Extended Data Fig. 14**).
- We have expanded the analyses presented to provide more biological insights at the molecular level by taking further advantage of the single-cell multi-omic readouts we obtain (e.g., examining gene expression differences underlying different HSC clonal behaviors) (new **Extended Data Fig. 12**).

Referee #1 (Remarks to the Author):

In this manuscript, Weng et al. have studied the clonal architecture of human hematopoiesis using single-cell multiomics and mtDNA-lineage tracing. For the first time, the authors have applied these methods to study the bone marrow from aged individuals, resulting in a very exciting and rich new dataset. The senior authors are experts in the development of these single-cell tracing methods, having pioneered much of the early technology that is applied throughout this new manuscript (e.g. PMID: 32788668). As expected from such a team, this is a superbly written manuscript, with clear figures that are easy to understand and a very thorough methods section that should enable data reproducibility. I expect this manuscript to be a tour de force, provided some concerns are addressed.

The paper begins by describing a significant methodological improvement of single-cell mtDNA sequencing and variant calling, ReDeeM, which uses probes to capture the mt transcriptome, and then UMIs to correct for seq errors. Whereas previous methods had to rely on a few shared mtDNA variants to connect cells in clones, ReDeeM now identifies many more shared variants per cell, which increases the confidence of cell phylogenies. The authors even validate ReDeeM with a different method of phylogenetic mapping, which uses CRISPR-based evolvable barcodes in mice.

With this tool, the authors then describe a series of ReDeeM profiling experiments on 2 young and 2 aged bone marrow aspirate donors. The authors profile about 60,000 cells in total, across all samples, including, longitudinal samples to track how stem cell clones behave dynamically in situ. Previous attempts at this had been limited by the low number of connecting mtDNA variants, and thus a much poorer capacity to connect stem cells and their progeny. ReDeeM appears to excel at this, although some amount of data-collapsing (i.e. pooling clones into “clades” and using network propagation) appears to be required to be able to reach meaningful results. Perhaps the most exciting result from this part is that the authors obtain a hematopoietic fate decision map of human hematopoiesis in situ, built just from the somatic mtDNA variant data (Fig 2f), something that had been impossible using previous methods for somatic variant calling.

Next, the paper turns to analyze the epigenetic states and fate biases of different HSCs. Intriguingly, the authors identify a striking variety of cell-state clusters within human HSCs (which, surprisingly, is also one of the best available multiome datasets on human hematopoiesis), and then group the 5400 HSCs into 78 “clonal groups” based on shared mtDNA variants to study three major properties: i) their cell-state heritability, ii) their differentiation output, iii) their fate bias. For the first time in native human hematopoiesis, the authors show they are able to identify “clonal groups” with state heritability/stability, differences in output, and differences in fate biases, although the effect sizes of these differences appear to be small in all cases, likely due to methodological reasons.

Finally, the authors profile two aged individuals. Focusing their analyses on the clonal composition, they confirm that aged hematopoiesis is more oligoclonal, with specific clades of HSCs being expanded (increased “fitness”), some of which could be explained by the loss of the Y chromosome (LOY).

In summary, this manuscript addresses the technological challenge of studying clones in human native hematopoiesis with cell-state information, and it uses this clonal information to help define

differences in HSC function and state with better confidence and thoroughness than previous mtDNA-tracing studies. I believe this is a relevant and timely topic, with technological innovations such as these being able to drive the field, but I have a series of issues that would first need addressing. These are mostly:

- some technical questions on the method and its validation
- the limitations of studying “clonal groups” and connecting these with progeny through network-propagation strategies
- some additional analyses/comparisons of young and aged hematopoiesis

These points (and other minor ones) are detailed below.

R1-1

The mtDNA matrix is binarized “to minimize coverage bias”, but this could mean that cells with widely different heteroplasmy levels for a certain mutation end up considered as part of the same clone. Isn't ReDeeM supposed to be able to estimate and avoid this to a certain extent, based on eUMI counts? Is binarization really essential?

We agree that the heteroplasmy level does have the potential to be exploited in future efforts to further strengthen the power of the ReDeem approach to lineage tracing analyses. Indeed, we have seen anecdotal evidence for a few variants where high heteroplasmy and low heteroplasmy levels may mark different subclones. However, as a general lineage tracing framework, we believe binarization is more reliable and provides sufficient resolution, given the number of variants identified per single cell. Below is our specific reasoning:

- The fluctuation in heteroplasmy level may not faithfully indicate lineage relationships in the context of dynamic changes or when there is purifying selection. In theory, the heteroplasmy level changes when there is a bottleneck (i.e, very few mtDNAs pass on from one cell to its progeny and the mutation proportion changes due to random drift) or if there is purifying selection due to non-neutral variants. Of note, there could be multiple bottlenecks during the development and differentiation, resulting in reversible heteroplasmy level change. Furthermore, functional selection for a variant may lead to convergence for cells that have similar selection mechanisms (e.g., selection for specific metabolic requirements, etc), but that arise from different lineages. Both possibilities discussed above would confound the real lineage relationships. Therefore, we should be cautious in systematically incorporating the heteroplasmy level in analyses until we fully explore these scenarios.
- The majority of the mtDNA mutations have low heteroplasmy levels and limited variation across the cells that carry the mutation (Interquartile Range (IQR) of heteroplasmy levels less than 0.05 or 5%, Response-To-Reviewer or **RTR Fig. 1a**). Consequently, the inclusion of the heteroplasmy level does not result in a significant gain of information.
- The binarized mutations provide sufficient resolution. We are able to identify a large number of mtDNA mutations (4,788 mutations in young-1 HSPCs, **RTR Fig. 1a**). Most of these mutations only present in less than 100 cells (on average 14 cells for each mutation,

RTR Fig. 1b). Therefore, even without using the heteroplasmy level information quantitatively, this data should provide good subclonal resolution by using mutation combinations.

Nonetheless, we still agree with the Reviewer on the potential value of using heteroplasmy levels in some scenarios, especially in studying the bottlenecks and presence of functional selection. As a result, we have added options in the ReDeeM-R package to allow for heteroplasmy level-based analysis and have explained in the Methods (Page 41 [L1341-1343]).

RTR Figure 1. mtDNA mutation variation and frequency in young-1 dataset. (a) Summary of mutations with different levels of variation in heteroplasmy level (as fraction). IQR: Interquartile Range. (b) A histogram representing the frequency of mutations based on the number of cells they are present in.

R1-2

The agreement of closeness measure is very interesting way of comparing the two methods (mtDNA- and CRISPR-based lineage tracing), but I still cannot figure out how significant this result is. In other words, how many times is the ReDeeM fate-mapping method going to misclassify (false positive) two cells that are actually not related? And how relevant is this when clone-aggregation (clades) and network-propagation strategies are used for measuring HSC output and fate bias?

We thank the Reviewer for these helpful comments.

First, we clarify our methodology of “agreement of closeness” in Revised Extended Data Fig. 5e, where we illustrate that we first define “clonal neighbors” in mtDNA mutation space and then compare the distance rank for these “clonal neighbors” in CRISPR indel space versus random reshuffling over 1000 times. This would generate the agreement of closeness with the permutation-based p-value that indicates significance of closeness at the single cell level. As is shown in Revised **Extended Data Fig. 5f**, most single cells show significant closeness with a p-value less than 0.05. In addition, we provided a more interpretable global metric to assess the

significance of this result: positive agreement of closeness (AOC) ratio which describes the fraction of cells with positive agreement (AOC>0) for the entire tree or a group of cells. On average, 0.74 or 74% of cells across the 6 tumors show positive agreement of closeness.

To further improve the interpretability of the dual lineage tracer experiment, we performed clonal group level analyses as suggested. First, using the same method of clonal group identification in Fig. 4 and Fig. 5, we defined clonal groups for the dual lineage tracer KP tumor data (shown as colored bars underneath the trees in the revised **Extended Data Fig 5f**). We computed the positive AOC ratio for each clonal group, shown as fractions within each colored bar. We found that the positive AOC ratio ranges from 0.5 to 0.92 across different clonal groups. Second, we computed the Rand index to measure the similarity between the clonal groups defined by mtDNA mutations and the “clonal clusters” identified in CRISPR indel space. We tested several different clonal clustering algorithms. The Rand indices may vary depending on the algorithm but are generally stable with a reasonable average value of 0.64 (**Extended Data Fig 5f, 6c**). Both the positive AOC ratio and Rand index provide more interpretable assessment at the clonal aggregation (clade) level and provide additional confidence for the downstream analyses of HSC output and fate bias. Notably, we would also emphasize that mtDNA mutation and CRISPR-defined clones are not expected to be precisely identical. This is because the precise definition of a “clone” is somewhat arbitrary, and the occurrence of CRISPR-driven indels and spontaneous mtDNA mutations may not be simultaneous during tumor development, especially considering the relatively shorter time in which mutations can accrue in the mouse model (6 months). Therefore, this benchmarking provides a minimal estimate of the accuracy of the mtDNA based lineage tracing we perform.

To further validate this benchmarking, we also performed an additional batch of dual lineage tracing experiment with 4 new tumors. Consistently, we observed reasonable agreement of closeness and Rand Indices in the new samples (New **Extended Data Fig 6**).

We have revised the manuscript based on these additional data and analysis (**Extended Data Fig 5, 6**, **Page 6 [L160-164], 35-36 [L1128-1130], 41-43 [L1361, L1381-1390, L1410-1421]**).

R1-3

In the approach of selecting lineage-informative mtDNA mutations for building the human hematopoietic lineage tree, how was it determined if a barcode was randomly distributed? How many total cells and “barcodes” were used for this analysis? Also, are these “lineage-informative” mtDNA mutations ever found in HSCs? If so, how many? I suspect they will be quite rare, and thus it will be quite difficult to find them frequently in HSCs.

We apologize for not explaining the method with enough details in the manuscript. Specifically, we first grouped all differentiated cell types into 4 major trajectories: myeloid (GMP, MDP, Mono), lymphoid (CLP, ProB, CD4, CD8, B, NK), megakaryocytic (MK progenitor), and erythroid (MEP, Erythroid progenitor). The frequency of each mtDNA mutation is compared between any two differentiation trajectories using a binomial test. If the p-values of all comparisons are greater than

0.05, the mtDNA mutation is defined as “randomly distributed”. We filtered out all randomly distributed mutations and generated the list of “lineage-informative” mtDNA mutations. We have added these details into the Methods section (Page 44 [L1426-1432]).

For young-1 in Fig. 2f, the analysis is derived from a matrix with 11,009 cells vs 631 variants. We also added this information into the figure legend (Fig. 2f, Page 25 [L727]).

As the Reviewer expects, the lineage-informative variants are indeed rarer in HSCs. We further analyzed the distribution of “lineage-informative” mtDNA mutations between HSCs and differentiated cells (RTR Fig. 2). Compared to other mutations, the “lineage-informative” mutations are significantly less likely to be found in HSCs ($p\text{-value} < 2.2 \times 10^{-16}$, Fisher’s Exact Test, Example mutations are shown in Extended Data Fig. 7b). Of note, we do not expect them to be totally absent in HSCs, because in theory, there are two types of “lineage-informative” variants. One is a group of mutations that have newly arisen during differentiation, which should be absent in HSCs, while the other is present in HSCs with differentiation biases.

RTR Figure 2. Frequency in HSCs between “Lineage informative variants” versus other variants.

R1-4

“Clonal groups” seem to be a rather arbitrarily designed method for grouping HSCs. For instance, why do they have to contain at least 50 single cells? Why that number? Also, why enable using just one shared (binarized) confidently assigned mutation to call a group? Shouldn’t authors take advantage of ReDeeM and increase confidence with more than 1 variant? As a note, what happens to the “agreement of closeness” if one chooses a similar grouping approach when using CRISPR-based benchmarking? What is the agreement of closeness of “clonal groups” in that case?

Yes, the Reviewer is correct. The “clonal group” is a somewhat arbitrary concept depending on parameters that determine the resolution of what we call a “clonal group”. In fact, the “clones” or “clonal groups” are in general arbitrary terms, because eventually every single cell is one clone. The motivation to define the “clonal group” in this study is to increase power (such as when assigning progenies), as well as to perform some comparative analysis with better interpretability. Specifically, our method to define “clonal groups” is designed to best reflect the tree structure with a specified resolution, by cutting the tree down iteratively to branches and sub-branches as long

as a reasonable number of cells are contained in both daughter branches. Therefore, the minimum size (default is 50) only affects how deep we cut the tree, but the general clonal structure would not be affected since it is determined by the tree topology.

Because there are arbitrary parameters involved in defining “clonal groups,” we are careful in drawing conclusions from it. When comparing the clonal diversity between young and aged donors, we tested varying parameters, e.g., min size=25, 50,75, 100; etc. The observation of multiple clonal expansions in aged donors holds true irrespective of what parameters we select (**Extended Data Fig. 13c**).

To clarify, the cutoff “1 or more than 1 variant assigned” is not the number of shared mutations in a clonal group, but the number of variants assigned to a single edge (“edge” refers to a line that connects two nodes in the phylogenetic tree). We only used this parameter to indicate confident branches before cutting the tree. We added more explanations in the Methods section for clarification (Page 45 [L1470-1471]). The actual number of shared mutations in each clonal group is usually much more than one. Although this information is not discussed in the manuscript, we provided a new function *Get_Clonal_Variants* in ReDeeM-R, which identifies the specific mtDNA mutations for each clonal group (Fisher’s Exact test with q-value correction, Default FDR<0.05). For example, for young-1 datasets (Related to **Fig. 5b**), each clonal group on average shares 7 specific mutations (FDR<0.05, mean odds ratio=208.1, **RTR Fig. 3**).

The agreement of closeness is a metric at the single cell level. But, we agree, it also helps for comparisons at the clonal level. This is discussed in the second point brought up in this RTR by the Reviewer. We tested several different clonal clustering algorithms for CRISPR-based lineage tracing. The Rand index is on average 0.64, suggesting a reasonable agreement at the clonal group level.

RTR Figure 3. # of specific mtDNA mutations in each clonal group.

R1-5

Based on the fact that “clonal groupings” are studied, instead of the fates of individual stem cells, the manuscript is sometimes clear on this, and sometimes not, assuming that the data are revealing the fate decisions (and fate heterogeneity) of individual HSCs. Rather, the “clonal groupings” used in the manuscript are likely to have an origin at some point during early development (based on their mtDNA burden, their size, and the BM aspirate sampling rate, I think it may even be possible to estimate when). I think it may be important for readers to fully grasp this as part of the main text.

The Reviewer is exactly right about what we think of as the HSC clonal group. We revised the manuscript to clarify the definition of HSC clonal group when we first introduced the term as follows (Page 9 [L270-271]): “For clarity, the term “HSC clones” or “clonal groups” used hereafter refer to a group of HSCs that share origins during development, rather than referring to individual HSCs.”

We would love to see the mtDNA mutations be used to potentially estimate when the clones form during development. There are several prerequisites we need to test and model before we can achieve this. For instance, it is unclear how stable the mtDNA mutation rates are during development, which might prevent such analyses. How the mtDNA mutations dropout during different stages of development may also alter these estimates. Therefore, it is challenging, but also very intriguing to pursue with more data and analyses in the future.

R1-6

The SCAVENGE-L method to use network propagation to connect HSCs with their progeny is very interesting but I worry about its classification error. The authors “validated” the method with the HSC clonal groupings, but these were created with more stringent rules. My understanding is that many (or even most) of the connections the authors make between HSCs and committed cells are being made through very large distances. This is not just due to the sparsity and data coverage, but because of the extreme cellular undersampling that is taking place when performing a BM aspirate. I fully understand the limitations in terms of clinical sampling and the will of the authors to extract the most information from their analysis. But perhaps it would be important to add this in a section mentioning the limitations of this study. It is definitely possible that the relatively small effect sizes that are being detected (Fig 4b and 4e) are partly driven by the chosen approach.

We agree with the Reviewer that the undersampling of HSCs in bone marrow aspirates limits the resolution of HSC clonal groups for examining cell-state and differentiation biases at the clonal group level. It would be interesting to see how the effect size may change with a better sampling in the future. We added the following discussion in the revised manuscript to highlight this limitation (Page 15 [L456-463]):

“Of note, we describe behavioral and cell-state biases for HSC clonal groups rather than individual HSCs. The cells in a clonal group share common ancestors at some point during the lifetime of each individual. Due to the limited sampling of cells in bone marrow aspirates, the HSCs in a clonal group may not be the most immediate siblings. Therefore, further improved sampling with increased cell number, locations, and time points will provide an improved view of the phylogenetic relationships and is crucial to reveal the mechanisms underlying the observed cell-state and behavioral biases for smaller clonal groups (i.e., more recently derived clonal groups) and even single HSCs in humans.”

R1-7

To my understanding, this is the first time someone connects HSC (clonal group) fates and states in native human hematopoiesis. However, the authors do not try to make any connections between the data in Fig 3f and Fig 4b/e. Are there any transcriptional or chromatin-accessibility differences between high and low-output HSC clones (clonal groups)? Are there any differences in myeloid- versus mega- versus lymphoid-biased HSC clones (clonal groups)?

While an extremely unique dataset, none of the findings in the aged hematopoiesis experiments seem to be very novel. Of course, it is good to validate previous findings with a new method, but I feel this dataset should be analyzed a bit more. Are there any differences in output in the aged HSC clonal groups (compared to the young)? What is the cell-state heterogeneity in the aged HSC clonal groups? Are there any transcriptional or chromatin-accessibility differences that characterize HSCs of different fitness groups in the aged setting?

We thank the Reviewer for this valuable comment. We agree that making connections between the gene expression/chromatin accessibility and the clonal behavioral biases is a very important and unique analysis we can do to better understand the mechanisms underlying clonal biases. We added more analysis and a new extended data figure (**Extended Data Fig. 12**) and revised the manuscript accordingly to discuss this point (Page 12 [L355-366], 46-47 [L1534-1565]).

To survey the potential molecular drivers of clonal biases, we developed a “clonal behavioral trajectory analysis”. Essentially, we ranked the 78 HSC clonal groups based on their behavioral metrics: 1) output activity, 2) myeloid biases, 3) erythroid biases, 4) megakaryocyte biases, and 5) lymphoid biases. Along each of those trajectories, we conducted regression analysis that examined the relationship between gene expression/peak intensity and each behavioral metric. We only identified a very limited number of differentially expressed genes, but in contrast, we found a considerable number of peaks that are significantly associated with one or more behavioral trajectories (2,931 differential peaks, FDR<0.01, **Extended Data Fig 12b, Supplementary Data 5**). We clustered all differential peaks into 5 peak groups based on the association patterns with different behavioral measurements (**Extended Data Fig 12c**). Next, we investigated all nearby genes for each of the peak groups by Gene Set Enrichment Analysis. Interestingly, the functional terms enriched for each peak group are consistent with the clonal specific behaviors. For example, there are 756 peaks that are differentially accessible in HSC clonal groups with lymphoid biases. We found that the genes near those peaks are significantly enriched with immune and lymphoid stem cell functions. Notably, most of those genes are not

showing up as differentially expressed genes and some are not even expressed in HSCs. We also performed a DNA binding motif analysis for each differential accessible peak group. We are able to identify several candidate motifs that might play a role in shaping the differential epigenetic landscapes that impact HSC clone specific behaviors. Taken together, these analyses suggest that the HSC clonal behavioral biases are likely encoded by epigenetic priming for genes that are necessary for downstream differentiation, which echoes previous reports (Weinreb et al. 2020; Lara-Astiaso et al. 2014). Nonetheless, further validation and studies are needed to confirm and further develop these observations.

We agree with the Reviewer that deciphering the mechanisms underlying changes in HSC function with aging would have tremendous value. Unfortunately, in aged donors, due to the limited cell numbers available from samples, we were unable to enrich sufficient HSCs as we did for young donors. Therefore, with our current sampling approaches, such questions cannot be addressed, but we hope to pursue these types of questions in future studies by improving our sampling methods.

R1-minors

In the methods, “sequencing” section, it is mentioned that 50-60,000 reads (per cell?) are used for mtDNA libraries. Is this coverage really necessary? Perhaps I am not understanding Ext Fig 1a.

50-60,000 reads per cell is needed to make full use of the eUMI-based consensus variant detection in ReDeeM. The purpose of Extended Data Fig. 1a is solely protocol optimization and does not involve sequencing at a sufficient depth for accurate consensus variant calling.

In a typical experiment, we detect 8,000 unique mtDNA fragments per cell. Importantly, multiple reads (Ideally 3 or more than 3) per unique fragment is the key to taking advantage of eUMI to identify real mutations against PCR and sequencing errors based on consensus strategy. To achieve that, ~50-60,000 reads per cell would be recommended (In theory $3 \times 8,000 = 24,000$ reads per cell minimally), given that the distribution of the number of reads per fragment may be skewed. However, it depends on the research goals and the different tissues with different mtDNA abundance. In situations where lower resolution is acceptable (e.g., to distinguish major subclones in a cancer sample), lower sequencing reads per cell is sufficient.

Figure 3d is called after Figure 3e.

Thank you. We have now changed the order of 3d and 3e and revised the text accordingly (Page 9 [L255, 269]).

ExtFigure 6g is called after ExtFig 6h-i.

Thank you. We have now changed the order of Ext Data Figure 6g and 6h-i (Now **Extended Data Figure 9**) and revised the text accordingly (Page 8-9 [L253, 255]).

In the methods section, “progines” instead of progenies.

Thank you. We fixed this error (Page 46 [L1514]).

- Alejo E. Rodriguez-Fraticelli

Referee #2 (Remarks to the Author):

The manuscript by Weng and colleagues presents a method, ReDeeM, for deep mitochondrial genotyping from single-cell mutli-ome data and thereby identify clones and ‘phylogenies’ of cells. If the proposed method works as claimed, it would constitute a major step forward and be of maximal interest, as it permits lineage tracing in human.

The authors then go on to apply their method to study clonal output and relationships in hematopoiesis; their biological findings in this context are original and of interest, but do not fundamentally add beyond what has been published in the last years both from the mouse and human system (e.g. work by Fernando Camargo, Hans-Reimer Rodewald and Peter Campbell). Therefore, in my opinion, the manuscript should be primarily evaluated on the claim that ReDeeM can reliably identify clones (sets of genetically related cells) and phylogenies.

We thank the Reviewer for appreciating the originality of our method and the biological implications. In terms of the concern regarding conceptual advance in comparison to prior mouse and human work, we would emphasize the following points:

- The clonal contributions and biases of HSCs in native hematopoiesis remain a controversial unanswered problem in the field, even in mouse systems. Therefore, it is crucial to utilize different and improved lineage tracing systems to elucidate HSC behaviors.
- Mouse and human hematopoiesis show differences in many aspects. Compared to the mouse system, the HSC clonal specific state and behaviors in humans are largely unknown. Our study provides new insights to fill this knowledge gap.
- Compared to using single-colony whole genome sequencing in humans, our method allows us to further survey the cell states (gene expression/epigenome, etc), as well as measure the differentiation behaviors that happen at steady state in humans *in vivo*, which provides new insights in molecular mechanisms of HSC behaviors in health and diseases (See response to Reviewer’s conceptual point 5). In addition, our approach allows for high-throughput readout which is essential to investigate the large number of HSC clones with high heterogeneity. Moreover, the ability to directly study cells of interest, rather than relying on growth of colonies enables many further analyses.

I do not doubt that mitochondrial genomes occasionally mutate somatically, and that such somatic mutations can then be used as lineage tracers. I am not convinced, however that this occurs at a sufficient rate to assign more than a few cells to clones and even assemble phylogenies.

- The frequency of mutations reported here conflicts with what is known about the frequency of somatic mitochondrial mutations in clonal hematological malignancies (see my first point)
- Controls included here are not convincing (see my 2nd and 3rd point)
- There are reasonable alternative biological explanations for the observations (see my 4th point)
- Hence the authors would need to provide strong further experimental validation of the clones and phylogenies they identify (see my 5th and most important point)

Besides these conceptual points, I have several technical points listed below, in particular regarding statistical analysis.

Conceptual point 1

Leukemia, in particular AML, involves a rapid and massive clonal expansion of a single HSC. Therefore, if really every HSC has (somewhat) specific mitochondrial mutations, in every leukemia patient there should be somatic mitochondrial mutations when comparing to germline. An analysis of PCAWG data (Yuan et al., *Nature Genetics* 2020, DOI: 10.1038/s41588-019-0557-x) suggests that this is only the case in ~30% of AML cases and <20% of Myeloproliferative Neoplasms (Figure 1f), and this includes mitochondrial mutations that occur after the leukemia initiating genetic event. Accordingly, ReDeeM should only identify mitochondrial mutations in maximally 20% of cells, rather less. Here, several mutations are reported in most cells.

This raises doubts on the validity of the results. How would the authors explain this difference?

We agree that the validity of the mtDNA mutations we detect in this study is fundamental to the downstream lineage tracing analyses we perform.

We thank the reviewer for the helpful suggestion of comparing our results to the PCAWG data analysis study. The number of mtDNA mutations reported in the cited Yuan et al., *Nature Genetics*, 2020 paper is actually not contradictory with our observation. The results in Yuan et al., Figure 1f specifically summarize non-silent mtDNA mutations since they were focusing on “potential functional contribution of mtDNA mutations”. However, they detect many more silent mutations that are not included in Figure 1f. Indeed, in their Figure 2a, the authors presented the distribution of the total number of mtDNA mutations across all cancer samples from which they reported “around three somatic substitutions per sample on average, with a standard deviation of 2.6”. We further re-analyzed their supplementary data to specifically look at the number of mtDNA mutations in myeloid leukemia samples, which also shows ~2-3 mtDNA mutations per sample on average (**RTR Fig. 4**). Therefore, the data from Yuan et al., provide strong support for our observation that there are multiple mtDNA somatic mutations in most if not all single cells.

RTR Figure 4. Number of mtDNA mutations in each Myeloid sample in Yuan et al., Nature Genetics 2020

The number of mtDNA mutations detected in Yuan et al., is still somewhat lower compared to what we achieve with ReDeeM. This is likely due to the differences in sensitivity between the approaches used in these two studies. Conventional NGS (such as Yuan et al., Mitchell et al.) have limited mitochondrial variant detectability (variant allele frequency, or VAF 1%-2%) due to PCR and sequencing errors even with deep sequencing. By contrast, ReDeeM introduces the eUMI into the mtDNA mutation calling pipeline, which removes both PCR and sequencing errors thereby dramatically increasing our ability to detect mtDNA mutations even at low heteroplasmy levels (see the new **Extended Data Fig. 3**). To demonstrate this principle, we simulated low heteroplasmy variants in one single cell with 1,000 mtDNA copies and computationally fragmented the mitochondrial genomes into small pieces, followed by running this through the ART sequencing simulator (Huang *et al.*, 2012). We compared mutation calling using ReDeeM-V pipeline versus conventional NGS method. As expected, most low heteroplasmy mtDNA mutations clearly stand out after the ReDeeM-V pipeline but cannot be distinguished from error background using conventional methods (**Extended Data Fig. 3c-d**). Together, the number of mtDNA mutations detected from ReDeeM is not surprising, given the significantly higher sensitivity. Moreover, this is consistent with the report of high mutation levels from recent studies using UMI-based ultra-sensitive sequencing technology (Duplex-seq) for mitochondrial genomes in bulk tissues (Arbeithuber *et al.*, 2020; Sanchez-Contreras *et al.*, 2023).

The high mtDNA mutation number is also supported in theory. Mitochondrial DNA is more susceptible to DNA damage with a more than 10-100 fold higher mutation rate than the nuclear genome (Brown, George and Wilson, 1979; Yakes and Van Houten, 1997; Xu *et al.*, 2012). In theory, the acquired mutation number = mutation frequency/site/year *age* 16.5kb* mtDNA copies per cell. The mitochondrial somatic mutation was estimated as $4.02 \pm 1.81 \times 10^{-7}$ SNV/site/year (Williams *et al.*, 2013). One HSC cell approximately has 500-1000 mtDNA copies.

Accordingly, in an adult at 30 years of age, one single cell in theory carries 96.48-279.84 mtDNA mutations. This rough estimation may vary depending on many factors, but it provides an approximate theoretic basis for the high mtDNA mutations in adult tissues.

The mutational signature is a strong and commonly used metric to support the validity of mitochondrial mutations (Yuan *et al.*, 2020). To confirm the validity of all mtDNA mutations identified by ReDeeM, we further stratify all mtDNA mutations into 4 groups with high to low VAF. Clearly, the mutational signatures for all 4 groups of mutations (including the mutations with low VAF) are strongly enriched by transitions (C->T/ T->C), which are classic true mitochondrial mutation signatures (RTR Fig. 5).

RTR Figure 5. Stratified mutational signature. All mutations are stratified into 4 groups based on variant allele frequency.

Moreover, in practice, we built the ReDeeM-V with careful consideration to avoid potential artifacts. In addition to the use of eUMI, the paired-end overlapping, strand-biases modeling, potential NUMT, etc. add to our confidence in mutation calling (See **Methods**, Page 39 [L1181-1235]).

Taken together, we have strong confidence in the validity of mtDNA mutation calling, given all of these analyses. We have added a new **Extended Data Fig. 3** and revised the manuscript that delineates these new analyses (Page 5 [L129-131], 39 [L1236-1249]).

Conceptual point 2

The most important control currently included is Figure EV3. Here, a genetically encoded CRISPR based lineage tracing system was used (in mouse) and the resulting clones are compared to the clones identified by the mitochondrial lineage tracing. The experimental design is elegant but the analysis is not convincing for several reasons:

- The examples from panel c seem cherry picked, all mitochondrial mutations should be shown (e.g. as a supplementary file if space is limiting). Even with the selected mutations, it appears that they often occur across different clones identified from CRISPR, does this not show that these mutations are not specific to clones? See also my 4th point.
- The CRISPR clonal identity (as in panel b) should be shown below the ReDeeM based trees in panel e to allow a real comparison.
- I don't understand the description of "Agreement of closeness" (panel d,e, Methods). Can the authors use established statistical tools to compare between CRISPR-based clones and ReDeeM-based clones, such as the Rand index?

We thank the Reviewer for the helpful comments.

There are indeed many more supportive data than the examples shown in Figure EV3c (or revised **Extended Data Fig. 5c**). We chose three representative examples due to space limitations. We agree that showing the complete mtDNA mutation profile is helpful. Of note, we do not expect mtDNA mutations to be precisely restricted in a CRISPR-defined clone, since the definition of "clone" is somewhat arbitrary and the generation of indels and mtDNA mutations are not necessarily coupled at the same time during tumor development. To be unbiased, we look at the mtDNA mutation distributions on CRISPR-based neighborhoods. Clearly, the examples in Figure EV3c (or revised **Extended Data Fig. 5c**) show local enrichment compared to a random distribution. To further generalize, for any given mtDNA mutation, we computed the average distance among cells that carry the mutation in the CRISPR space and compared them to randomly picked cells. We show the histogram of this normalized mutation local enrichment score on CRISPR space of all mtDNA variants (Revised **Extended Data Fig. 5d, Methods**). All the mtDNA mutations are strongly shifted toward the local distribution (to the right) in CRISPR space, suggesting a general agreement for most of the mtDNA mutations.

To examine the reproducibility and to improve the power, we added a new batch of dual lineage tracer experiments with 4 additional tumors. We showed 20 more examples of mtDNA mutations that are locally enriched in CRISPR-indel-based 2D maps from two of the additional tumors (**Extended Data Fig 6b**). We also observed significant agreement of closeness in individual cells between mtDNA- and CRISPR-based lineage inference (**Extended Data Fig. 6a**).

We apologize for not fully clarifying the method of agreement of closeness. We added a schematic to better illustrate the concept (Revised **Extended Data Fig 5e**). Briefly, we define two distance matrices, one using mtDNA mutations and the other by CRISPR indels. For a given single cell, we first define k-nearest neighbors in mtDNA-based distance space, we then look them up in the CRISPR-based space and compute the distance rank to compare with 1000 random reshuffles.

The % rank closer in real CRISPR space versus random is defined as “agreement of closeness” with a permutation-based p-value for each cell. We present our results using the metric of agreement of closeness, because (1) it is a measurement at single cell resolution, and (2) it is directly based on distance matrices and therefore unbiased by different algorithms that are employed for clonal clustering or phylogenetic reconstruction.

Nonetheless, we agree that clonal group level analysis is also helpful in improving the interpretability. We added the clonal group identification and computed the positive agreement of closeness ratio for each clone (i.e., fraction of cells show positive agreement of closeness). We do not directly show CRISPR clonal identity below the tree due to the lack of clarity of the alternative visualizations by different clustering algorithms. However, we computed the Rand indices to measure the similarity between the clonal groups defined by mtDNA mutations and those identified in CRISPR indel space using different clustering algorithms. The Rand indices may vary due to different algorithms but in general are stable with an average of 0.64, suggesting a reasonable concordance among the detected clones (**Extended Data Fig 6C**).

Finally, we would emphasize that, as expected based on the shorter duration of the mouse model (~ 6 months), the mouse experiments have lower numbers of mtDNA mutations than we detect from human hematopoietic cells and therefore this benchmarking provides a minimal estimate of the accuracy of the mtDNA based lineage tracing we perform.

We have revised the manuscript based on these additional data and analyses (please see **Extended Data Fig 5, 6, Page 6 [L160-164], 35-36 [L1128-1130], 41-43 [L1361, L1381-1390, L1410-1421]**).

Conceptual point 3

Another observation from the manuscript that raises doubts on the central claim is figure 5f. Here ReDeeM was applied to an elderly individual with loss of chromosome Y. It is very surprising to see that loss of Y is only weakly enriched in some “clades” identified by ReDeeM and by and large stochastically distributed across the phylogeny. Are these enrichment p values accounted for multiple hypothesis testing (several clades were tested for enrichment, so this is necessary)?

LoY is a genetic mark that should be inherited in clades. In the Mitchell et al Nature 2022 paper a similar individual is analyzed with whole genome sequencing of colonies. Loss of Y, although it occurs several times, is there consistently observed within more expanded clades (see their figure 3c), as expected. The authors should validate their unexpected result, e.g. by WGS of HSC-derived colonies from the same individual. See also my 5th point.

We thank the Reviewer for this point and would like to elaborate on why we consider our observation to be concordant with what was observed in Mitchell et al., *Nature*, 2022. They reported two aged donors, one showing a high percentage of cells with LOY (65% in KX003, Figure 3c) while the other showed a lower percentage (1% in KX007, Extended Data Figure 5b). In fact, the aged population of the 70s-80s typically shows a low percentage of cells with LOY,

according to population-based surveys on bulk samples (Forsberg *et al.*, 2014). In our **Figure 5c** for donor aged-1, we reported 119 cells with LOY out of 9519 cells, which is more similar to KX007 with a lower percentage. Indeed, if we downsample our data into 350 cells, the visualization of the LOY cells on the tree look similar to that of KX007 in Mitchell *et al.*, Extended Data Figure 5b (**RTR Figure 6a**). Our data suggest that the LOY events occurred with low frequency but happened multiple times independently. This observation is consistent with the findings of Mitchell *et al.*, who reported “at least 62 independent loss of Y events” despite a more limited sampling size of 267 HSC/MPP colonies. Last but not least, the detection of LOY using single-cell ATAC fragments is constrained by its limitations, as it is prone to both false positives and false negatives. This is primarily attributed to the scarcity of accessible reads on chromosome Y at the single-cell level, typically averaging less than 10 reads per cell (**RTR Figure 6b**). We acknowledged this limitation and revised the manuscript accordingly (Page 13 [L407-413])

Together, these analyses suggest that the LOY in our donors with low frequency is expected and likely not the driver of the expansion. Instead, the weak enrichment we observed may suggest the susceptibility of LOY in the expanded cells, which is an interesting observation increasingly being made with single cell approaches whose basis remains elusive in the field. Our method provides new insights and generates a number of hypotheses that future studies should follow up on.

We appreciate the Reviewer's suggestion on multiple hypothesis testing correction. The enrichment test is performed on the 4 expansions. As suggested, we provided the adjusted p-value using the qvalue method in the revised **Figure 5f** and legend accordingly (Page 26 [L765]).

RTR Figure 6. (a) Downsampling of the phylogenetic tree of Aged-1. (b) Number of reads on chromosome Y per cell in young-2 and aged-1

Conceptual point 4

There could be an alternative explanation for the observations presented in this study: Heteroplasmy fluctuates over time. Hence, mitochondrial variants present in germ line might just sometimes be present at higher heteroplasmy (and be captured by ReDeeM, which captures

approx. 50 of “100s to 1000s” mitochondrial genome copies) and sometimes at lower heteroplasmy (and mostly drop out in ReDeeM). Over ageing, some germline variants will disappear by random drift.

Can the authors prove that the mutations arise *de novo* and do not exist in germ line? Deep sequencing of (bulk) mitochondrial genomes from other tissues of the same donor (or maybe even T cells) could be used to investigate this.

If they exist in germ line, can the authors exclude that there is no selection for variants in certain lineages? Work by Leif Ludwig suggests that this is sometimes the case (10.1101/2022.11.20.517242) and we have no clue of the extent to which this happens (The metabolic requirements of erythroblasts are quite different from those of neutrophils...). This would fundamentally limit the ability of such variants to be used for lineage tracing.

The majority of mtDNA mutations we observed are less likely to exist in the germline, given the number of confident mutations we identified. In a recent study, Chongwei Bi et al. (<https://doi.org/10.1101/2020.12.28.424537>) developed a UMI-based high accuracy long read sequencing technology that allows for the detection of mtDNA mutations with low heteroplasmy levels (as low as 0.004%). They applied this technology in single human oocytes. In the three human oocytes they studied, they identified 141, 81, and 69 unique heteroplasmic SNVs respectively. This low number of mtDNA mutations in oocytes is consistent with the bottleneck hypothesis during oogenesis (Zhang, Burr and Chinnery, 2018; Zaidi *et al.*, 2019). Of note, most of these SNVs have heteroplasmy levels lower than 1%, which may be further lost during preimplantation development when mtDNA is not replicated (Wolf, Hayama and Mitalipov, 2017). This method and ReDeeM both apply UMI-based error correction and therefore have comparable sensitivity in principle. In a typical ReDeeM dataset, we identify more than 4,000 unique heteroplasmic SNVs in a given donor, which is significantly higher than the expected number of germline mtDNA SNVs, as discussed above. Therefore, we believe that most of the mtDNA mutations identified by ReDeeM should be *de novo* and acquired in somatic cells. We appreciate the Reviewer's suggestion to (bulk) sequence the mitochondrial genome in other tissues. Unfortunately, the conventional NGS method is not sensitive enough (see response to the Reviewer's point 1) since most potential germline mutations show very low heteroplasmy levels.

We agree that a small number of germline heteroplasmic mtDNA mutations may still exist. However, those could still be informative as a lineage tracer. As illustrated in **RTR Fig. 7**, if there is a low heteroplasmic mutation in the zygote, it may get lost in one of the descendent cells at a certain point (e.g., in certain early HSC clones) as well as all of its following descendents. In such a scenario, even though the mutation is from the germline, it is still informative in inferring the lineage relationships.

RTR Figure 7. Illustration of lineage informative germline mtDNA mutations.

The potential mtDNA mutation selection is an important consideration. However, usually selection does not take strong effect until a high heteroplasmy level is reached based on previous reports (Durham *et al.*, 2007; Ye *et al.*, 2014; Stewart and Chinnery, 2015). Consistently, the mtDNA mutations under purifying selections that we and others previously observed indeed show high and variable heteroplasmy levels (Walker *et al.*, 2020; Lareau *et al.*, 2022). In contrast, most heteroplasmic mutations identified by ReDeeM in the healthy hematopoiesis have low heteroplasmy levels and do not dramatically vary across cells that carry the mutation. For example, in the young-1 dataset, we identified 4,788 heteroplasmic mtDNA mutations with median heteroplasmy level of 0.02 (or 2%) with low variance (**RTR Fig. 1**). Furthermore, for the small number of mutations with relatively higher heteroplasmy levels and higher levels of variation (187 mutations with IQR>0.2), we saw limited changes associated with differentiation (more details are provided in our response to technical point 3). Altogether, we think selection will have a limited effect on the lineage tracing analysis presented in this manuscript. Additionally, binarization of the mtDNA mutation data further minimizes the potential effect of heteroplasmy level change due to functional selection. Nonetheless, we agree that functional selection should be kept in mind in other analyses, especially in disease contexts, and will be interesting to investigate in the future when much more data from many more cells are obtained with ReDeeM.

Lastly, the observation of increased mtDNA mutation burden in aged donors compared to young provides further evidence suggesting that the mutations we detected are most likely acquired *de novo* in somatic cells.

Conceptual point 5

The authors should add more validations of their phylogenies and sanity checks.

- WGS from HSC derived colonies plus ReDeeM for the same individual would provide a gold standard validation of ReDeeM. The same phylogenies should be obtained with both methods.
- The publicly available sequencing data from Lee-Six *et al.*, Mitchell *et al.* etc. (WGS of HSC-derived colonies) should contain deeply covered mitochondrial genomes. Are data from these mitochondrial genomes sufficient to reproduce the clonal hierarchies that these manuscripts report based on nuclear mutations? Resulting phylogenies would need to be qualitatively and quantitatively compared. This has the advantage that technical dropout of single cell methods will

not be a problem, so it can provide a basic proof of concept for the idea that phylogenies can be obtained from mitochondrial mutations only.

We appreciate the Reviewer's comments and agree on the importance in validating the phylogenies.

WGS of HSC-derived colonies is a great approach that can detect nuclear somatic mutations well for lineage tracing. However, as was discussed in the response to conceptual point 1, conventional WGS (without UMIs) have limited mitochondrial variant detectability even with deep sequencing of single colonies, since most of the informative mtDNA mutations have low heteroplasmy levels (**Extended Data Fig. 3**). Therefore, unfortunately, the number of true mtDNA mutations that can be resolved by WGS of single colonies is not at the same magnitude as ReDeeM and would not enable a robust benchmarking of the phylogenies.

The oligoclonal expansions we observe in aged donors in comparison to younger individuals serve as a “sanity check” of the validity of the phylogenetic tree. To further confirm this, we collected 5 additional young donors and 3 additional aged donors, including one with an identified CHIP mutation in bulk (ASXL1 Q373X in aged-5). As expected, we observe oligoclonal expansions and reduced clonal diversity specifically in aged donors (**Extended Data Fig. 14d-g**). And interestingly, in aged-5, the expanded clones are depleted for erythroid cells, which is reminiscent of the phenotype observed in *Asxl1* mutant mouse models (Nagase *et al.*, 2018; Fujino *et al.*, 2021). Although further incorporating single-cell genotyping with ReDeeM in the future will be valuable for definitively determining the clones with the driver mutations and to survey the molecular mechanisms, these new data provide additional sanity checks, and demonstrate the potential to investigate the mechanism of clonal expansions in aging using ReDeeM, providing more biological insights compared to previously available approaches. We added an additional **Extended Data Fig. 14** and revised the manuscript accordingly (Page 13 [L384-386], 14 [L417-418, 421-427]).

In addition, the dual lineage tracing experiment with CRISPR-based method provides an orthogonal sanity check. As discussed in our response to conceptual point 2, we further added a new batch of tumors and significantly expanded the analysis (**Extended Data Fig. 5-6**). The lineage inference performance of ReDeeM strongly supports its reproducibility and robustness.

Technical point 1

Assuming de novo, spontaneous somatic mutation, is the way that phylogenies are computed ideal? There is very large technical noise here, since only a small percentage of all “100s of 1000s” mitochondrial DNA molecules per cell are covered, and heteroplasmy levels of the observed mutations are low. So evidently there is plenty of dropout (false negatives). In such a noisy setting, simple distance metrics for clustering are known to produce incorrect phylogenies/clonal hierarchies; the computational methods by the Beerenwinkel group (SCITE etc) and PhISCS by

Salem Malikic address this issue and infer correct phylogenies even in a noisy setting. PhiSCS is faster/copies with a larger cell and mutation numbers. Such methods should preferably be used.

We thank the reviewer for the comments and suggestions, and we agree that algorithmic optimization of phylogeny inference is an important goal. Having evaluated these existing methods, we identified the following limitations that limit their suitability for our large-scale mtDNA mutation-based lineage tracing data:

- Several assumptions that are based on the nature of the diploid nuclear genome do not apply to the mitochondrial genomes that have 100s to 1000s of mtDNA molecules per cell. For example, PhiSCS uses the “three-gametes rule” that assumes diploid to correct false positive/negative. Also, bulk variant allele frequency is used to estimate cellular prevalence, which also assumes two alleles in one cell.
- Some mtDNA specific biases are not considered in existing methods. For example, the variable mtDNA copy coverage in different cells is normalized when calculating distances in the ReDeeM pipeline but would not be considered to be corrected in existing methods.
- Scalability - The probabilistic and combinatorial methods are less scalable than distance-based methods. Our datasets typically contain 10 thousand or more single cells. A distance-based method is a viable solution for such a large-scale study.

Although with the limitations, we agree that a comparison between ReDeeM-based trees to some of the existing methods that can address noise in data would be valuable.

We applied the MCMC-based SCITE to benchmark the ReDeeM lineage tree. Given the limited scalability of SCITE, we randomly picked 100 cells from the human hematopoietic stem cell data set. We reconstructed the phylogenetic tree using the ReDeeM pipeline and SCITE. As is shown in the **RTR Fig. 8**, the tree structures are similar and most of the splits are present in both methods. We further quantitatively compared the ReDeeM-tree to 8 SCITE-derived maximum likelihood trees. On average, more than 60% of cell triplets are ordered correctly, suggesting a reasonable consistency. (The metric of “Triplet Correct” was defined previously, M.G. Jones et al., *Genome Biology*, 2020).

We agree with the Reviewer that distance-based phylogenetic tree reconstruction methods are not perfect. Moreover, addressing noise in the data is important. Although it is out of the scope of this study, it is of interest as a future direction to further increase the mtDNA capture efficiency to further minimize the false negative rate and to develop new probabilistic or combinatorial phylogenetic tree building algorithms that take into account the special features of mitochondrial genomes and with improved computational efficiency. We appreciate these valuable suggestions and have added discussions in the revised manuscript (Page 16 [L487-490]).

RTR Figure 8. Comparison between ReDeeM and SCITE-based phylogenetic tree.

Technical point 2

It would be good to know more precisely how many mtDNA copies there are per cell, so as to compute the false negative rate of this method and its sensitivity.

We did not have a precise measurement of the mtDNA copies for our samples. But the number of mtDNA copies in peripheral blood mononuclear cells, a comparable reference, has been estimated based on bulk sequencing (Rausser *et al.*, 2021). Typically, total mtDNA copies in hematopoietic cells range from 300-1000 per cell. ReDeeM captures 30-70 (median 51.7, Fig 1b) unique mtDNA copies per cell, suggesting a recovery rate of approximately 10%.

We appreciate the limitations of the false negatives due to the incomplete capture. We attempted to mitigate this caveat in the following ways. First, multiple mutations are used together for lineage inference to compensate for the dropout issue of particular mutations. Second, we binarized the mtDNA mutations to avoid the use of unreliable heteroplasmy levels when coverage is insufficient.

We discuss this limitation in the revised manuscript (Page 16 [L483-487]). Further improving the mtDNA capture efficiency would be valuable to maximize the sensitivity and is an important future direction that we hope to pursue.

Technical point 3

Related to figure 1e: Does the heteroplasmy of mitochondrial mutations change/increase during differentiation? This could indicate a selection of variants (see also main point 4).

We systematically investigated the mtDNA mutation heteroplasmy level using young-1 data as an example (RTR Fig. 1). As expected, the 32 “homoplasmic variants” that are present in most cells show little variation with a median heteroplasmy level of 0.99 (or 99%), and Interquartile range (IQR)=0. For the 4,788 “heteroplasmic variants” that are present in less than 75% of cells, most of them also show low variation of heteroplasmy level across cells that carry the mutation (4,601

mutations show IQR<0.2, median heteroplasmy level=0.02). For the 187 mutations with IQR>0.2, we examined the heteroplasmy level change across the HSCs, early progenitors, late progenitors, and differentiated cells (mutations with at least 3 cells in each category are analyzed). As is shown in **RTR Fig. 9**, most of these mutations do not show significant heteroplasmy level changes with differentiation. Together, these data suggest that the heteroplasmy levels of most mtDNA mutations do not significantly vary across cells. And for a small number of mutations that do vary, we do not observe obvious functional selection during differentiation in normal hematopoiesis. But we agree that functional selection should be kept in mind in other analyses, especially in the context of disease.

RTR Figure 9. Heteroplasmy level change during differentiation for mtDNA mutation with high IQR.

Technical point 4

In figure 2f, I do not understand the color scale. They could show a percentage (i.e. out of k clonal neighbors of T cells, y% are B cells).

We apologize for the lack of clarity. In **Figure 2f**, the underlying measurement is the percentage, exactly as the Reviewer suggested (i.e., out of k clonal neighbors of T cells, y% are B cells). Since different cell types have variable total cell numbers, the more abundant cell type tends to always show a higher percentage, which compromises the suitability of the percentage as a visualization metric. Instead, the color in **Figure 2f** is the fold change of the percentage from the median across cell types, followed by log transformation. This normalized percentage allows for a fairer comparison when visualizing this data. We have clarified this in the Methods (Page 44 [L1439-1440])

Technical point 5

In figure 3g, are p values corrected for multiple hypothesis testing? Take note that a total of 13 * (51+27) = 1014 statistical tests were performed. It is not enough to correct all tests for a single HSC subpopulation or clonal group.

Related, in the context of figure 4: “we identified 47 (60%) HSC clonal groups with significant lineage enrichment (binomial test $pt1 < 0.05$ & $pt2 < 0.05$), while 31 (40%) HSC clones showed low or no lineage bias. Is this adjusted for multiple hypothesis testing?”

We thank the Reviewer for the helpful question.

In the original Fig. 3g, we did not show the corrected p-values. We had confidence in this data, since most of the enrichments show the same trend and hold significance across both time points which are independent assays. But we agree with the Reviewer that multiple test correction is necessary in this case to present the final significance. We used the qvalue package to compute the adjusted p-value, or FDR. We revised **Fig. 3f-g** using the FDR value for plotting. The results are consistent with what we had seen previously. We also provided the raw data of the enrichment fold changes, the original p values as well as the FDRs as a new **Supplementary Data 4**.

In Fig. 4, we also adjusted the p-values using the qvalue package. The results are consistent with what we had seen previously. We revised **Fig. 4e** and provided the raw data of the enrichment fold changes, the original p values, and the FDRs. Raw data are included in the new **Supplementary Data 4**.

We revised Fig.3 and Fig.4 and added more details in the Legends and Methods section (Page 26 [L740-742, L753-755], 45 [L1477-1481], 46 [L1529-1532]).

Technical point 6

In the clonal analysis of lineage bias (Figure 4e) and HSC subsets (figure 3g) effects are rather weak, and typically biological measurements suffer from some form of overdispersion. Are the observed lineage biases stronger than what one would observe in a sample without lineage biases, measured with the same method? I would suggest the following control: They could take a BM sample, split into 4 fractions, stain each fraction with a different hashing antibody, pool at some ratio (e.g. 5:10:25:60), and perform ReDeeM. Then they could check if different clones are “enriched” for different hashing antibodies. If that’s the case, the statistical test in figure 4e needs to be adjusted to account for technical overdispersion (e.g. beta-binomial test).

We thank the Reviewer for the comments and suggestions. We agree that we should rule out the possibility of technical overdispersion.

Our experimental design should be robust to pick the real biological signal rather than technical overdispersion. We collected the bone marrow from the same donor twice independently. These two collections serve as two independent biological replicates. Although most of the clonal specific differentiation biases are moderate, the enrichment trend is consistent across two time points. We provided detailed enrichment fold change p-values for both time points in **Supplementary Data 4**. The consistent enrichment from two independent samplings suggests that this observation is unlikely to be due to technical overdispersion.

Technical point 7

Data availability: It would be good to provide the data and annotations (clones, cell states, dimensionality reductions) also as a ready-to-use Seurat (or scanPy) object on a platform such as figshare.

We thank the Reviewer for the helpful suggestion. We have shared the Seurat objects (as RDS files) with annotations on figshare. The data are accessible with the doi: <https://doi.org/10.6084/m9.figshare.23290004>. We have also revised the Data Availability in the manuscript (Page 49 [L1627-1628]).

Referee #3 (Remarks to the Author):

In this manuscript, Weng et al report the development of a single-cell multi-omics methodology for simultaneous RNA-seq/ATAC-seq/clonal tracking(mtDNA-based) and use it to investigate human hematopoiesis in two young and two aged donors. The major methodological advance derives from improved recovery of mtDNA sequences by improving yield and applying a target-enrichment strategy.

Using this methodology, the authors derive a detailed snapshot of hematopoiesis including clonal output, lineage contribution, hematopoietic stem cell (HSC) numbers and lineage bias. By analyzing the same (young) person twice, 4 months apart, they provide insights into the stability of HSC states and lineage output/bias. By comparing two young (26 and 31 yrs) with two aged (76 and 78 yrs), they capture the oligoclonality and presence of large clones in the latter.

Whilst some of these findings are not novel in isolation, the ability to capture them simultaneously represents an important advance and I expect that the methodology will be used to study hematopoiesis, in many different contexts, in a detail that has not been possible before.

Below are some suggestions aiming to further improve the manuscript:

R3-1

1. The heteroplasmic nature of mitochondria and their inheritance introduces caveats in the use of mtDNA mutations as clonal markers (for example mtDNA mutations can be lost by stochastic drift in heteroplasmy). The authors should discuss this briefly, as readers will not necessarily be fully aware of the implications. Importantly, the demonstration that ReDeem agrees well with CRISPR-based lineage tracing provides important reassurance in this regard. However, the concordance between ReDeem and CRISPR was not complete. Overall, I am persuaded that the methodology is robust enough to support the authors' findings/conclusions. However, it would add to my confidence if the phylogenetic relationships are seen to be retained after pooling multiple

donor samples. This, for example, would provide confidence that the technical and analytical approaches are resistant to experimental conditions etc. So, pooling donors and re-running algorithms would be a better test of the robustness of the methodology.

Yes, the heteroplasmic fluctuation and dropouts due to stochastic drift are limitations for mtDNA mutation-based lineage tracing. We attempted to mitigate these caveats by the following approaches: First, the enhanced detectability of the mtDNA mutations enables us to establish lineage relationships based on information derived from multiple mutations, rather than relying solely on a single mutation, which is crucial in mitigating the dropout issue. Second, we binarized the mtDNA mutations, which avoided the potential biases due to fluctuation in heteroplasmy. Nonetheless, we appreciate that these designs do not fully eliminate the issue, and the mtDNA mutation-based lineage tracing is still not a perfect system. We now discuss these caveats in the manuscript, as suggested (Page 16 [L483-487]).

We thank the Reviewer for the appreciation of the dual lineage tracing experiment with CRISPR-based barcoding. The Reviewer is correct. The agreement between ReDeeM and the CRISPR-based inference is incomplete, which is expected to some extent, since the KP mouse experiment occurs in a short time window (6 months), and the mtDNA somatic mutagenesis and CRISPR-induced indel generation do not necessarily happen at the same time scales. To further validate the findings, we performed a new batch of this experiment (4 new tumors) and expanded our analysis (Revised **Extended Data Fig 5, 6**). We observed reasonable agreement across all samples analyzed. We have revised the manuscript based on these additional data and analysis (Page 6 [L160-164], 35-36 [L1128-1130], 41-43 [L1361, L1381-1390, L1410-1421]).

We thank and agree with the Reviewer for the suggestion to pool multiple donor samples. We collected 5 new young donors and 3 new aged donors. As suggested, the samples from different donors are hashed and pooled together (5 young donors are pooled together, 3 old donors are pooled together) followed by application of the ReDeeM protocol. The pooled donors can be definitively separated using only information from the mtDNA mutations (i.e., ignoring the hashing information) (**RTR Fig. 10**). After demultiplexing and rerunning the algorithms, we reconstructed the phylogenetic tree for each of these donors. As expected, we still observed the increased mtDNA mutation burden as well as oligoclonal expansion in aged donors (New **Extended Data Fig 14**). These new data and analyses suggest the robustness of the methodology that is resistant to experimental conditions. We revised the manuscript accordingly (Page 13 [L383-386], 14 [L417-418]).

RTR Figure 10. Clustering by mtDNA mutations for pooled donors, colored by cell hashing-defined donor labels.

R3-2

2. The use of BM (rather than blood) cells enriches for progenitor cell types, but may give a biased view of phylogenetic relationships given the physical proximity of sampled cells. Whilst this was not observed in colony-based studies, but these only analyzed very few samples and relatively small numbers of cells/colonies. Analysis of blood-derived HSPCs from at least one of the four donors would be very helpful and would give additional insights into the relationship between BM-resident and circulating HSCs/HSPCs.

We agree with the Reviewer that the sampling is critical in studying the phylogenetic relationships that occur in hematopoiesis.

We attempted to minimize the sampling bias by the following: First, bone marrow samples were obtained by taking multiple aspirates with needle rotation, ensuring that sampling was not limited to a single location in the marrow cavity. Second, we independently collected bone marrow samples from the young-1 donor on two occasions, with a 4-month interval between them, which served as two biological replicates to further mitigate the sampling biases. In the young-1 HSCs, we examined the cell compositions from time point 1 and time point 2 for each HSC clonal group. We found that the majority of the clonal groups are well captured in both time points, suggesting that the sampling is not biased and enables accurate representation of the HSC clonal structure (**Fig. 3e**).

To increase power, as mentioned in the response to point 1, we collected 5 new young donors and 3 new aged donors. We reproducibly resolved the polyclonal structure in young donors and observed consistent oligoclonal expansions in aged donors.

We appreciate the Reviewer's suggestion to analyze blood-derived HSPCs. This would be a very insightful experiment to learn more about the relationships between BM-resident and circulating HSCs/HSPCs. Unfortunately, we have not been able to obtain enough blood-derived HSPCs from our donors without mobilization. This is something we hope to pursue in the future and we appreciate this insightful suggestion.

We have now discussed these limitations in the manuscript (Page 15 [L456-463]).

R3-3

3. The aged donors display fairly large expanded clones, even in the absence of clonal hematopoiesis (CH) driver mutations. Previous publications (e.g. Mitchell et al, Nature 2022) have demonstrated this to be common in older individuals. However, it would be good to know that this is what the current manuscript/methodology is also showing (i.e. apparently "driverless" expansions). So, it would help to determine whether CH driver mutations can be identified in bulk DNA from the aged donors using conventional targeted sequencing (the clade sizes suggest that any such mutations should be identifiable in bulk DNA).

We appreciate these helpful comments from the Reviewer.

To address this, we performed the conventional targeted sequencing on aged-1,-2,-5 donors, for which sufficient DNA materials were available. Given the limited number of aged donors we have, we cannot comment on how common the "driverless expansion" is, although this will be a focus of future studies. Here, we did not detect known CHIP mutations in aged-1, whereas in aged-2 and aged-5, we confidently identified known CHIP mutations in *DNMT3A*, *FLT3*, and *ASXL1* (**RTR Table 1, Fig 5, New Extended Data Fig. 14, Methods**). This anecdotal evidence supports that CHIP mutations (at least at the limits of detection for existing bulk approaches) may or may not be present in aged donors with clonal expansions. Interestingly, in aged-5, the expanded clones are depleted by erythroid progenitors, which is reminiscent of the phenotype observed in *Asxl1* mutant mouse models (Nagase et al., 2018; Fujino et al., 2021). Although out of the scope of the current study, incorporating single-cell genotyping with ReDeeM in the future would be needed to determine the clones with the driver mutations and to survey the molecular mechanisms. A number of elegant studies have described valuable approaches for this (e.g., Nam et al., *Nature*, 2019; van Galen et al., *Cell*, 2019) and we plan to attempt to incorporate such approaches into the ReDeeM pipeline in the future.

We added these new data and discussions in the manuscript and Methods (Page 14 [421-427], 32 [968-973], 37 [1144-1165]).

RTR Table 1

aged-1	Not detected
aged-2	DNMT3A-R736H(11%);FLT3-V592I(13%)
aged-3	Not available
aged-4	Not Available
aged-5	ASXL1-Q373X(28.3%)

Referee #4 (Remarks to the Author):

In this work, Weng and colleagues improved an endogenous lineage tracing system by supporting mitochondrial variant enrichment and detection in a single-cell multiomics platform and applied it to study human hematopoiesis aiming to link cell states and lineages. Most of the analyses are technically solid and the paper is well written with results clearly presented.

A few key discoveries of this work include 1) HSC mtDNA clones showing functional differences and differentiation bias, where the fate bias is relatively small but can be sustainable for months; 2) HSC clonal diversity decreases with age, where positive expansion is observed with higher fitness. These biological findings are of potential interest. As not an expert in hematopoiesis, the reviewer will focus on technical/computational aspects.

R4-1

1. As far as I understand, this paper, for the first time, introduced mtDNA enrichment in the single-cell multi-omics platform. This is a technical innovation and has good application potential. However, mtDNA enrichment has been recently achieved in both ATAC level (mtscATAC, ref 29) and RNA level (MAESTER, ref 32) by some of the authors and colleagues. Therefore, it would be key to have a comparison to these technologies, in terms of mtDNA enrichment and mutation detection.

We thank the Reviewer for the helpful question. We added a new **Extended Data Figure 4** and a full section in **Methods** to discuss the following points (Page 5 [L141-142], 39-40[L1251-1322]).

All three approaches (ReDeeM, mtscATAC, MAESTER) are useful tools for single-cell mitochondrial sequence analysis. We will discuss the pros and cons among these methods from several aspects in principle first (bullet points below, **Extended Data Fig 4a**), and then we will also provide comparative analysis from our new experimental data.

- The source of sequence is important regarding the mutation discovery. ReDeeM and mtscATAC detect variants from mtDNA, while MAESTER utilizes mtRNA. In general, mtDNA and mtRNA have consistent variant representation. But since mtRNA capture depends on gene expression levels, the coverage from mtDNA is more even, and there are no RNA editing artifacts in mtDNA. Therefore, mtDNA can be more reliable in finding informative true positive variants. On the other hand, methods using mtRNA can be more broadly applied, since single-cell RNA-seq kits are more available and commonly used, so many applications will continue to employ this approach.

- The use of UMI or eUMI can substantially increase the mutation calling sensitivity (**Extended Data Figure 4**, simulation). Both MAESTER and ReDeeM applied UMI/eUMI in mutation calling, whereas mtscATAC does not apply UMI.
- Of note, ReDeeM uses eUMI to label a unique mtDNA molecule, while the MAESTER uses UMI to label a unique mtRNA molecule. In terms of methodology, labeling unique mtDNA molecules is less biased and more quantitative compared to mtRNA labeling. This distinction arises from the potential variability in transcription levels across different mitochondrial genomes. For instance, certain mitochondrial genomes may exhibit lower transcription activity, or their transcripts may be less stable compared to others within the same cell. This variability becomes particularly relevant when a mutation exerts functional effects on transcription processes.
- Cost-effective deep sequencing is critical for mutation calling. Both ReDeeM and MAESTER generate a specific mitochondrial library for sequencing (nearly 100% reads are mapped to mitochondrial reference in the sequencing library). But mtscATAC does not. Typically, in the mtscATAC library there are only ~10-15% reads covering the mitochondrial genome. Therefore, mtscATAC needs 10-fold more reads to achieve the same sequencing depth on mitochondria.
- The coupling with other -omic profiling is valuable. mtscATAC is ATAC only; MAESTER is RNA only; ReDeeM is coupled with both ATAC and RNA in the same cell. They are all useful, depending on the research goals.

We compared the ReDeeM and mtscATAC protocol using the young-1 CD34⁺ sample. We generate the "mtscATAC" data from the same donor using non-enriched ATAC library (From the same multi-ome assay for a fair comparison). As discussed above, the "mtscATAC" library only contains 10% reads that are mapped to the mitochondrial genome (New **Extended Data Fig 4b**). The median coverage is 14.3X in "mtscATAC" library and 51.7X in ReDeeM library (defined as # of unique mtDNA fragments that cover each position per cell). Using mgatk for mtscATAC data, we identified 311 mutations. In contrast, ReDeeM identified 4,831 confident mutations that are further validated by expected mutational signatures (**Fig. 1**).

We also added a comparison of the ReDeeM and MAESTER protocols using the Young-2 BMBC samples. We generate the "MAESTER" library using the cDNA from the same multi-ome assay for a fair comparison. As is shown in the New **Extended Data Fig 4e**, the reads distribution on the MAESTER library is biased in some regions. The average coverage in MAESTER is lower than ReDeeM. We took the top 307 mutations identified by MAESTER and the 4,087 mutations by ReDeeM. Both show expected mutational signatures. Although only a small fraction of the mutations identified by ReDeeM is well covered in MAESTER, they are consistent (78% of the well covered mutations are supported in MAESTER).

Taken together, ReDeeM has several practical and conceptual advances compared to previously described methods. Nonetheless, depending on the research goal, all these approaches are valuable and in some cases complementary.

R4-2

2. Related to the previous point, the multi-omics readout is a technical advance, which allows the authors to detect the variability of TF motif's frequencies (Fig. 2d & Extended Fig. 6g-i). On the other hand, such analysis can partly be achieved by computing a "gene set" score for each TF's target genes (e.g., from DoRothEA database) or by 5' based scRNA-seq probing TSS. Also, the scRNA+ATAC generally gives minimal contributions to cell type identification compared to scRNA only. Therefore, it is important to highlight if there are more benefits from multi-omics, as it costs a lot more on sequencing than scRNA-seq only (e.g., MAESTER system).

The benefit of using a multi-omic framework is two-fold: (1) More reliable and sensitive mtDNA mutation capture, and (2) More complete aspects to study clonal biases and fate decisions, such as epigenetic priming that may not be revealed at transcriptome level.

As discussed in the response to the Reviewer's point 1, the multi-ome framework of ReDeeM captures mitochondrial mutations from mtDNA which allows for more sensitive and accurate mutation detection, whereas MAESTER is from mtRNA (more details in the previous point, **New Extended Data Fig 4**).

The multi-ome framework allows us to survey the potential molecular drivers of clonal biases from more perspectives. In this revision, we developed a "clonal behavioral trajectory analysis" to identify differential gene expression or differential accessible chromatin peaks that may explain the different HSC clonal behaviors. Interestingly, this analysis suggests that the HSC clonal behavioral biases are likely encoded by epigenetic priming for genes that are not obviously differentially expressed at the moment (more details were discussed in the response to Reviewer 1 point 7, **New Extended Data Fig 12**, Revised manuscript **Page 12 [L355-365], 46-47 [L1534-1565]**). Although further validations would be needed to confirm these observations, it demonstrated the unique value of the epigenetic layer from the multiome framework to better understand molecular mechanisms driving stem cell function.

Nonetheless, we acknowledge that the multiome framework costs more. In situations where lower resolution is acceptable (e.g., to distinguish major subclones in a cancer sample) and solely gene expression analysis is the primary focus, alternative approaches such as MAESTER remain highly valuable.

R4-3

3. In the analysis, more than 4,000 variants (e.g., young-1) have been detected out of ~16.7KB mitochondrial genome. The number sounds a bit too high that one out of 4 genome positions has undergone somatic mutations. This is probably related to the setting that the variant calling pipeline aims for high selectivity by selecting all variants that are present in >1 and <all cells. However, it is not clear whether a more stringent variant calling will improve or harm the downstream analysis in both clonality reconstruction and progenitor estimation. The more

stringent variant calling can either be adding more criteria in their RedeemV pipeline or using alternative methods like mgatk/meagatk or MQuad.

The substantial number of mtDNA mutations (e.g., more than 4000 in young-1) is anticipated, because these mutations were observed across more than 10,000 cells, each containing hundreds to thousands of mtDNA copies, where the mutation rate is reasonably high. We have confidence in the validity of the mutations identified in ReDeeM for the following reasons:

- ReDeeM employed an eUMI-based consensus variant calling method, which effectively eliminates both PCR and sequencing errors, resulting in high sensitivity even for rare mutations. We demonstrated this principle by comparing it to conventional variant calling pipelines using a simulation analysis (**Extended Data Fig.3**). Furthermore, in a recent study, Duplex-seq, a highly accurate deep sequencing technology with robust error correction, was utilized to sequence bulk mitochondrial DNA across multiple tissues. The exceptional sensitivity of Duplex-seq enabled the authors to identify “a combined total of 77,017 mitochondrial SNVs”, nearly capturing every position (Sanchez-Contreras *et al.*, 2023). Although not directly comparable due to tissue and species differences, this observation aligns with the substantial number of mtDNA mutations detected by ReDeeM in our datasets.
- Mutational signatures with enrichment of C>T and T>C changes provide strong evidence for true mitochondrial mutation detection. To further support the validity of rare mutations, we stratified all mtDNA mutations based on the allele frequency level. We found that in all groups of the variants with different abundance, a clear mutational signature with enriched C>T and T>C mutations are observed, which support the validity of the mutations we detect, including rarer mutations with low VAF (**RTR Figure 5**).

In our response to Reviewer 2 (Conceptual point 1), we provided a more comprehensive discussion on a related issue on the validity of the mtDNA mutations, substantiated by additional supportive evidence and theoretical analysis. We have added a new **Extended Data Fig. 3** and revised the manuscript accordingly (**Page 5 [L129-131], 39 [L1236-1249]**).

Alternative methods such as mgatk use stringent criteria of variant filtering due to the challenge of distinguishing true mutations from PCR and sequencing errors, given the absence of an effective consensus-based strategy with UMIs. As a result, mgatk demonstrates limited detectability, with only 311 mtDNA mutations compared to 4,831 detected by ReDeeM. This limitation leads to low connectedness, as 43% of cells show zero detected variants and are unable to connect with other cells (**Fig 1e-f, Extended Data Fig. 4d**).

Nonetheless, we agree with the Reviewer’s suggestion on providing more criteria in the ReDeeM pipeline to allow adjusting stringency when needed. The ReDeeM-V pipeline allows for changing parameters to control the stringency of consensus variant calling, including the eUMI group size, consensus level, etc. The ReDeeM-R package allows for changing parameters of variant filtering based on the depth, number of cells carrying the mutation, etc. We added explanations to address this in the Methods section (**Page 38 [L1234-1235], 41 [L1334-1335]**).

R4-4

4. Here, the authors used the variants in a binarization way, i.e., either present or absent. Such a setting is probably reasonable given the coverage is even (compared to MEASTER and mtscATAC). However, the allele frequency in each cell can be informative for indicating the kinship between cells, hence it is worth a comparison using the allele frequency directly.

As another related note, although the analysis here focuses on “new” mutations for cell lineage analysis, it remains unclear how informative the allele frequency stochasticity is for lineage relatedness, especially those homoplasmic variants (i.e., the variants present in most or all cells).

We agree with the Reviewer on the potential value of considering allele frequency (or heteroplasmy level) in our datasets. However, we have several concerns in incorporating allele frequencies into our lineage reconstruction pipeline. We provide a comprehensive discussion in the response to Reviewer1-point1 on this issue and added an explanation in Methods (Page 41 [L1341-1343]).

Regarding the frequency stochasticity for homoplasmic variants, we specifically analyzed all mtDNA mutations that are present in more than 75% of cells. By this definition, we identified 32 homoplasmic variants in the young-1-HSPC dataset (**RTR Fig. 1a**). Interestingly, the stochastic levels of these variants are extremely low (IQR=0 for all 32 variants). To conclude, we think the homoplasmic variants are generally not informative for lineage relatedness in normal human hematopoiesis. But further investigations are needed for other tissues and diseases.

A few minor comments to consider:

R4-minors

5. In the paper, it says that the mtDNA enrichment has been achieved by using ~142 probes (120bp each) but it is unclear whether the enriched mtDNA is from the RNA layer, ATAC layer or the DNA layer. It will be helpful to write it clearly.

mtDNA is enriched from the ATAC layer, which is DNA (New **Extended Data Fig 4a**). We added clarifications to the Methods section (Page 40 [L1273]).

6. The eUMI is a nice strategy to remove PCR duplicates (or even fragmentation duplicates). However, this strategy may largely be similar to the commonly used strategy in MarkDuplicates (Picard) and also the MAESTER analysis pipeline (ref 32). Though the implementation for support droplet-based single-cell data is valuable, it would still be good to mention the relevant tools and possibly highlight the difference.

The eUMI makes use of all PCR duplicate reads for consensus analysis rather than simply removing duplicates. All reads amplified from the same fragment are grouped and only the mutations that are present with great consensus among these reads within the same group are considered as true mutations. In contrast, MarkDuplicates basically removes the PCR duplicates

by taking one reads for the downstream analysis. The consensus strategy by eUMI in ReDeeM removes errors during library prep and sequencing at single molecule level which substantially boosts the mutation detectability and accuracy. Similar strategies have been successfully applied in bulk studies where extreme rare variant detection is needed (e.g., Abascal et al., *Nature*, 2021).

We compared ReDeeM and MAESTER systematically as suggested. More details in the response to the Reviewer's point 1.

We highlighted these differences in the Methods section in the revised manuscript (**Extended Data Fig. 3, 4, Page 40 [L1279-1289]**).

7. The accuracy of eUMI is assessed by a simulated, where it is a bit unclear if this is based on an assumption of uniform distribution for the cutting sites. However, the uniformness may not broadly hold, hence it will be good to check the distribution of the cutting sites from data with real UMIs.

This is a great point. Yes, our simulation is based on an assumption of uniform distribution. However, the real Tn5 cutting site indeed is not totally uniform but has some preference. We agree that using the real empirical cutting site distribution is ideal for this simulation. We checked the total cutting sites from one of our dataset (in total 190,629,294 cutting sites), which was used as the empirical distribution for random sampling in our revised simulation. The simulated result is largely consistent with what we had seen previously, with an expected increase in simulated conflict rate (**RTR Fig. 11**). The range of mtDNA copies per cell in hematopoiesis is about 300 to 1000. The simulated conflict rate is ~3-5%. It is important to highlight that in eUMI consensus variant calling, the issue of collisions primarily results in false negatives. This means that when two distinct molecules are mistakenly considered as one, or placed within the same eUMI group, a mutation may not be accurately detected due to the resulting "poor consensus". Therefore, the ~3-5% of false negatives will have limited impact on downstream analysis and conclusions. We have revised the **Extended Data Fig 2a**, and the manuscript (**Page 5 [L127], 37-38 [L1200-1201, 1207-1210]**).

RTR Figure 11. eUMI collision simulation (Tn5 cutting site biases is modeled in)

8. For the TF frequency (Fig. 2d, right panel), it is not very clear how it has been performed. Adding some extra details to the methods section will help.

The single cell motif analysis was performed using ChromVar using the JASPAR human transcription factor (TF) database. The deviation of the TF frequency was computed for each cell for the downstream analysis including the visualization in **Fig 2d**. We added details on this in the revised Methods and legends (Page 25 [L724-725], 38 [L1178-1181]).

9. For Fig. 2g (and also text on page 7), why is a relatively high cutoff on $FDR < 0.2$ chosen (though fold change > 2)? Is it because many clades are relatively small? If so, how about using a larger clade size (e.g., with lower resolution in tree cut) and will it affect the analysis?

The Reviewer is correct. In this analysis, we wanted to minimize the false negatives for many small clades that are likely enriched by certain cell types but have limited power. To ensure fairness, we thoroughly tested and reported all clades of various sizes. Notably, there are many larger clades that also exhibit significant enrichment. We provided the complete information of test statistics for all clades with detailed information of clade size, enrichment fold change, original p-value, and FDR. Different filtering or selection thresholds can be applied by the readers to assess how this impacts the findings (**Supplementary Data 3**).

10. In Fig. 2e, it's very interesting to see lower mutation burdens in HSC. Adding some discussion on the possible relation between cell differentiation and division (or the potential reason) can be very valuable.

The HSC is considered to be primarily quiescent in adult bone marrow and thought to only divide approximately once a year (Catlin *et al.*, 2011). In contrast, the committed progenitors divide very actively to amplify and generate a large enough population for maintaining effective blood cell production. Our interpretation of the data is that the increased mutations are likely accumulated due to the rapid proliferation in these progenitors and propagated to their more differentiated progenies.

We added more discussions in the revised manuscript (Page 7 [L200-201]).

11. Fig. 3f, it is unclear what statistical method is used for performing the enrichment analysis. Please write it in the text or figure caption.

The hypergeometric test was used. We added in the revised manuscript and legends (Page 26 [L739]).

RTR References

- Arbeithuber, B. *et al.* (2020) 'Age-related accumulation of de novo mitochondrial mutations in mammalian oocytes and somatic tissues', *PLoS biology*, 18(7), p. e3000745.
- Brown, W.M., George, M., Jr and Wilson, A.C. (1979) 'Rapid evolution of animal mitochondrial DNA', *Proceedings of the National Academy of Sciences of the United States of America*, 76(4), pp. 1967–1971.
- Catlin, S.N. *et al.* (2011) 'The replication rate of human hematopoietic stem cells in vivo', *Blood*, 117(17), pp. 4460–4466.
- Durham, S.E. *et al.* (2007) 'Normal levels of wild-type mitochondrial DNA maintain cytochrome c oxidase activity for two pathogenic mitochondrial DNA mutations but not for m.3243AG', *American journal of human genetics*, 81(1), pp. 189–195.
- Forsberg, L.A. *et al.* (2014) 'Mosaic loss of chromosome Y in peripheral blood is associated with shorter survival and higher risk of cancer', *Nature genetics*, 46(6), pp. 624–628.
- Fujino, T. *et al.* (2021) 'Mutant ASXL1 induces age-related expansion of phenotypic hematopoietic stem cells through activation of Akt/mTOR pathway', *Nature communications*, 12(1), p. 1826.
- Huang, W. *et al.* (2012) 'ART: a next-generation sequencing read simulator', *Bioinformatics*, 28(4), pp. 593–594.
- Lareau, C.A. *et al.* (2022) 'Single-cell multi-omics reveals dynamics of purifying selection of pathogenic mitochondrial DNA across human immune cells', *bioRxiv*. Available at: <https://doi.org/10.1101/2022.11.20.517242>.
- Nagase, R. *et al.* (2018) 'Expression of mutant Asxl1 perturbs hematopoiesis and promotes susceptibility to leukemic transformation', *The Journal of experimental medicine*, 215(6), pp. 1729–1747.
- Rausser, S. *et al.* (2021) 'Mitochondrial phenotypes in purified human immune cell subtypes and cell mixtures', *eLife*, 10. Available at: <https://doi.org/10.7554/eLife.70899>.
- Sanchez-Contreras, M. *et al.* (2023) 'The multi-tissue landscape of somatic mtDNA mutations indicates tissue-specific accumulation and removal in aging', *eLife*, 12. Available at: <https://doi.org/10.7554/eLife.83395>.
- Stewart, J.B. and Chinnery, P.F. (2015) 'The dynamics of mitochondrial DNA heteroplasmy: implications for human health and disease', *Nature reviews. Genetics*, 16(9), pp. 530–542.
- Walker, M.A. *et al.* (2020) 'Purifying Selection against Pathogenic Mitochondrial DNA in Human T Cells', *The New England journal of medicine*, 383(16), pp. 1556–1563.
- Williams, S.L. *et al.* (2013) 'Somatic mtDNA mutation spectra in the aging human putamen', *PLoS genetics*, 9(12), p. e1003990.
- Wolf, D.P., Hayama, T. and Mitalipov, S. (2017) 'Mitochondrial genome inheritance and replacement in the human germline', *The EMBO journal*, 36(15), pp. 2177–2181.

Xu, S. *et al.* (2012) 'High mutation rates in the mitochondrial genomes of *Daphnia pulex*', *Molecular biology and evolution*, 29(2), pp. 763–769.

Yakes, F.M. and Van Houten, B. (1997) 'Mitochondrial DNA damage is more extensive and persists longer than nuclear DNA damage in human cells following oxidative stress', *Proceedings of the National Academy of Sciences of the United States of America*, 94(2), pp. 514–519.

Ye, K. *et al.* (2014) 'Extensive pathogenicity of mitochondrial heteroplasmy in healthy human individuals', *Proceedings of the National Academy of Sciences of the United States of America*, 111(29), pp. 10654–10659.

Yuan, Y. *et al.* (2020) 'Comprehensive molecular characterization of mitochondrial genomes in human cancers', *Nature genetics*, 52(3), pp. 342–352.

Zaidi, A.A. *et al.* (2019) 'Bottleneck and selection in the germline and maternal age influence transmission of mitochondrial DNA in human pedigrees', *Proceedings of the National Academy of Sciences of the United States of America*, 116(50), pp. 25172–25178.

Zhang, H., Burr, S.P. and Chinnery, P.F. (2018) 'The mitochondrial DNA genetic bottleneck: inheritance and beyond', *Essays in biochemistry*, 62(3), pp. 225–234.

Reviewer Reports on the First Revision:

Referees' comments:

Referee #1 (Remarks to the Author):

The revised version of the manuscript by Sankaran, Weissmann and colleagues has incorporated additional data, additional controls, and carefully reworded statements where significant alternative interpretations are considered. They also include important acknowledgment of limitations, which are always necessary in studies implementing new technologies at the edge of what's possible. I have read the author's replies to my concerns and those raised by other referees and I have concluded that the majority of these concerns were successfully addressed. Of course the methodology is imperfect in some cases (e.g. clonal assignment is good, but not great, LOY+ lineage clustering is spotty) and the data could always be a bit better (e.g. lack of sufficient cells in the aged dataset to make any significant conclusion about HSC clonal programs), but I remain convinced about two things I believe are the most important: 1) the increased power of this new method and 2) its validation regarding HSC clonal expansions with age. I stress here that recognizing the study limitations will always push others to further improve and solve the challenges of these technologies, which in the end is what keeps pushing the field forward. In that sense, the authors are now much more clear about how they estimated the agreement between the CRISPR and RedeeM methods (AOC and positive AOC). Still, I think it is critical to carefully interpret how good the median Rand index of 0.69 actually is. For that, it is important to know what would be the result of a fully random grouping of cells into clones (i.e. reporting the adjusted Rand Index). Also, it would be important to calculate this number across several points of the CRISPR lineage tree. In any case, while I believe the method is good, it is evident to me it still makes a few errors in assigning lineages, which can lead to problems across several analyses (beyond the other challenges I mentioned, which are now being fully acknowledged by the authors). Still, with all its imperfections at the forefront of this field, it is obvious that the signal/noise is good enough to extract quite some interesting novel insights, some of which make perfect sense even if they aren't followed up (e.g. the lymphoid TF accessibility program in Ly-biased clones). In sum, I believe that this will be a breakthrough technology that many will use, many will attempt to challenge, and some will hopefully surpass. And I think this is exactly what makes this an interesting study that everyone should read about.

—Alejo E. Rodriguez-Fraticelli

Referee #2 (Remarks to the Author):

While some of my original doubts have been clarified, the central claim that ReDeeM can delineate "Cell genealogies" (or even, that it can confidently identify clones) still lacks independent validation. The LoY experiment, as the authors convincingly argue, cannot be used as a validation for biological reasons; the CRISPR experiment provides evidence of some statistical association between ReDeeM "clusters" and CRISPR clones, but also indicates that there are massive differences between the two (see detail below). Further, as the authors state, it is not an ideal validation because "the generation of indels and mtDNA are not necessarily coupled" in this model, and the life span of the mice is small.

Hence, there is currently no hard validation in the manuscript that supports the central claim, that ReDeeM can "decipher cell genealogies". A proper validation of the claim "genealogies" would require, in my opinion, WGS of single cell derived colonies (see detail below). A minimal validation of the statement that ReDeeM can identify clones would be to include genotyping of some mutation driving a clonal expansion, and demonstrate a high overlap between the clonal marker

and a ReDeeM clade. For examples, the ASXL1 mutation in individual age-5 could be used, or if that's technically difficult, CALR in MPN is an extremely highly expressed gene that can readily be genotyped from the RNA-seq libraries (see Nam et al., 2019). This would, however, not validate the ability to delineate genealogies, and it would correspondingly decrease the ground-breaking nature of the manuscript (although it would then be a valuable tool to identify clones in human hematopoiesis, going beyond recent work by van Galen, Lareau and others).

As of now, the manuscript reports several statistically weak (q values on the order of 0.01 to 0.1) associations of putative mitochondria-defined clones and a) genotypes (CRISPR, LoY) or b) function. Without validation of the ReDeeM's ability to track clones I am worried that these cannot be trusted, as weakly significant associations in big data sets can appear for a multitude of reasons (e.g. confounders affecting both readouts, overdispersion, data not perfectly matching the assumption of the statistical test used, etc.).

Detail on Original Point 2 (CRISPR validation): The analysis is not convincing. While it is now clearer how agreement of closeness is defined, this measure just demonstrates some enrichment of mitochondrial mutations in a given clone, and not an ability of ReDeeM to identify clones. I still lack direct comparisons between the ReDeeM clones and the CRISPR clones. The authors chose not to show it, because of "a lack of clarity" (does this simply mean the match is too bad? These data in my opinion must be shown.). Instead they show rand indeces between ReDeem clones and CRISPR clones, but the reported Rand indeces (~ 0.64) correspond to random partitioning (the correct function to use would be `adj.rand.index` and not `rand.index`, here `rand.index` was used, see line 1421).

Example R code demonstrating that random partitions give large rand indeces:

```
> a <- sample(0:6,100, T)
> b <- sample(0:6,100, T)
> rand.index(a,b)
[1] 0.7515152
> adj.rand.index(a,b)
[1] 0.08548696
```

So overall, the conclusion here seems to be that clones defined by CRISPR and clones defined by ReDeeM are quite different (Rand index analysis in Figure EV6c), although mitochondrial variants are somewhat enriched in CRISPR clones (AOC analysis in figure EV5+6).

I do understand the argument that "the generation of indels and mtDNA are not necessarily coupled" in this model. I think it would be OK if this control experiment is discussed much more carefully and a different control is used to provide gold standard validation.

Detail on Original Point 3 (LoY): I follow the arguments by the authors, but if LoY is not giving rise to clonal expansion here, this experiment is not suited as a gold standard validation of ReDeeM.

Detail on Original Point 5 (Validation): Fundamentally not addressed. The ReDeeM method needs a gold standard validation to sustain the central claim that it can be used to identify cellular genealogies (and clones). Currently only WGS of HSC derived colonies can provide such validation. If standard WGS lacks sensitivity, a WGS method with UMIs (or duplex sequencing?) needs to be used. Alternatively, they could split the colonies in two parts, perform standard WGS on half of the colony, Hash the other half with a barcoded surface antibody and perform ReDeeM. Genealogies from colony WGS (mitochondrial mutations, if possible), colony WGS (nuclear mutations) and single cells / ReDeeM (mitochondrial mutations) need to be compared.

Additional question: I noticed that in figure 5c, the expanded clades have quite long distances in the dendrograms. Is this just a consequence of the display method? Or does it mean that these cells are more dissimilar within a clade, compared to cells from the smaller clades observed in the young individuals? If the latter, would it be appropriate to refer to them as an "expanded clade",

or rather as “residual cells not part of any clone”? I apologize for overlooking this before.

Referee #3 (Remarks to the Author):

The authors have responded adequately to my comments and suggestions by performing additional experiments and analyses. They have also performed substantial additional work and analyses in response to the comments of other reviewers. Overall, their revised and improved manuscript would be of substantial interest to the field.

One minor comment is that many would not consider the FLT3-V592I to be a driver of CHIP. The variant has been reported in some cancers, but it is not recurrent in CHIP. The authors may want to clarify this.

Referee #4 (Remarks to the Author):

The authors have addressed all my previous comments well.

I saw the authors have released the scRNA and scATAC data on figshare and raw reads on GEO, which are very good. On top of this, I would strongly suggest the authors release more intermediate data, especially the mitochondrial matrices of reference and alternative alleles and the called variants list. If the authors can share the analysis notebooks, it will also be very helpful to readers. I believe this will better ensure the reproducibility and more accessible re-analysis of the data from this new technology.

Additionally, for the redeemR package, please provide some demonstration vignettes. It is very hard to start using it.

Some minor comments:

- line 1352 (p42): the equation didn't show properly.
- Fig. 3: panel d and e are not matched with the order in the captions.
- Fig. 4b/c, the output activities of T1 and T2 are relative values, right? Just wonder if the author may have an estimation of the total cells in each time point or if it is assumed to have a similar-sized population.

Referee #5 (Remarks to the Author):

As requested by the Editor, I have limited my comments to the mtDNA analysis, and pass no comment on the significance of the findings for our understanding of hematopoiesis.

The techniques presented are an evolution of published methods that have emerged from the Sankaran lab in the last 3 years – of note, the median mtDNA coverage of ~50x resembles coverage using a less complex protocols PMID: 37386249, PMID: 36792778, PMID: 35210612

The data raise a number of questions which are important to resolve if we are to have confidence in the biological conclusions about hematopoiesis:

Q1. The eUMI collision rate seems high at 3% - is this 'random' collision or does it imply the biological relatedness to be expected? ie were the collisions more common in related cell types, are they due to mutation hot-spots? Were there number of collisions in the same sequencing read exactly as predicted by random mutations? All of these would influence whether we can use

mtDNA mutations and inert/neutral lineage tracers.

Q2. To what extent are the inferred dynamics of clonality really a reflection of the mtDNA dynamics in different cell types/lineages (eg mtDNA turnover leading to drift in heteroplasmy levels), as opposed to cell-type dynamics which is what the authors conclude. This could be addressed by comparing homoplasmic to heteroplasmic variants. If mtDNA dynamics are not important, both should yield the same results.

Q3. To what extent do the mtDNA variants drive the behaviour of the clones? This is an important analysis to do given the rich dataset. For example, have the authors compared silent vs non-silent variants? Or variants in sites known to influence mtDNA replication. Again, if some mtDNA variants do influence the frequency of detecting mutations (eg some tend to be propagated, some selected against – which I would expect), then we cannot consider them neutral tags and will need to incorporate this into the interpretation.

As an aside, it does not surprise me that the mutation rate in stem cells is low compared to their progeny. Most of the mutations are likely to be polymerase errors, the burden reflects the amount of mtDNA replication and cell turnover which is less in HSC than the cascade of proliferating progeny.

Response to Reviewer Comments:

We are grateful to the Reviewers for their valuable and supportive comments on our work. The remaining questions and suggestions have been very helpful in guiding our revisions to further strengthen the major conclusions and make the work more accessible.

Our manuscript presents the ReDeeM technique and demonstrates its ability to provide a unique view of human hematopoiesis both under normal conditions and during aging. We completely agree with the Reviewers on the importance of validating ReDeeM's ability to decipher genealogies, track clones, and illuminate significant biological insights. Any new technology that provides unprecedented information as ReDeeM does necessitates multiple lines of validation, as no single gold standard exists (more details in point-by-point discussions). In our revised manuscript, we added, strengthened, and highlighted several lines of evidence, including analyses with whole genome sequencing (WGS), which strongly support and validate the major biological conclusions enabled by ReDeeM.

The valuable perspectives on mitochondrial inheritance from Reviewer #5 have facilitated further validation, clarification, and yielded new insights.

We start by highlighting the major changes and modifications to the manuscript. Additionally, we address each of the comments in depth in a point-by-point manner below. Our responses are in blue font, while the original comments are in black font. In addition, all changes in the text are highlighted in blue font.

- Additional analyses of the dual-lineage tracer experiment (CRISPR-based+ReDeeM) provide further validations of ReDeeM. We have computed the Adjusted Rand Index at various clonal resolutions for all 10 tumors and illustrated examples of direct visual comparisons of clones. Both analyses significantly support the agreement between ReDeeM and CRISPR-based lineage tracing (see R2-3, revised **Extended Data Fig. 6c-d**). This demonstrates the ability of ReDeeM to provide valuable lineage tracing information at the subclonal level on a short timescale which is technically challenging because of the shorter time period (a few months) for mutations to accumulate compared to the years or even decades that HSCs persist.
- We have included additional genealogical and clonal tracing validation by carefully analyzing recently published whole genome sequencing (WGS) data from single colonies derived from myeloproliferative neoplasms (MPN) patients with JAK2 V617F somatic mutations (Van Egeren et al. 2021). The mitochondrial mutations that we confidently detect from WGS show great consistency with the nuclear-mutation-based phylogenetic tree on both clonal and subclonal structures. We also demonstrate the substantially enhanced ability to detect mitochondrial mutations by ReDeeM compared to WGS with this empirical data (see R2-5, new **Extended Data Fig. 3, 15**).
- We have enumerated and highlighted multiple independent lines of evidence that support the robustness of ReDeeM for genealogical and (sub)clonal tracking in the revised manuscript. These include, but are not limited to single colony WGS, the ability to

reconstruct known blood lineages, the detection of clonal expansion with relevant phenotypes, among others (see R2-1 and **Methods**).

- We have provided comprehensive analysis on mitochondrial mutation dynamics and the potential functional impact of mtDNA mutations, reinforcing their overall neutrality and suitability as inert lineage tracers. (R5, new **Extended Data Fig 16**).

We sincerely apologize for the delay on your manuscript, "Deciphering cell states and genealogies of human hematopoiesis with single-cell multi-omics." Your work has now been seen by 5 referees, whose comments are attached below. While reviewers #1, and 4 appear for the most part satisfied with the current draft, Reviewer #2 raises (re-raises) important concerns. As explained in our previous email, we contacted Reviewer #3 to comment upon Reviewer #2's concerns, who agreed with the importance of conducting WGS. At the same time, the editorial team had a concern whether expertise regarding mitochondrial inheritance was sufficiently covered, and Reviewer #5 was recruited to comment specifically upon this aspect of your work; their comments are also provided below. The remaining concerns of the reviewers must be addressed before we can consider publication in Nature.

Should further experiments allow you to address these criticisms, we would be happy to consider a revised manuscript (unless something similar has been accepted at Nature or appeared elsewhere in the meantime). We hope to receive your revised paper within four to six months. If you cannot complete the required revisions within this time frame, please let us know when you would anticipate being able to submit a revised manuscript. All reviewer concerns should be addressed in your future point-by-point response and revised manuscript, particularly,

- Report adjusted Rand index (reviewers #1, #2)
- Include additional genealogical validation (WGS) (Reviewers #2, #3)
- Include additional validation of clonal tracking (Reviewer #2)
- Include analysis requested by Reviewer #5

Referees' comments:

Referee #1 (Remarks to the Author):

The revised version of the manuscript by Sankaran, Weissmann and colleagues has incorporated additional data, additional controls, and carefully reworded statements where significant alternative interpretations are considered. They also include important acknowledgment of limitations, which are always necessary in studies implementing new technologies at the edge of what's possible. I have read the author's replies to my concerns and those raised by other referees and I have concluded that the majority of these concerns were successfully addressed. Of course the methodology is imperfect in some cases (e.g. clonal assignment is good, but not great, LOY+

lineage clustering is spotty) and the data could always be a bit better (e.g. lack of sufficient cells in the aged dataset to make any significant conclusion about HSC clonal programs), but I remain convinced about two things I believe are the most important: 1) the increased power of this new method and 2) its validation regarding HSC clonal expansions with age. I stress here that recognizing the study limitations will always push others to further improve and solve the challenges of these technologies, which in the end is what keeps pushing the field forward. In that sense, the authors are now much more clear about how they estimated the agreement between the CRISPR and Redeem methods (AOC and positive AOC). Still, I think it is critical to carefully interpret how good the median Rand index of 0.69 actually is. For that, it is important to know what would be the result of a fully random grouping of cells into clones (i.e. reporting the adjusted Rand Index). Also, it would be important to calculate this number across several points of the CRISPR lineage tree. In any case, while I believe the method is good, it is evident to me it still makes a few errors in assigning lineages, which can lead to problems across several analyses (beyond the other challenges I mentioned, which are now being fully acknowledged by the authors). Still, with all its imperfections at the forefront of this field, it is obvious that the signal/noise is good enough to extract quite some interesting novel insights, some of which make perfect sense even if they aren't followed up (e.g. the lymphoid TF accessibility program in Ly-biased clones). In sum, I believe that this will be a breakthrough technology that many will use, many will attempt to challenge, and some will hopefully surpass. And I think this is exactly what makes this an interesting study that everyone should read about.

—Alejo E. Rodriguez-Fraticelli

We thank the Reviewer for the helpful comments and suggestions, which have greatly improved our manuscript. Below is our response to the remaining questions.

We agree with the Reviewer that the Adjusted Rand Index (ARI) is a more interpretable metric. We have therefore computed the ARI for all 10 independent tumors (revised **Extended Data Fig. 6c-e**). To reflect different points on the tree, we used multiple different clonal clustering resolutions for each tumor to compute the ARI (each dot on the boxplot is a different resolution). As expected, the ARI suggests a good consistency (ARI ranges between 0.2 and 0.7), although it varies across different tumors and is dependent on the clustering resolution employed. On the other hand, we wish to point out that AOC is a single-cell resolution metric that does not require an arbitrary clustering parameter, and thus provides a fairer and clearer view on how the neighbors are supported across different points on the tree (**Extended Data Fig. 5f, 6a**). We present both the AOC and the ARI metrics in the revised manuscript, which we hope together can provide comprehensive and informative assessments about the Redeem's performance (**Page 6 [L165-168], 45 [L1451-1455]**).

Referee #2 (Remarks to the Author):

R2-1

While some of my original doubts have been clarified, the central claim that ReDeeM can delineate “Cell genealogies” (or even, that it can confidently identify clones) still lacks independent validation. The LoY experiment, as the authors convincingly argue, cannot be used as a validation for biological reasons; the CRISPR experiment provides evidence of some statistical association between ReDeeM “clusters” and CRISPR clones, but also indicates that there are massive differences between the two (see detail below). Further, as the authors state, it is not an ideal validation because “the generation of indels and mtDNA are not necessarily coupled” in this model, and the life span of the mice is small.

Hence, there is currently no hard validation in the manuscript that supports the central claim, that ReDeeM can “decipher cell genealogies”. A proper validation of the claim “genealogies” would require, in my opinion, WGS of single cell derived colonies (see detail below). A minimal validation of the statement that ReDeeM can identify clones would be to include genotyping of some mutation driving a clonal expansion, and demonstrate a high overlap between the clonal marker and a ReDeeM clade. For examples, the ASXL1 mutation in individual age-5 could be used, or if that’s technically difficult, CALR in MPN is an extremely highly expressed gene that can readily be genotyped from the RNA-seq libraries (see Nam et al., 2019). This would, however, not validate the ability to delineate genealogies, and it would correspondingly decrease the ground-breaking nature of the manuscript (although it would then be a valuable tool to identify clones in human hematopoiesis, going beyond recent work by van Galen, Lareau and others).

We appreciate the Reviewer's emphasis on ensuring we have maximally validated ReDeeM. These suggestions have motivated us to present further evidence and underscore the multifaceted lines of evidence that demonstrate ReDeeM’s performance. First, we wish to emphasize that the ultimate goal of validation is to support whether the improved capability of ReDeeM is sufficient to extract robust and novel biological conclusions. Therefore, both technical benchmarking and biological validation are equally important. Moreover, given the limitations inherent to existing validation approaches (see R2-5), we advocate for the collective consideration of multiple lines of evidence. In the revised manuscript, we enumerated multiple lines of independent evidence, including WGS analyses and multiple other supporting elements. All these supporting elements are important and together substantiate the claim that ReDeeM can decipher genealogies, track clones, and illuminate biological insights. This includes:

1. Reconstruction of the major known hematopoietic lineages (deciphering genealogies): The main point of **Fig 2** and **Extended Data Fig. 8** is specifically to demonstrate the capability of deciphering genealogies, where we can use mitochondrial DNA mutations to largely reconstruct the expected pattern seen in hematopoiesis, which requires subclonal level resolution. This represents a remarkable success of the ReDeeM approach, as pointed out by Reviewer #1 and others.

2. Detecting clonal expansions (clonal and subclonal tracking): With the expanded cohort, we show that we can detect oligoclonal structural alterations in all aged individuals, but none of the younger individuals, which is in strong agreement with our emerging understanding of the high prevalence of clonal hematopoiesis with aging including the very recent work from Campbell and colleagues (Mitchell et al. 2022). Critically, our work also observes output biases in expanded clones. These findings not only validate the capability of clonal and subclonal tracking, but also go beyond what was possible in the single colony WGS studies, which lose all information about cell state.
3. Phenotypic recapitulation of ASXL1 mutant clonal hematopoiesis of indeterminate potential [CHIP] (clonal tracking): The erythroid biases in expanded clones revealed by ReDeeM are consistent with phenotypic measurement in Asxl1 mutant mouse models (**Extended Data Fig. 14j**, (Nagase et al. 2018; Fujino et al. 2021). This provides further validation at a functional level.
4. The robust HSC-progeny assignment from different time points (clonal tracking and deciphering genealogies): The HSC-progeny assignment between differentiated blood cells, progenitors, and HSCs is highly reproducible and reveals sustained clonal specific state- and behavior biases over time, which not only supports the ability of deciphering genealogies but also provides new and previously unachievable insights into HSC biology (**Fig. 3-4, Extended Data Fig. 11**).
5. Comparisons between CRISPR-based lineage tracers versus ReDeeM (deciphering genealogies): This represents the most stringent test of the ability of ReDeeM to report on genealogies that unfold subclonal evolution over a short timescale of a few months starting from one single cell. We provide validation on two independent mouse batches involving 10 tumors. Statistically robust agreement between CRISPR- and ReDeeM-based tracing is shown, evidenced quantitatively by single-cell level metrics (AOC), and cluster-level Adjusted Rand Indices (ARI) (more discussions in R2-3).
6. Comparison between single colony WGS-based lineage tracing inferred by nuclear genome mutations with mitochondrial DNA mutations (clonal and subclonal tracking): We analyzed WGS data of 42 single colonies from an MPN patient with a known JAK2 V617F somatic driver mutation. We have seen great consistency between confidently called mitochondrial mutations and the nuclear genome mutation-based tree at both the clonal and subclonal levels. Of note, we also demonstrate the substantially advanced detectability of true mitochondrial DNA mutations in ReDeeM in comparison to WGS, especially for lower heteroplasmic variants (more discussions in R2-5), empirically with this analysis.

Specifically, for the concerns raised regarding clonal tracking, we believe that the evidence discussed above has provided substantial additional support. We agree that it would be nice to show that the ASXL1 genotype can be detected reliably in single cells. Unfortunately, the ASXL1 mutant site cannot be detected in our single cell library, as its expression level is low and the Q373X mutation is located 5,590bp away from the 3'-end where it is not covered in the 10X

Genomics 3'-based sequencing library. This is a challenging but valuable open problem to be solved in the future in general. Alternatively, we believe the phenotypic agreement we observed in this donor with the ASXL1 mutation provides strong evidence at a functional level, since both expanded clades are significantly depleted in erythroid cells, which echoes the reported functional effect of Asxl1 mutant in mouse models (**Extended Data Fig. 14j**) (Nagase et al. 2018; Fujino et al. 2021). Furthermore, the WGS data (more detailed discussion in R2-5) have shown that the expanded JAK2 mutant clone is strongly supported by multiple confident mitochondrial mutations which serve as a clear validation of clonal tracking.

We also wish to point out that we and others have published extensive characterization of clonal tracking with mitochondrial mutations previously. For example, we validated the clonal tracking ability by employing "ground truth" approaches using experimental lineage trees and using lentivirus-based static barcoding (Fig. 1-2, Ludwig et al. 2019; Fig. 3, Lareau et al. 2021). Moreover, we and others have published additional evidence supporting the validity of clonal tracking, including comparisons with somatic mutations present in various diseases in bulk and at the single-cell level (Xu et al. 2019; Miller et al. 2022; Velten et al. 2021; Myers et al. 2022; Lareau et al. 2023).

Altogether, our additional analyses and the additional data we have highlighted serve as a systematic framework that provides strong validation of ReDeeM. We have modified the manuscript to provide additional discussion on these points, as well (**Page 49-50 [L1606-1661]**).

R2-2

As of now, the manuscript reports several statistically weak (q values on the order of 0.01 to 0.1) associations of putative mitochondria-defined clones and a) genotypes (CRISPR, LoY) or b) function. Without validation of the ReDeeM's ability to track clones I am worried that these cannot be trusted, as weakly significant associations in big data sets can appear for a multitude of reasons (e.g. confounders affecting both readouts, overdispersion, data not perfectly matching the assumption of the statistical test used, etc.).

We agree about the importance of interpreting statistical significance carefully. Indeed, we maintain a cautious approach in formulating biological conclusions, ensuring that they are drawn only when observations are reproducible across multiple independent replicates with consistent trends and statistical significance. This principle is applied in general throughout our manuscript, particularly in relation to the major findings reported. We are confident that the reproducibility we observed across our multiple findings serves as strong support for both our technology and the underlying biological insights revealed. This includes, but is not limited to the following:

1. We reported that ReDeeM reconstructs the major known hematopoietic lineages using mitochondrial DNA mutations, which validate ReDeeM's ability to decipher genealogies and provide new insights for cell-type origins. This was based on

reproducible observations in two independent human donors (**Fig. 2** and **Extended Data Fig. 8**).

2. To study HSC clonal-specific cell-state and behavior preferences, we sampled bone marrows from the same donor twice independently across four months. The reproducibility across two time points serves as the foundation for our analysis. The functional and behavior level preferences are only reported when the significant enrichments are observed across both time points (**Fig. 3-4**).
3. The consistency between ReDeeM and CRISPR-based lineage tracing is independently validated in 10 tumors from two batches of mice, reinforcing the robustness (**Extended Data Fig. 5-6**).
4. The clonal structure alterations in aging are reproducibly observed in 5 independent aged donors (**Fig. 5, Extended Data Fig. 14**).

In summary, we have great confidence in the biological insights we have presented, as they are consistently and significantly substantiated by a multitude of independent technical and biological observations.

R2-3

Detail on Original Point 2 (CRISPR validation): The analysis is not convincing. While it is now clearer how agreement of closeness is defined, this measure just demonstrates some enrichment of mitochondrial mutations in a given clone, and not an ability of ReDeeM to identify clones. I still lack direct comparisons between the ReDeeM clones and the CRISPR clones. The authors chose not to show it, because of “a lack of clarity” (does this simply mean the match is too bad? These data in my opinion must be shown.). Instead they show rand indeces between ReDeem clones and CRISPR clones, but the reported Rand indeces (~0.64) correspond to random partitioning (the correct function to use would be `adj.rand.index` and not `rand.index`, here `rand.index` was used, see line 1421).

Example R code demonstrating that random partitions give large rand indeces:

```
> a <- sample(0:6,100, T)
> b <- sample(0:6,100, T)
> rand.index(a,b)
[1] 0.7515152
> adj.rand.index(a,b)
[1] 0.08548696
```

So overall, the conclusion here seems to be that clones defined by CRISPR and clones defined by ReDeeM are quite different (Rand index analysis in Figure EV6c), although mitochondrial variants are somewhat enriched in CRISPR clones (AOC analysis in figure EV5+6).

I do understand the argument that “the generation of indels and mtDNA are not necessarily coupled” in this model. I think it would be OK if this control experiment is discussed much more carefully and a different control is used to provide gold standard validation.

We thank the Reviewer for the questions on CRISPR validation, as this allows us to further refine our analysis and improve the clarity of how we present our data. We respectfully disagree with the Reviewer's assertion that the clones defined by CRISPR and ReDeeM are quite different. In fact, the Adjusted Rand Index and the direct visualization demonstrate good agreement.

First, as a context for these experiments, it is critical to note that the KP tumors develop from one single cancer cell over a few months rather than the decades over which HSCs persist (Yang et al. 2022). Thus, this represents a far greater challenge for the ReDeeM approach than does the problem of human hematopoiesis. This stringent design allows us to explore whether mitochondrial lineage tracers can provide lineage information even at the subclonal level when there would be far less opportunity to accumulate new mitochondrial (or nuclear genome) mutations. The fact that we see clear statistical signals of consistency provides a solid benchmark for ReDeeM to track subclones or decipher genealogies (in a scenario where we detect less mitochondrial DNA mutations than we would otherwise for studies of human hematopoiesis).

We have specifically calculated the Adjusted Rand Index (ARI) as requested (using `adj.rand.index`), observing values that suggest good consistency between CRISPR and ReDeeM methods (**RTR-2 Fig. 1a**). To provide an unbiased assessment, we defined the clonal clusters at various resolutions for both CRISPR- and ReDeeM-based clones for each tumor (shown as each dot on the boxplot). As expected, the ARI values are dependent on different clustering resolutions and they vary in different samples. Nonetheless, all ARI values fall within a reasonable range (0.2 to 0.7), indicative of robust agreement at a subclonal level between the two orthogonal lineage tracers. It is important to note, however, that the clustering resolution is an arbitrary parameter. The reason we chose not to visualize a direct (sub)clonal cluster comparison in our earlier manuscript version was to avoid an arbitrary presentation, rather than failing to show this because the match is too bad, as the Reviewer suggests. Here, we visualize ReDeeM- and CRISPR-based (sub)clonal clusters on the same embedding using the examples of Tumor-1 and Tumor-9, which clearly illustrates good agreement (**RTR-2 Fig. 1b**). We also wish to point out that AOC is a single-cell resolution metric that does not require an arbitrary clustering parameter, and thus provides an unbiased view on how the lineage relatedness are supported across different points on the tree (**Extended Data Fig. 5f, 6a**). We agree with the Reviewer that the cluster-level comparison is helpful to provide an intuitive presentation, as we now provide. In our revised manuscript, we report both AOC (single-cell level metrics) and ARI (cluster level metrics) as well as the direct visualization to provide a comprehensive and informative assessment of the benchmarks we have performed (**Page 6 [L165-168], 44-45 [L1451-1455], revised Extended Data Fig. 6**).

RTR-2 Fig. 1 (Revised Extended Data Fig. 6). Clonal cluster level comparison between CRISPR-based lineage tracer and ReDeeM. a, adjusted rand index for clonal cluster consistency across 10 tumors (T1-T10). Various louvain cluster resolutions are tested (presented as each dot). **b**, Direct visualization of the clonal cluster consistency on CRISPR-based embedding (T1 and T9 are shown). Colors indicate either Mito-based or CRISPR-based cluster identification using one cluster resolution.

R2-4

Detail on Original Point 3 (LoY): I follow the arguments by the authors, but if LoY is not giving rise to clonal expansion here, this experiment is not suited as a gold standard validation of ReDeeM.

The causes and consequences of mLOY remain unclear, making it an area of considerable interest for further investigation. Our primary motivation for conducting the LOY analysis is to provide new biological insights, rather than to only serve as a validation. Our analysis suggests that mLOY likely occurs independently multiple times and is enriched in cells with higher fitness scores which advances our understanding of mLOY.

We agree with the Reviewer on the importance of validation. We have incorporated multiple lines of supportive evidence including newly added analyses that collectively contribute to a comprehensive validation, as detailed in the response to R2-1. For enhanced clarity, in the revised manuscript, we have included notes in the Methods section that systematically discuss both technical benchmarks and biological validations, underscoring the validity of ReDeeM (**Page 49-50 [L1606-1661]**).

R2-5

Detail on Original Point 5 (Validation): Fundamentally not addressed. The ReDeeM method needs a gold standard validation to sustain the central claim that it can be used to identify cellular genealogies (and clones). Currently only WGS of HSC derived colonies can provide such validation. If standard WGS lacks sensitivity, a WGS method with UMIs (or duplex sequencing?) needs to be used. Alternatively, they could split the colonies in two parts, perform standard WGS on half of the colony, Hash the other half with a barcoded surface antibody and perform ReDeeM. Genealogies from colony WGS (mitochondrial mutations, if possible), colony WGS (nuclear mutations) and single cells / ReDeeM (mitochondrial mutations) need to be compared.

We thank the Reviewer for the helpful comment, as this has led us to include WGS-derived mtDNA analysis that provides valuable additional validation in addition to all of the other evidence we have provided and also demonstrates the strongly enhanced mutation detection ability of ReDeeM using empiric data. In the revised manuscript, we carefully analyzed a single colony WGS dataset derived from a patient with a myeloproliferative neoplasm (MPN) due to a JAK2 mutation. Notably, all confident mtDNA mutations show great consistency with the nuclear genome based clonal and subclonal structures (see details below). However, it is also important to acknowledge and underscore the inherent limitations of single colony WGS, particularly concerning the detectability of mitochondrial mutations (see details below).

We reanalyzed the recently published WGS data from 42 single colonies from an MPN patient with a known JAK2 V617F somatic mutation, which serves as an ideal case to validate both clonal and subclonal consistency (New **Extended Data Fig. 15a**) (Van Egeren et al. 2021). This dataset is deeply sequenced and the mitochondrial genome is well covered (**Extended Data Fig. 15b-c**). We used the nuclear genome somatic mutations to infer a phylogenetic tree and compared mitochondrial mutations against the tree. Of note, it is challenging to accurately detect mitochondrial DNA mutations in WGS data because most genuine mitochondrial mutations are of low frequency, as observed in this study and reported by others, which makes them hard to differentiate from PCR/sequencing errors (**Extended Data Fig. 3**, Bi et al. 2020; Sanchez-Contreras et al. 2023a). We grouped candidate mitochondrial mutations based on variant allele frequencies and assessed whether the group of mutations was reliable based on the mutational signature compared to the expected enrichment pattern in C > T and T > C transitions (Ju et al. 2014). As anticipated, the lower the variant allele frequency, the weaker the mutational signature pattern is (**Extended Data Fig. 15e**). We pinpointed the top 5% of mutations (comprising 41 mutations) that display clean enrichment patterns, classifying them as confident mtDNA mutations. After excluding homoplasmic mutations (N=36, present in all colonies) and singular mutations (N=2, found in one colony), three confident heteroplasmic somatic mutations remain. Strikingly, all these three confident mutations align almost perfectly with the lineage tree inferred from the nuclear genome (**Extended Data Fig. 15f-g**). Specifically, 15562A>G and 14581T>C clearly mark the expanded JAK2 mutant clade, while 5237G>A is further restricted to a subclade of the JAK2 expansion, providing subclonal information. The combination of these mutations together provides robust lineage information. Indeed, ReDeeM utilized multiple mutation combinations and thus can be resilient to a certain level of mutation dropout (**RTR-2 Fig. 4**).

Furthermore, we examined the less confident (less stringent) mitochondrial mutations. Although the signal is much noisier, they successfully reveal the closer distance (i.e., more shared mutations) within the expanded JAK2 mutant clone (**Extended Data Fig. 15h**). Therefore, despite the limited ability to detect mtDNA mutations from WGS, there is clear consistency with the nuclear-based tree at both the clonal and subclonal levels.

We wish to also emphasize that a direct comparison between mitochondrial mutations and nuclear genome mutations in single colony WGS is not possible due to inherent limitations in detectability. As we have discussed extensively, detecting true somatic mutations from the nuclear genome is considerably easier than detecting those in the mitochondrial genome. If a mutation is localized within the nuclear genome of a given cell, we expect a variant allele frequency (VAF) of 50%, reflecting a heterozygous state. In the post-expansion WGS data, once germline mutations are excluded, the search primarily targets heterozygous mutations with a VAF of 50% (**Extended Data Fig. 3a** top panel). This methodology is both reliable and efficient. In contrast, mutations in the mitochondrial genomes, which have hundreds to thousands of copies in the single starting cell, often begin with a markedly low initial variant frequency, perhaps around 1% or even lower. Consequently, despite the cellular expansion into a single colony, the mitochondrial mutations most likely preserve this low frequency (**Extended Data Fig. 3a** bottom panel). In our simulations, even without factoring in PCR errors, these low-frequency mitochondrial mutations cannot be easily distinguished from sequencing errors (**Extended Data Fig. 3c**). This explains the observed high background noise, or weak mutational signature of mitochondrial mutations derived from WGS, especially as the variant allele frequency gets lower (**Extended Data Fig. 15e**).

As a comparison, ReDeeM, due to the incorporation of eUMI, efficiently corrects errors at the single molecule level (**Extended Data Fig. 3d,e**). We examined all 3,808 mitochondrial mutations identified by ReDeeM from one donor. All of these mutations, regardless of variant allele frequency, show very clean mutational signatures enriched in C>T and T>C transitions, as expected (**Extended Data Fig. 15i-j**). Therefore, ReDeeM is able to detect true mitochondrial mutations with much higher sensitivity and accuracy, including the low-frequency ones, compared to single colony WGS.

We are grateful for the Reviewer's suggestions concerning alternative experiments since they led us to provide more discussions on the limitations of current technologies and what is possible in the future. We have deliberated thoroughly on the feasibility and potential strategies to achieve single-colony WGS with UMI and half-colony cell hashing in conjunction with ReDeeM. To our knowledge, several major hurdles persist, making it unclear whether this endeavor can be achieved. First and foremost, HSPC colonies cease growth upon maturity within ~10 divisions, leaving each colony with a very limited number of cells, often around or fewer than 2,000, which complicates both proposed approaches. Below are specific discussion points.

- Duplex sequencing is a great tool to achieve high-sensitivity mutation calling. However, it typically requires hundreds of nanograms of DNA, a quantity significantly greater than what is available from a single colony (Sanchez-

Contreras et al. 2023b; Arbeithuber et al. 2020). Low-input WGS protocols that employ UMIs to achieve ultra-high sensitivity are in development. Several emerging technologies aim to improve WGS duplex-seq but still have limitations (e.g., the restriction enzyme- dependent fragmentation biases and limited coverage, etc) and none has been tailored specifically for the mitochondrial genome yet in this low-input setting (Abascal et al. 2021; Bae et al. 2023). To achieve this goal, new UMI-based protocols need to be developed and optimized to be amenable to low-input DNA material and facilitate unbiased fragmentation and enrichment of the mitochondrial genome, while also preserving the quality of the nuclear genome library for WGS. While this might be an interesting area for future investigation, this stands on its own right as a major technical contribution and is out of the scope of this manuscript.

- Half-colony cell hashing plus ReDeeM is an intriguing idea. However, there are several major technical challenges. The ReDeeM protocol typically requires 100-200K cells to start with, and thus a couple of hundred half-colonies need to be pooled. Each half-colony needs to be stained by a unique oligo-conjugated antibody separately. Given the large number of colonies, in-house preparation of a couple of hundred efficient and balanced hashing antibodies requires careful optimizations and thorough validations given that commercial hashing antibodies are likely not a good option in this case. Moreover, the washing steps for each half-colony after staining are very challenging, given the limited cell numbers (about 1000 cells each). The cell loss, hashing contamination, and unbalanced pooling are considerable concerns. Establishing a robust method for this would be a valuable independent contribution. We appreciate the Reviewer's suggestion, but as is clear from the open challenges in performing such studies, this is out of the scope of the current manuscript.
- We also would like to kindly note that the proposed WGS with UMI may lead to a notable further increase in sequencing costs on top of the large expenses of regular WGS. The need for multiple reads for consensus variant calling with each UMI in the mitochondrial genomes can make the total number of reads associated with this method greatly surpass those of regular WGS. For a sample size of one hundred colonies, it's estimated that around 50-100 billion reads, or possibly more, sequencing reads might be required (approximately \$200,000 or potentially more).

In addition to the technical hurdles above, conceptually it is important to note that the single colony assay, while valuable, carries inherent limitations. It operates on the premise that no alterations take place during *ex vivo* growth and that this growth is consistently balanced. These foundational assumptions remain to be fully validated. While we believe the assay is suitable for nuclear genome WGS, comparing it with the mitochondrial genome may not be appropriate because the mitochondrial genome can accumulate a greater number of subclonal somatic mutations, owing to its higher mutation rate, even during *ex vivo* growth to some extent as we have previously shown (see Fig. 1, Ludwig et al., *Cell*, 2019). This suggests that mtDNA mutation data could offer additional unique insights into subclonal (*ex vivo*) lineages, compared to nuclear genome mutations. In our opinion, it might be worth reconsidering how we characterize the single colony

WGS as the 'gold standard' since the nuclear genome may lack true subclonal information that mitochondria can provide in some scenarios.

Altogether, while WGS is constrained by sensitivity, it provides clear validation on both clonal and subclonal level. This strengthens our confidence when considered alongside various other lines of evidence we present (see response to R2-1).

We greatly appreciate the Reviewer's suggestions to ensure rigor in validating ReDeeM. We have incorporated the WGS analysis into our revised manuscript and provided more discussion on limitations of current technologies and future directions (**Page 6 [L168-174], 45-47 [1456-1533]**).

R2-6

Additional question: I noticed that in figure 5c, the expanded clades have quite long distances in the dendrograms. Is this just a consequence of the display method? Or does it mean that these cells are more dissimilar within a clade, compared to cells from the smaller clades observed in the young individuals? If the latter, would it be appropriate to refer to them as an "expanded clade", or rather as "residual cells not part of any clone"? I apologize for overlooking this before.

The phylogenetic trees in our manuscript are cladograms (for example in Fig. 5c), essentially focusing on the topology of the tree structure, where we can learn a lot from the hierarchical relationship, but we do not draw conclusions from the branch length. We are cautious in inferring branch lengths due to the challenges presented by mitochondrial mutation dynamics which could confound the inference of evolutionary time. Similarly, in previous CRISPR lineage tracing efforts, cladograms have been widely employed without interpretation of branch lengths due to the challenges of Cas9 editing dynamics. The cladograms have consistently proven to be robust and informative by us and other groups (Jones et al. 2020; Quinn et al. 2021; Yang et al. 2022; Raj et al. 2018; Zafar, Lin, and Bar-Joseph 2020).

Therefore, we limit ourselves to only drawing conclusions based on the topology from the tree structure, including identifying expansions of specific clones. The clonal expansions were estimated with two methods: clone-based or clade-based, both of which are independent from branch lengths. For the clone-based method, the clones are identified by cutting the tree topology and the clonal sizes are examined (**Methods**). The Shannon diversity indexes are also computed to measure the global clonal diversity, which are robust with various clonal clustering parameters (**Extended Data Fig. 13b-c**). For the clade-based method, we applied a statistical test for the expansion clades as previously described from the *Cassiopeia* package (Yang et al. 2022). Briefly, we compared the number of cells contained in the subclade to its direct "sisters" and computed a probability of this observation under neutral selection with a coalescent model, which reflects local topology. In Fig. 5c-f, the expansions we showed were supported by both methods. For example, the expansion A is a large clone, and it shows significant expansion against the null hypothesis of a neutral selection.

We have clarified our visualization and expansion identification methods in our revised manuscript. In the future, we believe further incorporating mitochondrial dynamic modeling will potentially enable confident branch length inference, which will be valuable to allow more analysis. We added more to discuss our limitations and future possibilities in the revised manuscript (**Page 16 [L492-499], 42 [L1372-1373]**).

Referee #3 (Remarks to the Author):

The authors have responded adequately to my comments and suggestions by performing additional experiments and analyses. They have also performed substantial additional work and analyses in response to the comments of other reviewers. Overall, their revised and improved manuscript would be of substantial interest to the field.

One minor comment is that many would not consider the FLT3-V592I to be a driver of CHIP. The variant has been reported in some cancers, but it is not recurrent in CHIP. The authors may want to clarify this.

We thank the Reviewer for the supportive comments and all the valuable suggestions which have greatly improved our manuscript.

We agree with the Reviewer regarding the nature of the FLT3-V592I variant, and have revised the manuscript to remove our mention of FLT3-V592I as a driver CHIP mutation.

Referee #4 (Remarks to the Author):

The authors have addressed all my previous comments well.

We appreciate the Reviewer's helpful and supportive comments which have led us to significantly improve our manuscript.

I saw the authors have released the scRNA and scATAC data on figshare and raw reads on GEO, which are very good. On top of this, I would strongly suggest the authors release more intermediate data, especially the mitochondrial matrices of reference and alternative alleles and the called variants list. If the authors can share the analysis notebooks, it will also be very helpful to readers. I believe this will better ensure the reproducibility and more accessible re-analysis of the data from this new technology.

We thank the Reviewer for the valuable suggestions as these help us to further improve our data accessibility and ensure the reproducibility of our analyses.

We have now provided the mutation calling table after ReDeeM-V which includes variant call lists and consensus information. We have also shared the ReDeeM-R object to figshare, which includes 1) cell-mtDNA mutation count matrix, 2) cell-mtDNA mutation binary matrix, 3) cell-cell distances, 4) inferred phylogenetic tree, etc.

We will make sure to share the analysis notebooks together with the publication once we finalize all of the analyses for this manuscript.

Additionally, for the redeemR package, please provide some demonstration vignettes. It is very hard to start using it.

We have now included the tutorial document Getting started in the Github repository. It included the most essential steps involved in ReDeeM R analyses. We will continue working on it to add more modules and further improve the readability of the documents, as we prepare to have this work published. We will make sure to provide thorough documentation with the publication.

Some minor comments:

We thank the Reviewer for pointing these issues out out.

- line 1352 (p42): the equation didn't show properly.

We have corrected this and made sure the equation now is appropriately displayed.

- Fig. 3: panel d and e are not matched with the order in the captions.

We have fixed the order in the figure legend (**Page 25-26 [L752-754]**).

- Fig. 4b/c, the output activities of T1 and T2 are relative values, right? Just wonder if the author may have an estimation of the total cells in each time point or if it is assumed to have a similar-sized population.

This is a great question. Yes, the output activities are relative values. We measured the clonal output level by counting the number of progenies for each HSC clonal group followed by normalization with HSC clone size (the number of HSCs per clonal group). We clarified this in the revised Methods section (**Page 53 [L1759-1761]**).

Referee #5 (Remarks to the Author):

As requested by the Editor, I have limited my comments to the mtDNA analysis, and pass no comment on the significance of the findings for our understanding of hematopoiesis.

The techniques presented are an evolution of published methods that have emerged from the Sankaran lab in the last 3 years – of note, the median mtDNA coverage of ~50x resembles coverage using a less complex protocols PMID: 37386249, PMID: 36792778, PMID: 35210612

We thank the Reviewer for bringing up this question as this allows us to clarify and further highlight the technical advantages of ReDeeM compared to our previous protocols.

In this manuscript, we have updated our approach to measuring coverage, adopting a more stringent and intuitive method to enhance clarity. We sincerely apologize for any confusion arising from this transition. In our previous reports, we counted the unique reads from paired-end sequencing twice, counting Read1 and Read2 separately. This was done because many mutations are covered by only one of the reads in our prior approaches, and we used this data to model strand biases. In contrast, ReDeeM is a single-molecule-based method, counting all reads with the same eUMI only once as one molecule. Therefore, in this manuscript 50x indicates there are 50 unique molecules covering one position in one cell. This metric is more intuitive and readily interpretable, as ReDeeM's variant calling is grounded in single-molecule detection. Consequently, this method of counting is more stringent, and it would be approximately 100X or more if the previous counting method is used. Of note, the mitochondrial content can vary significantly across different samples, and thus to provide a fair comparison, we have applied the ReDeeM and mtscATAC protocols on the same sample and used the same method to measure the coverage. We have observed about a 3-fold increase in coverage using the new protocol (**Fig. 1b, Extended Data Fig. 4c**).

More importantly, our goal is to increase the sensitivity of confident mutation calling to improve the lineage tracing performance. To that end, ReDeeM has improved this calling substantially by more than 10-fold (**Fig. 1c, e, f**). Indeed, in our previous methods, we had limited efficiency to distinguish true mutations from artifacts due to errors in PCRs and sequencing. As a result, we previously only had confidence in a subset of mutations that are abundant and not susceptible to strand biases. In contrast, ReDeeM generates a specific mtDNA library for deep sequencing and this allows single-molecule level consensus mutation calling which is designed to remove these errors and reveal the true signals with high sensitivity.

In summary, ReDeeM enables a 3-fold improvement in mtDNA coverage and more than 10-fold improvement in confident mutation detection. We have revised the manuscript to clarify the coverage measurement differences and further stressed the technical advantages of ReDeeM. We have provided a systematic comparative analysis of ReDeeM, mtscATAC, and MAESTER in the Methods section. (**Extended Data Fig. 4, Methods, Page 40-41 [L1260-1337]**).

The data raise a number of questions which are important to resolve if we are to have confidence in the biological conclusions about hematopoiesis:

R5-1

Q1. The eUMI collision rate seems high at 3% - is this 'random' collision or does it imply the biological relatedness to be expected? If were the collisions more common in related cell types, are they due to mutation hot-spots? Were there number of collisions in the same sequencing read exactly as predicted by random mutations? All of these would influence whether we can use mtDNA mutations and inert/neutral lineage tracers.

We thank the Reviewer for bringing up the concern about eUMI collision rate, since this guided us to further clarify how we use eUMIs and the foundational assumptions in our work. Furthermore, this has now enabled us to highlight the essential role of eUMIs in ReDeeM.

eUMI, or endogenous unique molecular identifier, is composed of a single cell barcode plus starting and ending sites of a sequencing read that determine the original single molecule giving rise to a particular read. This identification allows us to perform consensus-based mutation detection, the key step to achieving high sensitivity. For example, if there are 4 reads that have the same cell barcode and have exactly the same starting and ending position, we infer that these 4 reads are amplified from the same molecule and thus only the variants that are consistent across those 4 reads are considered as real mutations. This strategy can efficiently remove PCR and sequencing errors and substantially improves mutation calling accuracy. The eUMI collision rate is to estimate to what extent we can trust eUMI labels for one molecule. Given one single cell, there are hundreds to a couple of thousand mitochondrial genomes, each is a 16,569 bp DNA molecule. Tn5 almost randomly cuts the mtDNA. The question is whether the probability of cutting exactly the same starting and ending position in two different mtDNA molecules, namely eUMI collision, is low enough to make such assumptions to use the eUMI to label a single molecule. In this simulation, we specifically considered the Tn5 cutting preference based on empirical data which slightly decreases the randomness of cutting sites. Nonetheless, the collision rate is 3%, which means 3% of reads are incorrectly labeled as the same molecule but in fact should be different molecules. We consider this collision rate quite low because the impact is negligible. As illustrated in **RTR-2 Fig. 2**, when two different molecules are incorrectly labeled as one, the consensus mutation calling tends to fail due to the apparent inconsistency, and thus leads to false negatives, or the dropout of a true mutation. 3% of mutation dropout due to eUMI collision will have negligible effects, as further modeled below in response to R5-2 (**RTR-2 Fig. 4**).

As clarified above, eUMI collision is a metric applied at the experimental level that boosts mutation detection. The eUMI collision happens when we treat the cells with Tn5, and therefore this collision would not be affected by biological variation, such as differences among cell types.

In addition, we wish to address another potential concern about "mutation collisions." This phenomenon occurs when identical mutations spontaneously manifest in two unrelated cells purely by coincidence. It is challenging to explicitly measure the rate of mutation collision, but we have modeled the probability of "mutation collision" by random simulation. From ReDeeM data, we typically detect around 10 mutations in each single cell across the mitochondrial genome that is 16,569 bp in length. By random simulation, the probability that two cells have 1 shared mutation

by chance is 0.6%. Notably, ReDeeM uses the combination of multiple mutations with a median of 4 mutations that are shared with their nearest neighbors. The probability of sharing 4 mutations by coincidence would be extremely low, 1.4×10^{-11} (RTR-2 Fig.3). In the following responses to R5-3, we have further examined the mutation distributions across the mitochondrial genome. We did not observe specific enrichment or depletion of mutations on certain mitochondrial genes or regions, and it is not significantly affected by differentiation trajectories or cell types (please see more details in R5-3)

We have accordingly revised our manuscript to clarify the eUMI collision definition and its potential impact, as well as added more discussion about mutation collision to prevent any confusion about this (Page 38[L1215-1218, 1226-1233], 42-43 [1374-1384]).

RTR-2 Fig. 2 (Methods). Schematics of eUMI collision. When two different molecules are incorrectly labeled as one, the consensus mutation calling tends to fail due to the apparent inconsistency, and thus it leads to false negatives, or the dropout of a true mutation.

RTR-2 Fig. 3 Histogram of the number of shared mutations between nearest neighbors for cells in Young-1 dataset. The probability by chance is calculated by randomizing 10 mutations across the 16,569 bp mitochondrial genome per cell.

R5-2

Q2. To what extent are the inferred dynamics of clonality really a reflection of the mtDNA dynamics in different cell types/lineages (eg mtDNA turnover leading to drift in heteroplasmy levels), as opposed to cell-type dynamics which is what the authors conclude. This could be addressed by comparing homoplasmic to heteroplasmic variants. If mtDNA dynamics are not important, both should yield the same results.

We thank the Reviewer for bringing up the important point on mitochondrial dynamics, as this allows us to perform more analyses to benchmark the impact of mitochondria drift, highlight our method to mitigate the biases, and discuss more about the potential limitations.

To achieve robust lineage tracing, ReDeeM employs meticulous data preprocessing steps. This involves binarizing mitochondrial mutation counts, a data transformation step ensuring that lineage distances between cells are determined solely by the number of shared mutations, without being impacted by fluctuations in heteroplasmy levels. This approach helps mitigate potential biases stemming from quantitative drift due to mitochondrial dynamics. We have added more clarification and discussion in our revised manuscript (**Methods**).

While our method has reduced the direct impact of heteroplasmy level fluctuations, the drift in heteroplasmy level could lead to mutation detection dropout, i.e., false negatives, an issue that cannot be readily addressed by binarization. However, in principle, ReDeeM has the capability to withstand a certain degree of dropout. This is because ReDeeM uses combinations of mutations to infer lineage connections rather than on a singular mutation, benefiting from the enhanced mutation detectability. Theoretically, multiple mutations can compensate one another during partial dropout that minimizes the impact of false negatives. To validate this, we conducted a mutation dropout simulation. Essentially, we proposed that there is some chance (ranging from 10% to 50%) for a cell to lose a mutant allele copy for every mutation present due to drift or detection challenges. By repeatedly simulating mutation dropouts, we measured the correlation

of cell-to-cell lineage distances between different independent dropout simulations. As anticipated, even with a 50% dropout rate, the correlation remained strong (with a median Pearson's correlation of 0.57). This suggests that ReDeeM can accommodate a certain extent of mutation dropout, ensuring robust lineage inference.

RTR-2 Fig. 4 Simulation of the robustness of cell-cell distance by mtDNA mutation dropout. mtDNA mutations have been partially dropped with various probabilities modeled by binomial distribution. The Pearson's correlation coefficient of cell-cell distances between independent drop-out simulations are shown. The randomly reshuffled mutation matrix is used as the background.

We agree with the Reviewer's suggested analysis in principle, but in fact all homoplasmic mutations are present in all cells and therefore do not provide lineage information. There are no mutations that are "homoplasmic" in a subset of cells in our normal hematopoiesis datasets. Nonetheless, to address the Reviewer's point from a slightly different angle, we investigated the degree of variation in mtDNA mutation heteroplasmy levels across cells before the binarization, using young-1 data as an example (**RTR-2 Fig. 5**). The 32 "homoplasmic variants" are present in all cells and show no variation with a median heteroplasmy level of 0.99 (or 99%), and Interquartile range (IQR)=0. Of the 4,788 "heteroplasmic variants", which are present in a subset of cells, the majority also exhibited minimal heteroplasmy variation (4,601 mutations had an IQR<0.2, with a median heteroplasmy level of 0.02). We further analyzed the 187 mutations with an IQR>0.2 by examining their heteroplasmy level variations across HSCs, early progenitors, late progenitors, and differentiated cells, focusing on mutations present in at least 3 cells in each category. Most of these mutations did not display significant changes in heteroplasmy levels across the differentiation stages. These analyses suggest that the majority of mtDNA mutations have limited fluctuations in levels of heteroplasmy across cells. For the few that do exhibit variations, we have not seen functional selection that influences cell fate decisions during differentiation in normal hematopoiesis.

Taken together, we showed that 1) there are limited fluctuations in mtDNA mutation heteroplasmy levels in our datasets; 2) the binarization of the mutation matrices can further mitigate the potential biases due to quantitative fluctuations; 3) ReDeeM exhibits resilience to a certain extent of mtDNA mutation dropout by leveraging combinations of multiple mutations.

Finally, we completely agree with the Reviewer that mitochondrial dynamics is an important consideration and should be kept in mind in future analyses, particularly in studying the bottlenecks and selection in the context of diseases. We have revised our manuscript accordingly to clarify the above points and discussed limitations (**Page 16 [L492-499], 42 [L1356-1360], 47-48 [L1535-1570]**).

RTR-2 Fig. 5 mtDNA mutation variation and frequency in young-1 dataset. (a) Summary of mutations with different levels of variation in heteroplasmy level (as fraction). IQR: Interquartile Range. (b) Heteroplasmy level change during differentiation for mtDNA mutation with high IQR.

R5-3

Q3. To what extent do the mtDNA variants drive the behaviour of the clones? This is an important analysis to do given the rich dataset. For example, have the authors compared silent vs non-silent variants? Or variants in sites known to influence mtDNA replication. Again, if some mtDNA variants do influence the frequency of detecting mutations (eg some tend to be propagated, some selected against – which I would expect), then we cannot consider them neutral tags and will need to incorporate this into the interpretation.

We are grateful for the Reviewer's constructive suggestion that led us to evaluate the mtDNA mutation neutrality more carefully and incorporate functional level analyses. We agree that assessing the potential functional impact of mtDNA mutations is critical and our dataset provides rich information to perform this analysis. We used the Young-1 dataset as an example (given that this dataset has the largest representation of cell types), and we performed the following new analyses to address the Reviewer's question.

To begin with, we analyzed the mtDNA mutation distributions across the mitochondrial genome and observed a fairly uniform distribution spanning different genes and non-coding regions. We annotated all mitochondrial coding genes and the D-loop region which contains regulatory

elements for mtDNA replication and transcription. No significant mutation enrichment or depletion was observed based on the analysis of specific mtDNA regions (**Extended Data Fig. 16a**).

Next, we classified all mutations into 4 categories: missense, nonsense, synonymous, and non-coding. All these mutation types were observed (**Extended Data Fig. 16a**). To measure the potential functional selections on mitochondrial mutations, we computed the dN/dS ratios, a normalized ratio of nonsynonymous to synonymous mutations (a score of 1.0 represents overall neutrality; a ratio <1.0 suggests an excess of negative selection over positive selection and vice versa for $dN/dS >1.0$). Based on this metric, we surveyed the degree of functional selections for mitochondrial mutations with different single-cell heteroplasmy levels. For missense mutations, significant negative selections were observed only in homoplasmic mutations. For nonsense mutations, negative selections became noticeable when the single-cell heteroplasmy level exceeded 0.1. Notably, the majority of our mutations show low heteroplasmy levels (lower than 0.1). These mutations have a dN/dS ratio of around 1, suggesting overall neutrality (**Extended Data Fig. 16a-b**). This is consistent with our understanding that mutant alleles at low heteroplasmy levels contribute minimally among predominant wildtype alleles, and therefore, their impact on cellular fitness is likely negligible. We also extended our analysis to a subset of mutations that present in a relatively larger fraction of cells ($>1\%$) to examine their potential functional implications on cellular proliferation. Through dN/dS analysis, we did not detect significant selection for either missense or nonsense mutations suggesting overall neutrality (**Extended Data Fig. 16c**).

Lastly, we focused on the lineage-restricted mitochondrial mutations that are enriched in certain cell types or differentiation trajectories, from which we inferred lineage relationships and cell type origins. We examined the potential functional impact of these mutations. We found that these mutations can appear on all coding genes without significant preference across different cell types or differentiation stages. The dN/dS analysis also suggests no significant selection for these mutations, as well (**Extended Data Fig. 16d-e**).

In summary, these analyses suggest that most mitochondrial mutations utilized by ReDeeM exhibit minimal selection, as far as we can assess with the current depth of data we have, and can serve as neutral tags. However, we acknowledge that the functional impact is a very important aspect to consider and report. We added these new analyses to the revised manuscript. We have also incorporated a functional analysis module into ReDeeM-R to report all these annotations and selection statistics (**Page 5 [L155-158], 48-49 [L1571-1606]**).

As an aside, it does not surprise me that the mutation rate in stem cells is low compared to their progeny. Most of the mutations are likely to be polymerase errors, to the burden reflects the amount of mtDNA replication and cell turnover which is less in HSC than the cascade of proliferating progeny.

We thank the Reviewer for the insightful comments. We agree and this also serves as a great sanity check which suggests that the mutations identified by ReDeeM are informative in reflecting the process of differentiation of HSCs into more mature progeny, as we know from numerous

studies we and others have contributed to understanding hematopoiesis. This process happens in a short time scale and requires high-resolution lineage tracing.

References

- Abascal, Federico, Luke M. R. Harvey, Emily Mitchell, Andrew R. J. Lawson, Stefanie V. Lensing, Peter Ellis, Andrew J. C. Russell, et al. 2021. "Somatic Mutation Landscapes at Single-Molecule Resolution." *Nature* 593 (7859): 405–10.
- Arbeithuber, Barbara, James Hester, Marzia A. Cremona, Nicholas Stoler, Arslan Zaidi, Bonnie Higgins, Kate Anthony, Francesca Chiaromonte, Francisco J. Diaz, and Kateryna D. Makova. 2020. "Age-Related Accumulation of de Novo Mitochondrial Mutations in Mammalian Oocytes and Somatic Tissues." *PLoS Biology* 18 (7): e3000745.
- Bae, Jin H., Ruolin Liu, Eugenia Roberts, Erica Nguyen, Shervin Tabrizi, Justin Rhoades, Timothy Blewett, et al. 2023. "Single Duplex DNA Sequencing with CODEC Detects Mutations with High Sensitivity." *Nature Genetics* 55 (5): 871–79.
- Bi, Chongwei, Lin Wang, Yong Fan, Gerardo Ramos-Mandujano, Baolei Yuan, Xuan Zhou, Jincheng Wang, et al. 2020. "Single-Cell Individual Complete MtDNA Sequencing Uncovers Hidden Mitochondrial Heterogeneity in Human and Mouse Oocytes." *BioRxiv*. <https://doi.org/10.1101/2020.12.28.424537>.
- Fujino, Takeshi, Susumu Goyama, Yuki Sugiura, Daichi Inoue, Shuhei Asada, Satoshi Yamasaki, Akiko Matsumoto, et al. 2021. "Mutant ASXL1 Induces Age-Related Expansion of Phenotypic Hematopoietic Stem Cells through Activation of Akt/MTOR Pathway." *Nature Communications* 12 (1): 1826.
- Jones, Matthew G., Alex Khodaverdian, Jeffrey J. Quinn, Michelle M. Chan, Jeffrey A. Hussmann, Robert Wang, Chenling Xu, Jonathan S. Weissman, and Nir Yosef. 2020. "Inference of Single-Cell Phylogenies from Lineage Tracing Data Using Cassiopeia." *Genome Biology* 21 (1): 92.
- Ju, Young Seok, Ludmil B. Alexandrov, Moritz Gerstung, Inigo Martincorena, Serena Nik-Zainal, Manasa Ramakrishna, Helen R. Davies, et al. 2014. "Origins and Functional Consequences of Somatic Mitochondrial DNA Mutations in Human Cancer." *ELife* 3 (October). <https://doi.org/10.7554/eLife.02935>.
- Lareau, Caleb A., Sonia M. Dubois, Frank A. Buquicchio, Yu-Hsin Hsieh, Kopal Garg, Pauline Kautz, Lena Nitsch, et al. 2023. "Single-Cell Multi-Omics of Mitochondrial DNA Disorders Reveals Dynamics of Purifying Selection across Human Immune Cells." *Nature Genetics* 55 (7): 1198–1209.
- Lareau, Caleb A., Leif S. Ludwig, Christoph Muus, Satyen H. Gohil, Tongtong Zhao, Zachary Chiang, Karin Pelka, et al. 2021. "Massively Parallel Single-Cell Mitochondrial DNA Genotyping and Chromatin Profiling." *Nature Biotechnology* 39 (4): 451–61.
- Ludwig, Leif S., Caleb A. Lareau, Jacob C. Ulirsch, Elena Christian, Christoph Muus, Lauren H. Li, Karin Pelka, et al. 2019. "Lineage Tracing in Humans Enabled by Mitochondrial Mutations and Single-Cell Genomics." *Cell* 176 (6): 1325-1339.e22.
- Miller, Tyler E., Caleb A. Lareau, Julia A. Verga, Erica A. K. DePasquale, Vincent Liu, Daniel Ssozi, Katalin Sandor, et al. 2022. "Mitochondrial Variant Enrichment from High-Throughput Single-Cell RNA Sequencing Resolves Clonal Populations." *Nature Biotechnology* 40 (7): 1030–34.
- Mitchell, Emily, Michael Spencer Chapman, Nicholas Williams, Kevin J. Dawson, Nicole Mende, Emily F. Calderbank, Hyunchul Jung, et al. 2022. "Clonal Dynamics of Haematopoiesis across the Human Lifespan." *Nature* 606 (7913): 343–50.
- Myers, Robert M., Franco Izzo, Sanjay Kottapalli, Tamara Prieto, Andrew Dunbar, Robert L.

- Bowman, Eleni P. Mimitou, et al. 2022. "Integrated Single-Cell Genotyping and Chromatin Accessibility Charts JAK2V617F Human Hematopoietic Differentiation." *BioRxiv*. <https://doi.org/10.1101/2022.05.11.491515>.
- Nagase, Reina, Daichi Inoue, Alessandro Pastore, Takeshi Fujino, Hsin-An Hou, Norimasa Yamasaki, Susumu Goyama, et al. 2018. "Expression of Mutant Asxl1 Perturbs Hematopoiesis and Promotes Susceptibility to Leukemic Transformation." *The Journal of Experimental Medicine* 215 (6): 1729–47.
- Quinn, Jeffrey J., Matthew G. Jones, Ross A. Okimoto, Shigeki Nanjo, Michelle M. Chan, Nir Yosef, Trevor G. Bivona, and Jonathan S. Weissman. 2021. "Single-Cell Lineages Reveal the Rates, Routes, and Drivers of Metastasis in Cancer Xenografts." *Science* 371 (6532). <https://doi.org/10.1126/science.abc1944>.
- Raj, Bushra, Daniel E. Wagner, Aaron McKenna, Shristi Pandey, Allon M. Klein, Jay Shendure, James A. Gagnon, and Alexander F. Schier. 2018. "Simultaneous Single-Cell Profiling of Lineages and Cell Types in the Vertebrate Brain." *Nature Biotechnology* 36 (5): 442–50.
- Sanchez-Contreras, Monica, Mariya T. Sweetwyne, Kristine A. Tsantilas, Jeremy A. Whitson, Matthew D. Campbell, Brenden F. Kohn, Hyeon Jeong Kim, et al. 2023a. "The Multi-Tissue Landscape of Somatic MtDNA Mutations Indicates Tissue-Specific Accumulation and Removal in Aging." *ELife* 12 (February). <https://doi.org/10.7554/elife.83395>.
- . 2023b. "The Multi-Tissue Landscape of Somatic MtDNA Mutations Indicates Tissue-Specific Accumulation and Removal in Aging." *ELife* 12 (February). <https://doi.org/10.7554/eLife.83395>.
- Van Egeren, Debra, Javier Escabi, Maximilian Nguyen, Shichen Liu, Christopher R. Reilly, Sachin Patel, Baransel Kamaz, et al. 2021. "Reconstructing the Lineage Histories and Differentiation Trajectories of Individual Cancer Cells in Myeloproliferative Neoplasms." *Cell Stem Cell* 28 (3): 514-523.e9.
- Velten, Lars, Benjamin A. Story, Pablo Hernández-Malmierca, Simon Raffel, Daniel R. Leonce, Jennifer Milbank, Malte Paulsen, et al. 2021. "Identification of Leukemic and Pre-Leukemic Stem Cells by Clonal Tracking from Single-Cell Transcriptomics." *Nature Communications* 12 (1): 1366.
- Xu, Jin, Kevin Nuno, Ulrike M. Litzemberger, Yanyan Qi, M. Ryan Corces, Ravindra Majeti, and Howard Y. Chang. 2019. "Single-Cell Lineage Tracing by Endogenous Mutations Enriched in Transposase Accessible Mitochondrial DNA." *ELife* 8 (April). <https://doi.org/10.7554/eLife.45105>.
- Yang, Dian, Matthew G. Jones, Santiago Naranjo, William M. Rideout 3rd, Kyung Hoi Joseph Min, Raymond Ho, Wei Wu, et al. 2022. "Lineage Tracing Reveals the Phylodynamics, Plasticity, and Paths of Tumor Evolution." *Cell* 185 (11): 1905-1923.e25.
- Zafar, Hamim, Chieh Lin, and Ziv Bar-Joseph. 2020. "Single-Cell Lineage Tracing by Integrating CRISPR-Cas9 Mutations with Transcriptomic Data." *Nature Communications* 11 (1). <https://doi.org/10.1038/s41467-020-16821-5>.

Reviewer Reports on the Second Revision:

Referees' comments:

Referee #2 (Remarks to the Author):

I appreciate that the authors have taken my concerns seriously and added useful analyses focused on controls.

In particular, the WGS of colonies experiment (Figure EV15) is supportive of the utility of mtDNA for phylogeny reconstruction. Together with figure EV3, it makes a good point that ReDeeM selects "true" mitochondrial mutations well. However, with regard to the conclusions regarding clone/lineage tracing, Figure EV15 contradicts a recent preprint by Chapman & Campbell (<https://doi.org/10.21203/rs.3.rs-3083262/v1>). C&C use a related, mathematically more sophisticated, but possibly less stringent, strategy to select 'correct' mitochondrial mutations from colony WGS data (compare Figure EV15e from this manuscript and Figure EV2a by Chapman et al). They reach the conclusions that the mutations selected by this process mostly do not reflect phylogeny (Chapman et al., figure 5a). I think it will be impossible to resolve this controversy during the review process of this paper, but I would appreciate if the authors could cite and comment on the preprint in their discussion.

The only direct comparison of a ground truth and ReDeeM included in the paper remains the CRISPR experiment. I appreciate that the adjusted rand index is now shown for several such experiments, as it provides a measure that's easy to understand and compare between studies. I would recommend to still include the CRISPR clonal labels as colors/labels under the dendrograms in figure EV5f and EV6a, as this makes it easier to visualize for readers what an ARI of 0.2-0.6 means.

I remain somewhat skeptical of ReDeeM, as the direct validations remain weak (ARI of 0.2-0.6 on a possibly not ideally selected CRISPR lineage tracing model). On the other hand, there are now multiple lines of indirect evidence supporting that ReDeeM captures phylogenetic signal (findings match expectations on hierarchies of blood progenitors, clone sizes in ageing, bias of ASXL1 mutant clones, phylogenetic signals of mitochondrial variants in colony data, etc).

Referee #5 (Remarks to the Author):

I must thank the authors for comprehensively tackling the points I raised. I realise this was a considerable amount of work. Overall, I am satisfied that the authors have addressed the points that I raised.

I have a few comments for the authors, and I would value a short response:

- By heteroplasmy level of 0.02, do they mean 2% of molecules or 0.2% of molecules with eUMIs in a single cell?
- It is very interesting that the heteroplasmic variants showed minimal variation. I would not have expected this given recent datasets showing considerable cell-cell variation in heteroplasmy (eg PMID: 32786181). It would be very interesting to compare the behaviour of synonymous and non-synonymous variants in this regard. Can the authors comment on this?
- Do the authors think the lack of any evidence of selection against mtDNA variants reflect the limited number of variants detected (power)? Or simply that there is no selection. This is also surprising given the citation above – admittedly this paper relates to a high level pathogenic heteroplasmic mutation.

Author Rebuttals to Second Revision:

Response to Reviewer Comments:

Referees' comments:

Referee #2 (Remarks to the Author):

I appreciate that the authors have taken my concerns seriously and added useful analyses focused on controls.

We are grateful to the Reviewer for their overall supportive comments. Please see below for point-by-point responses to the remaining questions raised.

In particular, the WGS of colonies experiment (Figure EV15) is supportive of the utility of mtDNA for phylogeny reconstruction. Together with figure EV3, it makes a good point that ReDeeM selects “true” mitochondrial mutations well. However, with regard to the conclusions regarding clone/lineage tracing, Figure EV15 contradicts a recent preprint by Chapman & Campbell (<https://doi.org/10.21203/rs.3.rs-3083262/v1>). C&C use a related, mathematically more sophisticated, but possibly less stringent, strategy to select ‘correct’ mitochondrial mutations from colony WGS data (compare Figure EV15e from this manuscript and Figure EV2a by Chapman et al). They reach the conclusions that the mutations selected by this process mostly do not reflect phylogeny (Chapman et al., figure 5a). I think it will be impossible to resolve this controversy during the review process of this paper, but I would appreciate if the authors could cite and comment on the preprint in their discussion.

We thank the Reviewer for the appreciation of our WGS data re-analysis and for bringing up the pre-print from Chapman, Campbell, and colleagues (<https://doi.org/10.21203/rs.3.rs-3083262/v1>), a valuable reanalysis of another single colony WGS dataset. By carefully looking at the data analysis presented, we found their observations are broadly consistent with our analysis of WGS data of the Van Egeren et al. data (see below). Notably, the Van Egeren et al. datasets were more deeply sequenced with a PCR free method. Therefore, compared to our analysis, the Chapman et al. datasets are likely even more prone to introduction of noises when analyzing mtDNA mutations at medium to low VAF variants. This is confirmed by the fact that the mutational signature for the medium to low VAF variants they called diverge from the canonical mitochondrial mutational signature. We added the citation for this work and included more detailed discussion in the main text and supplementary Note shown as noted below.

“These findings agree with a recent report, which shows agreement in high frequency mtDNA mutations with whole genome sequencing of colonies, but more noise in lower frequency mtDNA mutations⁴⁴ (Supplementary Notes).” [Main text]

“Our observations are broadly consistent with a recent report⁴⁴ from Chapman et al. that re-analyzed mtDNA mutations from single hematopoietic colony WGS. Consistent with our analyses

of the Van Egeren et al. data, the Chapman et al. data suggest that, with this method, low VAF mtDNA variants are contaminated by mutations with mutational signatures that diverge from those characteristic of mitochondria suggestive of sequencing artifacts or mutations acquired in vitro, while mtDNA mutations with higher VAF display the canonical mitochondrial mutational signature and show significant concordance with the phylogenetic trees derived from nuclear somatic mutations. Therefore, the use of mtDNA mutations detected at high VAF from single colony WGS data for lineage tracing are consistent between our analysis and that of Chapman et al. However, it is challenging to investigate rarer mtDNA mutations in single colonies. A more sensitive WGS method (such as UMI based WGS) will be needed to achieve sufficient detectability. Of note, ReDeeM provides substantially enhanced sensitivity for rare mtDNA mutations and we have demonstrated its capability of fine-scale lineage tracing through multiple lines of evidence.” [Supplementary Notes]

The only direct comparison of a ground truth and ReDeeM included in the paper remains the CRISPR experiment. I appreciate that the adjusted rand index is now shown for several such experiments, as it provides a measure that’s easy to understand and compare between studies. I would recommend to still include the CRISPR clonal labels as colors/labels under the dendrograms in figure EV5f and EV6a, as this makes it easier to visualize for readers what an ARI of 0.2-0.6 means.

We thank the Reviewer for their suggestion to include the ARI and their appreciation that this is now included. We agree that showing an intuitive visualization to help indicate what the ARI means would be helpful. Indeed, the scatter plot in extended Data Fig. g, h simultaneously shows both ReDeeM-based and CRISPR-based clonal clusters using an example clustering resolution, which serve as a straightforward visualization for readers. We also revised the Extended Data Fig. g,h by providing the exact ARI to directly connect the visualizations to ARI measures.

I remain somewhat skeptical of ReDeeM, as the direct validations remain weak (ARI of 0.2-0.6 on a possibly not ideally selected CRISPR lineage tracing model). On the other hand, there are now multiple lines of indirect evidence supporting that ReDeeM captures phylogenetic signal (findings match expectations on hierarchies of blood progenitors, clone sizes in ageing, bias of ASXL1 mutant clones, phylogenetic signals of mitochondrial variants in colony data, etc).

We thank the Reviewer for recognizing that our use of multiple lines of evidence provides support for the robustness of ReDeeM. We have also discussed the limitations of ReDeeM and the potential improvements we can make in the future, which we hope will inspire further work on this method.

Referee #5 (Remarks to the Author):

I must thank the authors for comprehensively tackling the points I raised. I realise this was a considerable amount of work. Overall, I am satisfied that the authors have addressed the points that I raised.

We thank the Reviewer for the supportive comments and suggestions. Please see below for the point-by-point responses.

I have a few comments for the authors, and I would value a short response:

- By heteroplasmy level of 0.02, do they mean 2% of molecules or 0.2% of molecules with eUMIs in a single cell?

We apologize for not making it clear before. 0.02 means 2% of molecules. We have revised the legend of Extended Data Fig. 4 to clarify this.

- It is very interesting that the heteroplasmic variants showed minimal variation. I would not have expected this given recent datasets showing considerable cell-cell variation in heteroplasmy (eg PMID: 32786181). It would be very interesting to compare the behaviour of synonymous and non-synonymous variants in this regard. Can the authors comment on this?

This is a very interesting question. As suggested by the Reviewer, we further broke down the analysis from Supplementary Fig. 11 into synonymous and non-synonymous mtDNA mutations. We found that both synonymous and non-synonymous show a similar heteroplasmy level on average and display a similar distribution of the variation across cells (measured by IQR).

This observation is likely due to the high sensitivity of ReDeeM that picked up a large number of low heteroplasmy mtDNA mutations compared to prior work (eg. PMID: 32786181). We think this can be well reconciled with the fact that higher heteroplasmy mtDNA mutations are more likely to display high variation due to functional purifying selection (Supplementary Fig. 11), while the lower frequency mutations contribute less to cell fitness and therefore are subject to a lesser extent of selection and thus are less variable.

a**b**
- Do the authors think the lack of any evidence of selection against mtDNA variants reflect the limited number of variants detected (power)? Or simply that there is no selection. This is also surprising given the citation above – admittedly this paper relates to a high level pathogenic heteroplasmic mutation.

We think the lack of evidence of selection is largely due to the investigation of low heteroplasmy mtDNA mutations. The number of mutations examined is large, in a range of hundreds to thousands and thus it is less likely that power was the issue.

We agree with the Rviewer that one of the major differences in this cited paper is the use of high level pathogenic heteroplasmy mtDNA mutations versus the use of non-pathogenic low level mutations. We believe heteroplasmy levels play a critical role in determining whether a mutation could lead to purifying selection. We think it would be interesting to further quantify and model the relationship between the heteroplasmy level and the level of selection in the future, particularly as we acquire additional data across more cell states.